# ROBUST ANGULAR SYNCHRONIZATION VIA DIRECTED GRAPH NEURAL NETWORKS

**Yixuan He** [*]**& Gesine Reinert**
Department of Statistics
University of Oxford
Oxford, United Kingdom
{yixuan.he, reinert}@stats.ox.ac.uk

**David Wipf**
Amazon
Shanghai, China
daviwipf@amazon.com

**Mihai Cucuringu**
Department of Statistics
University of Oxford
Oxford, United Kingdom
mihai.cucuringu@stats.ox.ac.uk

## ABSTRACT

The angular synchronization problem aims to accurately estimate (up to a constant additive phase) a set of unknown angles $\theta_1, \ldots, \theta_n \in [0, 2\pi)$ from $m$ noisy measurements of their offsets $\theta_i - \theta_j \mod 2\pi$. Applications include, for example, sensor network localization, phase retrieval, and distributed clock synchronization. An extension of the problem to the heterogeneous setting (dubbed $k$-synchronization) is to estimate $k$ groups of angles simultaneously, given noisy observations (with unknown group assignment) from each group. Existing methods for angular synchronization usually perform poorly in high-noise regimes, which are common in applications. In this paper, we leverage neural networks for the angular synchronization problem, and its heterogeneous extension, by proposing GNNSYNC, a theoretically-grounded end-to-end trainable framework using directed graph neural networks. In addition, new loss functions are devised to encode synchronization objectives. Experimental results on extensive data sets demonstrate that GNNSync attains competitive, and often superior, performance against a comprehensive set of baselines for the angular synchronization problem and its extension, validating the robustness of GNNSync even at high noise levels.

## 1 INTRODUCTION

The group synchronization problem has received considerable attention in recent years, as a key building block of many computational problems. Group synchronization aims to estimate a collection of group elements, given a small subset of potentially noisy measurements of their pairwise ratios $\Upsilon_{i,j} = g_i \, g_j^{-1}$. Some applications are • over the group SO(3) of 3D rotations: rotation-averaging in 3D computer vision (Arrigoni & Fusiello, 2020; Janco & Bendory, 2022) and the molecule problem in structural biology (Cucuringu et al., 2012b); • over the group $\mathbb{Z}_4$ of the integers $\{0, 1, 2, 3\}$ with addition mod 4 as the group operation: solving jigsaw puzzles (Huroyan et al., 2020); • over the group $\mathbb{Z}_n$, resp., SO(2): recovering a global ranking from pairwise comparisons (He et al., 2022a; Cucuringu, 2016), and, • over the Euclidean group of rigid motions $\text{Euc}(2) = \mathbb{Z}_2 \times \text{SO}(2) \times \mathbb{R}^2$: sensor network localization (Cucuringu et al., 2012a).

An important special case is *angular synchronization*, also referred to as *phase synchronization*, which can be viewed as group synchronization over SO(2). The angular synchronization problem aims at obtaining an accurate estimation (up to a constant additive phase) for a set of unknown angles $\theta_1, \ldots, \theta_n \in [0, 2\pi)$ from $m$ noisy measurements of their pairwise offsets $\theta_i - \theta_j \mod 2\pi$. This problem has a wide range

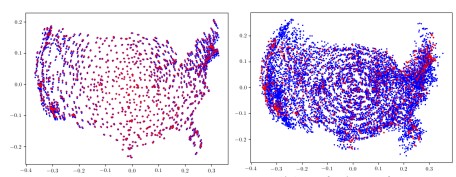

(a) Low noise.  (b) High noise.
Figure 1: Sensor network localization map.

---

[*]This work was partially done during an internship at Amazon.

of applications, such as distributed clock synchronization over wireless networks (Giridhar & Kumar, 2006), image reconstruction from pairwise intensity differences (Yu, 2009; 2011), phase retrieval (Forstner et al., 2020; Iwen et al., 2020), and sensor network localization (SNL) (Cucuringu et al., 2012a). In engineering, the SNL problem seeks to reconstruct the 2D coordinates of a cloud of points from a sparse set of pairwise noisy Euclidean distances; in typical divide-and-conquer approaches that aid with scalability, one first computes a local embedding of nearby points (denoted as *patches*) and is left with the task of stitching the patches together in a globally consistent embedding (Cucuringu et al., 2012a). Fig. 1 is an example of SNL on the U.S. map, where our method recovers city locations (in blue) and aims to match ground-truth locations (in red). Most works in the SNL literature that focus on the methodology development consider only purely synthetic data sets in their experiments; here we consider a real-world data set (actual 2D layout with different levels of densities of cities across the U.S. map), and add synthetic noise to perturb the local patch embeddings for testing the robustness to noise of the angular synchronization component.

An extension of angular synchronization to the heterogeneous setting is *k-synchronization*, introduced in Cucuringu & Tyagi (2022), and motivated by real-world graph realization problems (GRP) and ranking. GRP aims to recover coordinates of a cloud of points in $\mathbb{R}^d$, from a sparse subset (edges of a graph) of noisy pairwise Euclidean distances (the case $d = 2$ is the above SNL problem). The motivation for *k-synchronization* arises in structural biology, where the distance measurements between pairs of atoms may correspond to $k$ different configurations of the molecule, in the case of molecules with multiple conformations. In ranking applications, the $k = 2$ sets of disjoint pairwise measurements may correspond to two different judges, whose latent rankings we aim to recover.

A key limitation of existing methods for angular synchronization is their poor performance in the presence of considerable noise. High noise levels are not unusual; measurements in SO(3) can have large outliers in certain biological settings (cryo-EM and NMR spectroscopy), see for example Cucuringu et al. (2012b). Therefore, we need new methods to push the boundary of signal recovery when there is a high level of noise. While neural networks (NNs), in principle, could be trained to address high noise regimes, the angular synchronization problem is not directly amenable to a standard NN architecture due to the directed graph (digraph) structure of the underlying data measurement process and the underlying group structure; hence the need for a customized graph neural network (GNN) architecture and loss function for this task. Here we propose a GNN method called GNNSync for angular synchronization, with a novel cycle loss function, which downweights noisy observations, and explicitly enforces cycle consistency as a quality measure. GNNSync's novelty does not lie in simply applying a data-driven NN to this task, but rather in proposing a framework for handling the pairwise comparisons encoded in a digraph, accounting for the underlying SO(2) group structure, and designing a loss function for increased robustness to noise and outliers, with theoretical support.

Our main contributions are summarized as follows.
• We demonstrate how the angular synchronization problem can be recast as a theoretically-grounded directed graph learning task by first incorporating the inductive biases of classical estimators within the design of a more robust GNN architecture, called GNNSync, and then pairing with a novel training loss that exploits cycle consistency to help infer the unknown angles.
• We perform extensive experiments comparing GNNSync with existing state-of-the-art algorithms from the angular synchronization and $k$-synchronization literature, across a variety of synthetic outlier models at various density and noise levels, and on a real-world application. GNNSync attains leading performance, especially in high noise regimes, validating its robustness to noise.

## 2 RELATED WORK

### 2.1 ANGULAR SYNCHRONIZATION

The seminal work of Singer (2011) introduced spectral and semidefinite programming (SDP) relaxations for angular synchronization. For the spectral relaxation, the estimated angles are given by the eigenvector corresponding to the largest eigenvalue of a Hermitian matrix $\mathbf{H}$, whose entries are given by $\mathbf{H}_{i,j} = \exp(\iota \mathbf{A}_{i,j}) \mathbb{1}(\mathbf{A}_{i,j} \neq 0)$, where $\iota$ is the imaginary unit, and $\mathbf{A}_{i,j}$ is the observed potentially noisy offset $\theta_i - \theta_j \mod 2\pi$. Singer (2011) also provided an SDP relaxation involving the same matrix $\mathbf{H}$, and empirically demonstrated that the spectral and SDP relaxations yield similar experimental results. A row normalization was introduced to $\mathbf{H}$ prior to the eigenvector computation by Cucuringu et al. (2012a), which showed improved results. Cucuringu et al. (2012b) generalized this approach to the 3D setting $\text{Euc}(3) = \mathbb{Z}_2 \times \text{SO}(3) \times \mathbb{R}^3$, and incorporated into the optimization pipeline the ability to operate in a semi-supervised setting, where certain group elements are known a-priori. Cucuringu & Tyagi (2022) extended the angular synchronization problem to a heterogeneous setting, to the so-called

*k-synchronization* problem, whose goal is to estimate $k$ sets of angles simultaneously, given only the graph union of noisy pairwise offsets, which we also explore in our experiments. The key idea in their work is to estimate the $k$ sets of angles from the top $k$ eigenvectors of the angular embedding matrix $\mathbf{H}$.

Boumal (2016) modeled the angular (phase) synchronization problem as a least-squares non-convex optimization problem, and proposed a modified version of the power method called the Generalized Power Method (GPM), which is straightforward to implement and free of parameter tuning. GPM often attains leading performance among baselines in our experiments, and the iterative steps in the GPM method motivated the design of the projected gradient steps in our GNNSync architecture. However, GPM is not directly applicable to $k$-synchronization with $k > 1$ while GNNSync is. For $k = 1$, GNNSync tends to perform significantly better than GPM at high noise levels. Bandeira et al. (2017) studied the tightness of the maximum likelihood semidefinite relaxation for angular synchronization, where the maximum likelihood estimate is the solution to a nonbipartite Grothendieck problem over the complex numbers. A truncated least-squares approach was proposed by Huang et al. (2017) that minimizes the discrepancy between the estimated angle differences and the observed differences under some constraints. Gao & Zhao (2019) tackled the angular synchronization problem with a multi-frequency approach. Liu et al. (2023) unified various group synchronization problems over subgroups of the orthogonal group. Filbir et al. (2021) provided recovery guarantees for eigenvector relaxation and semidefinite convex relaxation methods for weighted angular synchronization. Lerman & Shi (2022) applied a message-passing procedure based on cycle consistency information, to estimate the corruption levels of group ratios and consequently solve the synchronization problem, but the method is focused on the restrictive setting of adversarial or uniform corruption and sufficiently small noise. In addition, Lerman & Shi (2022) requires post-processing based on the estimated corruption levels to obtain the group elements, while GNNSync is trained end-to-end. Maunu & Lerman (2023) utilized energy minimization ideas, with a variant converging linearly to the ground truth rotations.

## 2.2 Directed graph neural networks

Digraph node embeddings can be effectively learned via directed graph neural networks (He et al., 2022c). For learning such an embedding, Tong et al. (2020) constructed a GNN using higher-order proximity. Zhang et al. (2021) built a complex Hermitian Laplacian matrix and proposed a spectral digraph GNN. He et al. (2022b) introduced imbalance objectives for digraph clustering. Our GNNSync framework can readily incorporate any existing digraph neural network.

## 2.3 Relationship with other group synchronization methods

Angular synchronization outputs can be used to obtain global rankings by using a one-dimensional ordering. To this end, recovering rankings of $n$ objects from pairwise comparisons can be viewed as group synchronization over $\mathbb{Z}_n$. To recover global rankings from pairwise comparisons, GN-NRank (He et al., 2022a) adopted an unfolding idea to add an inductive bias from Fogel et al. (2014) to the NN architecture. Inspired by He et al. (2022a), we adapt their framework to borrow strength from solving a related problem. We adapt their "innerproduct" variant to $k$-synchronization, remove the 1D ordering at the end of the GNNRank framework, and rescale the estimated quantities to the range $[0, 2\pi)$. We also borrow strength from the projected gradient steps in GPM (Boumal, 2016) and add projected gradient steps to our GNNSync architecture. Another key novelty is that we devise novel objectives, which reflect the angular structure of the data, to serve as our training loss functions. The architectures are also very different: While in GNNRank the proximal gradient steps play a vital role from an unrolling perspective, and the whole architecture could be viewed as an unrolling of the SerialRank algorithm, here, although we borrow strength from the GPM method, the whole architecture is different from merely unrolling GPM. Furthermore, the baselines serve as initial guesses for the "proximal baseline" variant in GNNRank, but serve as input node features in our approach.

Other methods have been introduced for group synchronization, but mostly in the context of SO(3). Shi & Lerman (2020) proposed an efficient algorithm for synchronization over SO(3) under high levels of corruption and noise. Shi et al. (2022) provided a novel quadratic programming formulation for estimating the corruption levels, but again its focus is on SO(3). Unrolled algorithms (which are NNs) were introduced for SO(3) in Janco & Bendory (2022). While an adaptation to SO(2) may be possible in principle, as its objective functions are based on the level of agreement between the estimated angles and ground-truth, its experiments require ground-truth during training, usually not available in practice. In contrast, our GNNSync framework can be trained without any known angles.

## 3 Problem definition

The *angular synchronization* problem aims at obtaining an accurate estimation (up to a constant additive phase) for a set of $n$ unknown angles $\theta_1, \ldots, \theta_n \in [0, 2\pi)$ from $m$ noisy measurements of

their offsets $\theta_i - \theta_j \bmod 2\pi$, for $i, j \in \{1, \ldots, n\}$. We encode the noisy measurements in a digraph $\mathcal{G} = (\mathcal{V}, \mathcal{E})$, where each of the $n$ elements of the node set $\mathcal{V}$ has as attribute an angle $\theta_i \in [0, 2\pi)$. The edge set $\mathcal{E}$ represents pairwise measurements of the angular offsets $(\theta_i - \theta_j) \bmod 2\pi$. The weighted directed graph has a corresponding adjacency matrix $\mathbf{A}$ with $\mathbf{A}_{i,j} = (\theta_i - \theta_j) \bmod 2\pi \geqslant 0$. Estimating the unknown angles from noisy offsets amounts to assigning an estimate $r_i \in [0, 2\pi)$ to each node $i \in \mathcal{V}$. For computational complexity considerations, we randomly keep one of $\mathbf{A}_{i,j}$ and $\mathbf{A}_{j,i}$ as observed quantity and set the other of these to zero. Thus, at most one of $\mathbf{A}_{i,j}$ and $\mathbf{A}_{j,i}$ can be nonzero by construction; the other original entry can be inferred from $\mathbf{A}_{i,j} + \mathbf{A}_{j,i} = 0 \bmod 2\pi$.

An extension of the above problem to the heterogeneous setting is the *k-synchronization* problem, which is defined as follows. We are given only the graph union of $k$ digraphs $\mathcal{G}_1, \ldots, \mathcal{G}_k$, with the same node set and disjoint edge sets, which encode noisy measurements of $k$ sets $(\theta_{i,l} - \theta_{j,l}) \bmod 2\pi$, for $l \in \{1, \ldots, k\}, i, j \in \{1, \ldots, n\}$, of angle differences modulo $2\pi$. Its adjacency matrix is denoted by $\mathbf{A}$. The problem is to estimate these $k$ sets of $n$ unknown angles $\theta_{i,l} \in [0, 2\pi), \forall l \in \{1, \ldots, k\}, i \in \{1, \ldots, n\}$, simultaneously. Note that we are given only $\mathcal{G} = \mathcal{G}_1 \cup \cdots \cup \mathcal{G}_k$ and the value of $k$, and each edge in $\mathcal{G}$ belongs to exactly one of $\mathcal{G}_1, \ldots, \mathcal{G}_k$. To unify notations, we view the normal angular synchronization problem as a special case of the more general $k$-synchronization problem where $k = 1$.

## 4 LOSS AND EVALUATION

### 4.1 LOSS AND EVALUATION FOR ANGULAR SYNCHRONIZATION

For a vector $\mathbf{r} = [r_1, \ldots, r_n]^\top$ with estimated angles as entries, we define $\mathbf{T} = [(\mathbf{r}\mathbf{1}^\top - \mathbf{1}\mathbf{r}^\top) \bmod 2\pi] \in \mathbb{R}^{n \times n}$. Then $\mathbf{T}_{i,j} = (r_i - r_j) \bmod 2\pi$ estimates $\mathbf{A}_{i,j}$. We only compare $\mathbf{T}$ with $\mathbf{A}$ at locations where $\mathbf{A}$ has nonzero entries. We introduce the residual matrix $\mathbf{M}$ with entries

$$\mathbf{M}_{i,j} = \min\left((\mathbf{T}_{i,j} - \mathbf{A}_{i,j}) \bmod 2\pi, (\mathbf{A}_{i,j} - \mathbf{T}_{i,j}) \bmod 2\pi\right)$$

if $\mathbf{A}_{i,j} \neq 0$, and $\mathbf{M}_{i,j} = 0$ if $\mathbf{A}_{i,j} = 0$. Then our upset loss is defined as

$$\mathcal{L}_{\text{upset}} = \|\mathbf{M}\|_F / t, \tag{1}$$

where the subscript $F$ means Frobenius norm, and $t$ is the number of nonzero elements in $\mathbf{A}$. Despite the non-differentiablility of the loss function, using the concept of a limiting subdifferential from Li et al. (2020) we can give the following theoretical guarantee on the minimization of eq. (1); its proof is in Appendix (App.) A.1, where also the case of general $k$ is discussed.

**Proposition 1.** *Every local minimum of eq.* (1) *is a directional stationary point of eq.* (1).

For *evaluation*, we employ a Mean Square Error (MSE) function with angle corrections, considered in Singer & Shkolnisky (2011). As the offset measurements are unchanged if we shift all angles by a constant, denoting the ground-truth angle vector as $\mathbf{R}$, this evaluation function can be written as

$$\mathcal{D}_{\text{MSE}}(\mathbf{r}, \mathbf{R}) = \min_{\theta_0 \in [0, 2\pi)} \sum_{i=1}^{n} \left[\min(\delta_i \bmod 2\pi, (-\delta_i) \bmod 2\pi)\right]^2, \tag{2}$$

where $\delta_i = r_i + \theta_0 - \theta_i, \forall i = 1, \ldots, n$. Additional implementation details are provided in App. C.4.

### 4.2 CYCLE CONSISTENCY RELATION

For noiseless observations, every cycle in the angular synchronization problem ($k = 1$) or every cycle whose edges correspond to the same offset graph $\mathcal{G}_l$ ($k > 1$) satisfy the *cycle consistency* relation that the angle sum mod $2\pi$ is 0. For 3-cycles $(i, j, q)$, such that $\mathbf{A}_{i,j} \cdot \mathbf{A}_{j,q} \cdot \mathbf{A}_{q,i} > 0$, this leads to

$$(\mathbf{A}_{i,j} + \mathbf{A}_{j,q} + \mathbf{A}_{q,i}) \bmod 2\pi = (\theta_i - \theta_j + \theta_j - \theta_q + \theta_q - \theta_i) \bmod 2\pi = 0,$$

as $(a + b \bmod m) = \{(a \bmod m) + (b \bmod m) \bmod m\}$. Hence we obtain the 3-cycle condition

$$(\mathbf{A}_{i,j} + \mathbf{A}_{j,q} + \mathbf{A}_{q,i}) \bmod 2\pi = 0, \forall (i, j, q) \text{ such that } \mathbf{A}_{i,j} \cdot \mathbf{A}_{j,q} \cdot \mathbf{A}_{q,i} > 0. \tag{3}$$

With $\mathbb{T} = \{(i, j, q) : \mathbf{A}_{i,j} \cdot \mathbf{A}_{j,q} \cdot \mathbf{A}_{q,i} > 0\}$, we define the *cycle inconsistency level* $\frac{1}{|\mathbb{T}|} \sum_{(i,j,q) \in \mathbb{T}} [(\mathbf{A}_{i,j} + \mathbf{A}_{j,q} + \mathbf{A}_{q,i}) \bmod 2\pi]$. We devise a loss function to minimize the cycle inconsistency level with reweighted edges.

### 4.3 LOSS AND EVALUATION FOR GENERAL K-SYNCHRONIZATION

The upset loss for general $k$ is defined similarly as in Sec. 4.1. Recall that the observed graph $\mathcal{G}$ has adjacency matrix $\mathbf{A}$. Given $k$ groups of estimated angles $\{r_{1,l}, \ldots, r_{n,l}\}, l = 1, \ldots, k$, we define the matrix $\mathbf{T}^{(l)}$ with entries $\mathbf{T}_{i,j}^{(l)} = (r_{i,l} - r_{j,l}) \bmod 2\pi$, for $i, j \in \{1, \ldots, n\}, l \in \{1, \ldots, k\}$. We define $\mathbf{M}^{(l)}$ by $\mathbf{M}_{i,j}^{(l)} = \min((\mathbf{T}_{i,j}^{(l)} - \mathbf{A}_{i,j}) \bmod 2\pi, (\mathbf{A}_{i,j} - \mathbf{T}_{i,j}^{(l)}) \bmod 2\pi)$ if $\mathbf{A}_{i,j} \neq 0$, and $\mathbf{M}_{i,j}^{(l)} = 0$ if $\mathbf{A}_{i,j} = 0$. Define $\mathbf{M}$ by $\mathbf{M}_{i,j} = \min_{l \in \{1, \ldots, k\}} \mathbf{M}_{i,j}^{(l)}$. The upset loss is as in eq. (1), $\mathcal{L}_{\text{upset}} = \|\mathbf{M}\|_F / t$.

In addition to $\mathcal{L}_{\text{upset}}$, we introduce another option as a loss function based on the cycle consistency relation from Sec. 4.2, which adds a regularization that helps in guiding the learning process for certain challenging scenarios (e.g., with sparser $\mathcal{G}$ or larger $k$). Since measurements are typically noisy, we first estimate the corruption level by entries in $\mathbf{M}$, and use them to construct a confidence matrix $\tilde{\mathbf{C}}$ for edges in $\mathcal{G}$. We define the unnormalized confidence matrix $\mathbf{C}$ by $\mathbf{C}_{i,j} = \frac{1}{1+\mathbf{M}_{i,j}}\mathbb{1}(\mathbf{A}_{i,j} \neq 0)$, then normalize the entries by $\tilde{\mathbf{C}}_{i,j} = \mathbf{C}_{i,j}\frac{\sum_{u,v}\mathbf{A}_{u,v}}{\sum_{u,v}\mathbf{A}_{u,v}\cdot\mathbf{C}_{u,v}}$. The normalization is chosen such that $\sum_{i,j}\mathbf{A}_{i,j}\tilde{\mathbf{C}}_{i,j} = \sum_{u,v}\mathbf{A}_{u,v}$. Keeping the sum of edge weights constant is carried out in order to avoid reducing the cycle inconsistency level by only rescaling edge weights but not their relative magnitudes. Based on the confidence matrix $\tilde{\mathbf{C}}$, we reweigh edges in $\mathcal{G}$ to obtain an updated input graph, whose adjacency matrix is the Hadamard product $\mathbf{A} \odot \tilde{\mathbf{C}}$. This graph attaches larger weights to edges $\mathbf{A}_{i,j}$ for which $\mathbf{T}_{i,j}^{(l)}$ is a good estimate when the edge $(i,j)$ belongs to graph $\mathcal{G}_l$. As the graph assignment of an edge $(i,j)$ is not known during training, we estimate it by

$$g(i,j) = \arg\min_{l\in\{1,\dots,k\}}\mathbf{M}_{i,j}^{(l)}, \text{ and set } g(j,i) = g(i,j), \tag{4}$$

thus obtaining our estimated graphs $\tilde{\mathcal{G}}_1,\dots,\tilde{\mathcal{G}}_k$, which are also edge disjoint. Next, aiming to minimize 3-cycle inconsistency of the updated input graph given our graph assignment estimates, we introduce a loss function denoted as the *cycle inconsistency loss* $\mathcal{L}_{\text{cycle}}$; for simplicity, we only focus on 3-cycles (triangles). We interpret the matrix $\tilde{\mathbf{A}} = (\mathbf{A} \odot \tilde{\mathbf{C}} - (\mathbf{A} \odot \tilde{\mathbf{C}})^{\top}) \bmod 2\pi$ as the adjacency matrix of another weighted directed graph $\tilde{\mathcal{G}}$. The entry $\tilde{A}_{i,j}$ of the new adjacency matrix approximates angular differences of a reweighted graph, with noisy observations downweighted. Note that we only reweigh the adjacency matrix in the cycle loss definition, but do not update the input graph. The underlying idea is that this updated denoised graph may display higher cycle consistency than the original graph. From our graph assignment estimates, we obtain estimated adjacency matrices $\tilde{\mathbf{A}}^{(l)}$ for $l \in \{1,\dots,k\}$, where $\tilde{\mathbf{A}}_{i,j}^{(l)} = \mathbb{1}(g(i,j) = l)\tilde{\mathbf{A}}_{i,j}$. Let $\mathbb{T}^{(l)} = \{(i,j,q) : \tilde{A}_{i,j}^{(l)}\cdot\tilde{A}_{j,q}^{(l)}\cdot\tilde{A}_{q,i}^{(l)} > 0\}$ denote the set of all triangles in $\tilde{\mathcal{G}}_l$, and set $S_{i,j,q}^{(l)} = \tilde{\mathbf{A}}_{i,j}^{(l)} + \tilde{\mathbf{A}}_{j,q}^{(l)} + \tilde{\mathbf{A}}_{q,i}^{(l)}$ for $(i,j,q) \in \mathbb{T}^{(l)}$. We define

$$\mathcal{L}_{\text{cycle}}^{(l)} = \frac{1}{|\mathbb{T}^{(l)}|}\sum_{(i,j,q)\in\mathbb{T}^{(l)}}\min(S_{i,j,q}^{(l)} \bmod 2\pi, (-S_{i,j,q}^{(l)}) \bmod 2\pi) \tag{5}$$

and set $\mathcal{L}_{\text{cycle}} = \frac{1}{k}\sum_{l=1}^{k}\mathcal{L}_{\text{cycle}}^{(l)}$. The default training loss for $k \geqslant 2$ is $\mathcal{L}_{\text{cycle}}$ or $\mathcal{L}_{\text{upset}}$ alone; in the experiment section, we also report the performance of a variant based on $\mathcal{L}_{\text{upset}} + \mathcal{L}_{\text{cycle}}$.

For evaluation, we compute $\mathcal{D}_{\text{MSE}}$ with eq. (2), for each of the $k$ sets of angles, and consider the average. As the ordering of the $k$ sets can be arbitrary, we consider all permutations of $\{1,\dots,k\}$, denoted by $perm(k)$. Denoting the ground-truth angle matrix as $\mathbf{R}$, whose $(i,l)$ entry is the ground-truth angle $\theta_{i,l}$, and the $l$-th entry of the permutation $pe$ by $pe(l)$, the final MSE value is

$$\mathcal{D}_{\text{MSE}}(\mathbf{r},\mathbf{R}) = \frac{1}{k}\min_{pe\in perm(k)}\sum_{l=1}^{k}\mathcal{D}_{\text{MSE}}(\mathbf{r}_{:,pe(l)},\mathbf{R}_{:,l}). \tag{6}$$

Note that the MSE loss is not used during training as we do not have any ground-truth supervision; the MSE formulation in eq. (2) is only used for evaluation. The lack of ground-truth information in the presence of noise is precisely what renders this problem very difficult. If any partial ground-truth information is available, then this can be incorporated into the loss function.

## 5  GNNSYNC ARCHITECTURE

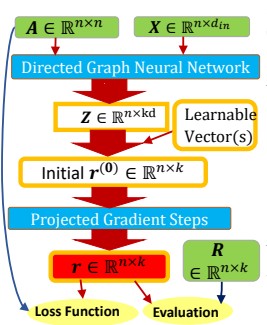

Figure 2: GNNSync overview: starting from an adjacency matrix $\mathbf{A}$ encoding (noisy) pairwise offsets and an input feature matrix $\mathbf{X}$, GNNSync first applies a directed GNN to learn node embeddings $\mathbf{Z}$. It then calculates the inner product with a learnable vector (or $k$ learnable vectors for $k > 1$) to produce the initial estimated angles $r_{i,l}^{(0)} \in [0,2\pi)$ for $l \in \{1,\dots,k\}$, after rescaling. It then applies several projected gradient steps to the initial angle estimates to obtain the final angle estimates, $r_{i,l} \in [0,2\pi)$. Let the ground-truth angle matrix be $\mathbf{R} \in \mathbb{R}^{n\times k}$. The loss function is applied to the output angle matrix $\mathbf{r}$, given $\mathbf{A}$, while the final evaluation is based on $\mathbf{R}$ and $\mathbf{r}$. Orange frames indicate trainable vectors/matrices, green squares fixed inputs, the red square the final estimated angles (outputs), and the yellow circles the loss function and evaluation.

## 5.1 OBTAINING DIRECTED GRAPH EMBEDDINGS

For obtaining digraph embeddings, any digraph GNN that outputs node embeddings can be applied, e.g. DIMPA by He et al. (2022b), the inception block model by Tong et al. (2020), and MagNet by Zhang et al. (2021). Here we employ DIMPA; details are in A.2. Denoting the final node embedding matrix by $\mathbf{Z} \in \mathbb{R}^{n \times kd}$, the embedding vector $\mathbf{z}_i$ for a node $i$ is $\mathbf{z}_i = (\mathbf{Z})_{(i,:)} \in \mathbb{R}^{kd}$, the $i^{\text{th}}$ row of $\mathbf{Z}$.

## 5.2 OBTAINING INITIAL ESTIMATED ANGLES

To obtain the initial estimated angles for the angular synchronization problem, we introduce a trainable vector $\mathbf{a}$ with dimension equal to the embedding dimension, then calculate the unnormalized estimated angles by the inner product of $\mathbf{z}_i$ with $\mathbf{a}$, plus a trainable bias $b$, followed by a sigmoid layer to force positive values, and finally rescale the angles to $[0, 2\pi)$; in short: $r_i^{(0)} = 2\pi \, \text{sigmoid}(\mathbf{z}_i \cdot \mathbf{a} + b)$.

For general $k$-synchronization, we apply independent $\mathbf{a}, b$ values to obtain $k$ different groups of initial angle estimates based on different columns of the node embedding matrix $\mathbf{Z}$. In general, denote $\mathbf{Z}_{i,u:v}$ as the $(v - u + 1)$-vector whose entries are from the $i$-th row and the $u$-th to $v$-th columns of the matrix $\mathbf{Z}$. With a trainable vector $\mathbf{a}^{(l)}$ for each $l \in \{1, \dots, k\}$ with dimension equal to $d$, we obtain the unnormalized estimated angles by the inner product of $\mathbf{Z}_{i,(l-1)d+1:ld}$ with $\mathbf{a}^{(l)}$, plus a trainable bias $b_l$, followed by a sigmoid layer to force positive angle values, then rescale the angles to $[0, 2\pi)$; in short: $r_{i,l}^{(0)} = 2\pi \, \text{sigmoid}(\mathbf{z}_{i,(l-1)d+1:ld} \cdot \mathbf{a}^{(l)} + b_l)$.

## 5.3 PROJECTED GRADIENT STEPS FOR FINAL ANGLE ESTIMATES

---
**Algorithm 1** Projected Gradient Steps

---
**Input**: Initial angle estimates $\mathbf{r}^{(0)} \in \mathbb{R}^{n \times k}$, Hermitian matrix $\mathbf{H} \in \mathbb{R}^{n \times n}$, number of steps $\Gamma$ (default: 5).
**Parameter**: (Initial) parameter set $\{\alpha_\gamma \geq 0\}_{\gamma=1}^{\Gamma}$ that could either be fixed or trainable (default: fixed value 1).
**Output**: Updated angle estimates $\mathbf{r} \in \mathbb{R}^{n \times k}$.

1: $l \leftarrow 1$;
2: **for** $l \leq k$ **do**
3: $\quad \gamma \leftarrow 1; \mathbf{y} \leftarrow \mathbf{r}_{:,l}^{(0)}$;
4: $\quad$ **for** $\gamma \leq \Gamma$ **do**
5: $\quad\quad \tilde{\mathbf{y}} \leftarrow \exp(\iota \mathbf{y})$;
6: $\quad\quad \tilde{\mathbf{y}} \leftarrow \alpha_\gamma \tilde{\mathbf{y}} + \mathbf{H}\tilde{\mathbf{y}}$;
7: $\quad\quad \mathbf{y} \leftarrow \text{angle}(\tilde{\mathbf{y}})$ to obtain elementwise angles in radians from complex numbers;
8: $\quad\quad \gamma \leftarrow \gamma + 1$.
9: $\quad$ **end for**
10: $\quad \mathbf{r}_{:,l} \leftarrow \mathbf{y}$.
11: $\quad l \leftarrow l + 1$.
12: **end for**
13: **return** $\mathbf{r}$.

---

Our final angle estimates are obtained after applying several (default: $\Gamma = 5$) projected gradient steps to the initial angle estimates. In brief, projected gradient descent for constrained optimization problems first takes a gradient step while ignoring the constraints, and then projects the result back onto the feasible set to incorporate the constraints. Here the projected gradient steps are inspired by Boumal (2016). We construct $\mathbf{H}$ by $\mathbf{H}_{i,j} = \exp(\iota \mathbf{A}_{i,j})\mathbb{1}(\mathbf{A}_{i,j} \neq 0)$, and update the estimated angles using Algo. 1, where $\mathbf{r}_{:,l}$ denotes the $l$-th column of $\mathbf{r}$. In Algo. 1 the gradient step is on line 6, while the projection step on line 7 projects the updated matrix to elementwise angles. Fig. 2 shows the GNNSync framework.

If graph assignments can be estimated effectively right after the GNN, one can replace $\mathbf{H}$ with $\mathbf{H}^{(l)}$ for each $l = 1, \dots, k$ separately, where $\mathbf{H}_{i,j}^{(l)} = \exp(\iota \mathbf{A}_{i,j}^{(l)})\mathbb{1}(\mathbf{A}_{i,j}^{(l)} \neq 0)$, and $\mathbf{A}_{i,j}^{(l)} = \mathbb{1}(g(i,j) = l)\mathbf{A}_{i,j}$ is the estimated adjacency matrix for graph $\mathcal{G}_l$ using network assignments from $g(i,j)$ from eq. (4) applied to the initial angle estimates $\mathbf{r}^{(0)}$. Yet, separate $\mathbf{H}^{(l)}$'s may make the architecture sensitive to the accuracy of graph assignments after the GNN, and hence for robustness we simply choose a single $\mathbf{H}$. We also find in Sec. 6.4 that the use of $\mathbf{H}$ instead of separate $\mathbf{H}^{(l)}$'s is essential for satisfactory performance. Besides, it is possible to make Algo. 1 parameter-free by further fixing the $\{\alpha_\gamma\}$ values (default: $\alpha_\gamma = 1, \forall \gamma$); we find that using fixed $\{\alpha_\gamma\}$ does not strongly affect performance in our experiments. GNNSync executes the projected gradient descent steps at every training iteration as part of a unified end-to-end training process, but one could also use Algo. 1 to post-process predicted angles without putting the steps in the end-to-end framework. We find that putting Algo. 1 in our end-to-end training framework is usually helpful.

## 5.4 ROBUSTNESS OF GNNSYNC

Measurement noise that perturbs the edge offsets can significantly impact the performance of group synchronization algorithms. To this end, we demonstrate the robustness of GNNSync to such noise perturbations, with the following theoretical guarantee, proved and further discussed in App. A.2

**Proposition 2.** *For adjacency matrices* $\mathbf{A}, \hat{\mathbf{A}}$, *assume their row-normalized variants* $\mathbf{A}_s, \hat{\mathbf{A}}_s, \mathbf{A}_t, \hat{\mathbf{A}}_t$ *satisfy* $\left\|\mathbf{A}_s - \hat{\mathbf{A}}_s\right\|_F < \epsilon_s$ *and* $\left\|\mathbf{A}_t - \hat{\mathbf{A}}_t\right\|_F < \epsilon_t$, *where subscripts* $s, t$ *denote source and target,*

*resp. Assume further their input feature matrices* $\mathbf{X}, \hat{\mathbf{X}}$ *satisfy* $\left\| \mathbf{X} - \hat{\mathbf{X}} \right\|_F < \epsilon_f$. *Then their initial angles* $\mathbf{r}^{(0)}, \hat{\mathbf{r}}^{(0)}$ *from a trained GNNSync using DIMPA satisfy* $\left\| \mathbf{r}^{(0)} - \hat{\mathbf{r}}^{(0)} \right\|_F < B_s \epsilon_s + B_t \epsilon_s + B_f \epsilon_f$, *for values* $B_s, B_t, B_f$ *that can be bounded by imposing constraints on model parameters and input.*

## 6 EXPERIMENTS

Implementation details are in App. C and extended results in App. D.

### 6.1 DATA SETS AND PROTOCOL

Previous works in angular synchronization typically only consider synthetic data sets in their experiments, and those applying synchronization to real-world data do not typically publish the data sets. To bridge the gap between synthetic experiments and the real world, we construct synthetic data sets with both correlated and uncorrelated ground-truth rotation angles, using various measurement graphs and noise levels. In addition, we conduct sensor network localization on two data sets.

For synthetic data, we perform experiments on graphs with $n = 360$ nodes for different measurement graphs, with edge density parameter $p \in \{0.05, 0.1, 0.15\}$, noise level $\eta \in \{0, 0.1, \ldots, 0.9\}$ for $k = 1$, and $\eta \in \{0, 0.1, \ldots, 0.7\}$ for $k \in \{2, 3, 4\}$. The graph generation procedure is as follows (with further details in App. B.1): •1) Generate $k$ group(s) of ground-truth angles. One option is to generate each angle from the same Gamma distribution with shape 0.5 and scale $2\pi$. We denote this option with subscript "1". As angles could be highly correlated in practical scenarios, we introduce a more realistic but challenging option "2", with multivariate normal ground-truth angles. The mean of the ground-truth angles is $\pi$, with covariance matrix for each $l \in \{1, \ldots, k\}$ defined by $\mathbf{w}\mathbf{w}^\top$, where entries in $\mathbf{w}$ are generated independently from a standard normal distribution. We explore two more options in the SI. We then apply mod $2\pi$ to all angles. • 2) Generate a noisy background adjacency matrix $\mathbf{A}_{\text{noise}} \in \mathbb{R}^{n \times n}$. • 3) Construct a complete adjacency matrix where $\eta$ portion of the entries are noisy and the rest represent true angular differences. • 4) Generate a measurement graph and sparsify the complete adjacency matrix by only keeping the edges in the measurement graph.

We construct 3 types of measurement graphs from NetworkX (Hagberg et al., 2008) and use the following notations, where the subscript $o \in \{1, 2, 3, 4\}$ is the option mentioned in step 1) above:
• Erdős-Rényi (ER) Outlier model: denoted by $\text{ERO}_o(p, k, \eta)$, using as the measurement graph the ER model from NetworkX, where $p$ is the edge density parameter for the ER measurement graph;
• Barabasi Albert (BA) Outlier model: denoted by $\text{BAO}_o(p, k, \eta)$, where the measurement graph is a BA model with the number of edges to attach from a new node to existing nodes equal to $\lceil np/2 \rceil$, using the standard implementation from NetworkX Hagberg et al. (2008); and
• Random Geometric Graph (RGG) Outlier model: denoted by $\text{RGGO}_o(p, k, \eta)$, with NetworkX parameter "distance threshold value (radius)" $2p$ for the RGG measurement graph. For $k = 1$, we omit the value $k$ and subscript $o$ in the notation, as the two options coincide in this special case.

For real-world data, we conduct sensor network localization on the U.S. map and the PACM point cloud data set (Cucuringu et al., 2012a) with a focus on the SO(2) component, as follows, with data processing details provided in App. B.2. •1) Starting with the ground-truth locations of $n = 1097$ U.S. cities (resp., $n = 426$ points), we construct patches using each city (resp., point) as a central node and add its 50 nearest neighbors to the corresponding patch. •2) For each patch, we add noise to each node's coordinates independently. •3) We then rotate the patches using random rotation angles (ground-truth angles generated as in 1) for synthetic models). For each pair of patches that have at least 6 overlapping nodes, we apply Procrustes alignment (Gower, 1975) to estimate the rotation angle based on these overlapping nodes and add an edge to the observed measurement adjacency matrix. •4) We perform angular synchronization to obtain the initial estimated angles and update the estimated angles by shifting by the average pairwise differences between the estimated and ground-truth angles, to eliminate the degree of freedom of a global rotation. •5) Finally, we apply the estimated rotations to the noisy patches and estimate node coordinates by averaging the estimated locations for each node from all patches that contain this node.

### 6.2 BASELINES

In our numerical experiments for angular synchronization, we compare against **7 baselines**, where results are averaged over 10 runs: • Spectral Baseline (Spectral) by Singer (2011), • Row-Normalized Spectral Baseline (Spectral_RN) by Cucuringu et al. (2012a), • Generalized Power Method (GPM) by Boumal (2016), • TranSync by Huang et al. (2017), • CEMP_GCW, • CEMP_MST by Lerman & Shi (2022), and • Trimmed Averaging Synchronization (TAS) by Maunu & Lerman (2023).

For more general $k$-synchronization, we compare against two baselines from Cucuringu & Tyagi (2022), which are based on the top $k$ eigenvectors of the matrix $\mathbf{H}$ or its row-normalized version. We use names • Spectral and • Spectral_RN to denote them as before. To show that GNNSync (as well as the baselines) deviate from trivial or random solutions, we include an additional baseline denoted "Trivial" for each $k$, where all angles are predicted equal (with value 1, for simplicity).

## 6.3 MAIN EXPERIMENTAL RESULTS

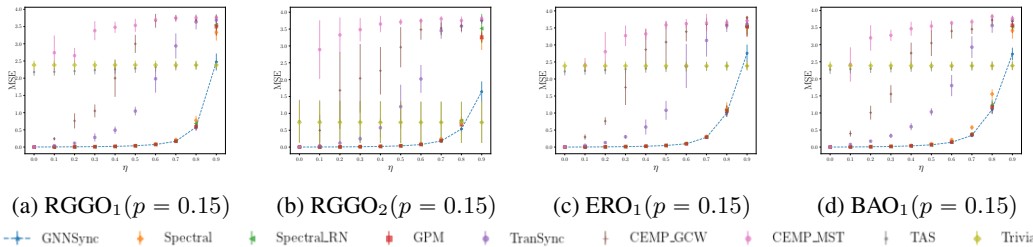

(a) $\text{RGGO}_1(p = 0.15)$    (b) $\text{RGGO}_2(p = 0.15)$    (c) $\text{ERO}_1(p = 0.15)$    (d) $\text{BAO}_1(p = 0.15)$

Figure 3: MSE performance on angular synchronization ($k = 1$). Error bars indicate one standard deviation. Dashed lines highlight GNNSync variants.

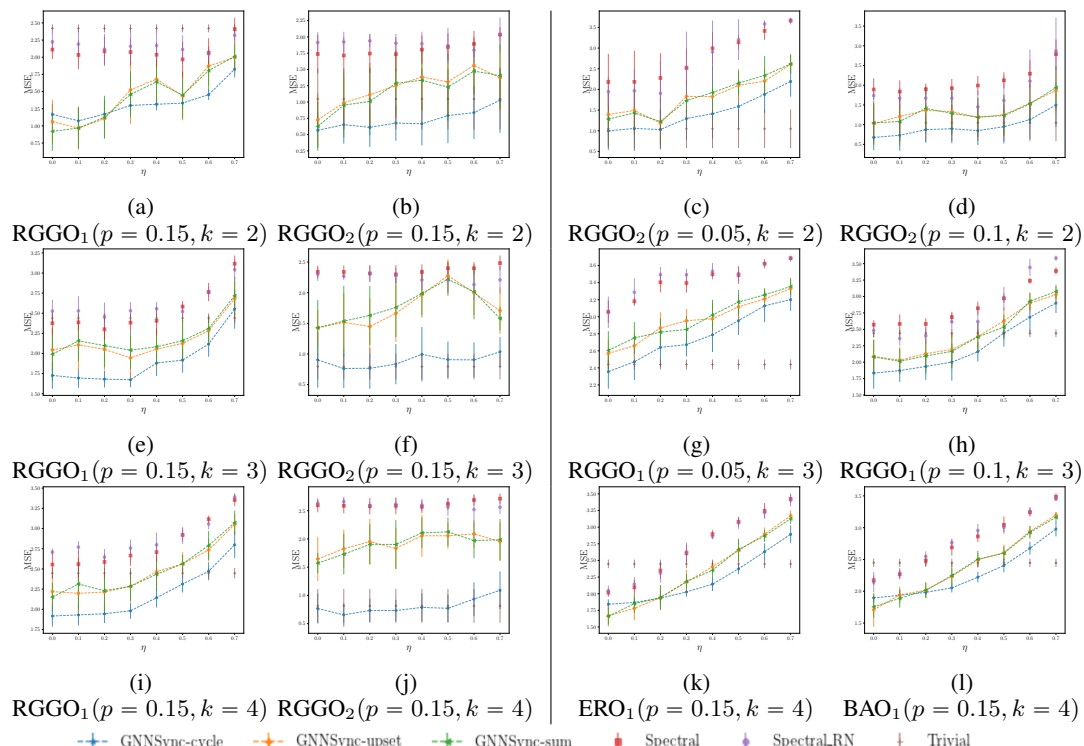

(a)    (b)    (c)    (d)
$\text{RGGO}_1(p = 0.15, k = 2)$   $\text{RGGO}_2(p = 0.15, k = 2)$   $\text{RGGO}_2(p = 0.05, k = 2)$   $\text{RGGO}_2(p = 0.1, k = 2)$

(e)    (f)    (g)    (h)
$\text{RGGO}_1(p = 0.15, k = 3)$   $\text{RGGO}_2(p = 0.15, k = 3)$   $\text{RGGO}_1(p = 0.05, k = 3)$   $\text{RGGO}_1(p = 0.1, k = 3)$

(i)    (j)    (k)    (l)
$\text{RGGO}_1(p = 0.15, k = 4)$   $\text{RGGO}_2(p = 0.15, k = 4)$   $\text{ERO}_1(p = 0.15, k = 4)$   $\text{BAO}_1(p = 0.15, k = 4)$

Figure 4: MSE performance on $k$-synchronization for $k \in \{2, 3, 4\}$. $p$ is the network density and $\eta$ is the noise level. Error bars indicate one standard deviation. Dashed lines highlight GNNSync variants.

By default, we use the output angles of the baseline "Spectral_RN" as input features for GNNSync, and thus $d_{\text{in}} = k$. The main experimental results are shown in Fig. 3 for $k = 1$, and Fig. 4 for general $k \in \{2, 3, 4\}$, with additional results reported in App. D. For $k > 1$, we use "GNNSync-cycle", "GNNSync-upset" and "GNNSync-sum" to denote GNNSync variants when considering the training loss function $\mathcal{L}_{\text{cycle}}$, $\mathcal{L}_{\text{upset}}$, and $\mathcal{L}_{\text{upset}} + \mathcal{L}_{\text{cycle}}$, respectively.

From Fig. 3 (with additional figures in App. D Fig. 6–8), we conclude that GNNSync produces generally the best performance compared to baselines, in angular synchronization ($k = 1$). From Fig. 4 (see also App. D Fig. 9–17), we again conclude that GNNSync variants attain leading performance for $k > 1$. The first two columns of Fig. 4 compare the performance of the two options of ground-truth angles on RGGO models. In columns 3 and 4, we show the effect of varying density parameter $p$, and different synthetic models under various measurement graphs.

For $k > 1$, GNNSync-upset performs better than both baselines in most cases, with $\mathcal{L}_{\text{upset}}$ simple yet effective to train. GNNSync-cycle generally attains the best performance. As the problems become harder (with increasing $\eta$, decreasing $p$, increasing $k$, more complex measurement graph RGG), GNNSync-cycle outperforms both baselines and other GNNSync variants by a larger margin. The performance of GNNSync-sum lies between that of GNNSync-upset and GNNSync-cycle, but is closer to that of GNNSync-upset, see App. D for more discussions on linear combinations of the two losses. We conclude that while GNNSync-upset generally attains satisfactory performance, GNNSync-cycle is more robust to harder problems than other GNNSync variants and the baselines. Accounting for the performance of trivial guesses, we observe that GNNSync variants are more robust to noise, and attain satisfactory performance even when the competitive baselines are outperformed by trivial guesses. We highlight that there is a clear advantage of using cycle consistency in the pipeline, especially when the problem is harder, thus reflecting the angular nature of the problem. For 3-cycle consistency and the cycle loss $\mathcal{L}_{\text{cycle}}$, gradient descent in principle drives down the (non-negative) values $S$ of the sum of three predicted angular differences. To minimize the $S$ values, we encourage a reweighing process of the initial edge weights so that cycle consistency roughly holds. Unlike $\mathcal{L}_{\text{upset}}$ which explicitly encourages small $\mathbf{M}_{i,j}$ values for all edges, $\mathcal{L}_{\text{cycle}}$ only implicitly encourages small $\mathbf{M}_{i,j}$ values via the confidence matrix reweighing process for edges with relatively small noise. In an ideal case, we only have large $\mathbf{M}_{i,j}$ values on noisy edges. In this case, the reweighing process would downweight these noisy edges, which results in a smaller value of the cycle loss function. This is also the underlying reason why $\mathcal{L}_{\text{cycle}}$ is more robust to noise than $\mathcal{L}_{\text{upset}}$. For $k > 1$, we hence recommend using the more intricate $\mathcal{L}_{\text{cycle}}$ function as the training loss function, and we will focus on GNNSync-cycle in the ablation study.

From Fig. 1 (see also App. D Tab. 1–4, Fig. 18–27), we observe that GNNSync is able to align patches and recover coordinates effectively, and is more robust to noise than baselines. GNNSync attains competitive MSE values and Average Normalized Error (ANE) results, where ANE (defined explicitly in App. D.1) measures the discrepancy between the predicted locations and the actual locations.

## 6.4 ABLATION STUDY AND DISCUSSION

In this subsection, we justify several model choices for all $k$: • the use of the projected gradient steps; • an end-to-end framework instead of training first without the projected gradient steps and then applying Algo. 1 as a post-processing procedure; • fixed instead of trainable $\{\alpha_\gamma\}$ values. For $k > 1$, we also justify the use of the $\mathbf{H}$ matrix in Algo. 1 instead of separate $\mathbf{H}^{(l)}$'s based on estimated graph assignments of the edges. To validate the ability of GNNSync to borrow strength from baselines, we set the input feature matrix $\mathbf{X}$ as a set of angles that is estimated by one of the baselines (or $k$ sets of angles estimated by one of the baselines for $k > 1$) and report the performance.

Due to space considerations, results for the ablation study are reported in App. D. For $k = 1$, Fig. 22–24 report the MSE performance for different GNNSync variants. Improvements over all possible baselines when taking their output as input features for $k = 1$ are reported in Fig. 34–36. For $k > 1$, we report the results when using $\mathcal{L}_{\text{cycle}}$ as the training loss function in Fig. 25–33. We conclude that Algo. 1 is indeed helpful in guiding GNNSync to attain lower loss values (we omit loss results for space considerations) and better MSE performance, and that end-to-end training usually attains comparable or better performance than using Algo. 1 for post-processing, even when there is no trainable parameter in Algo. 1. Moreover, the baselines are still outperformed by GNNSync if we apply the same number of projected gradient steps as in GNNSyc as fine-tuning post-processing to the baselines, as illustrated in Fig. 34 and 35. We observe across all data sets, that GNNSync usually improves on existing baselines when employing their outputs as input features, and never performs significantly worse than the corresponding baseline; hence, GNNSync can be used to enhance existing methods. Further, setting $\{\alpha_\gamma\}$ values to be trainable does not seem to boost performance much, and hence we stick to fixed $\{\alpha_\gamma\}$ values. For $k > 1$, using separate $\mathbf{H}^{(l)}$'s instead of the whole $\mathbf{H}$ in Algo. 1 harms performance, which can be explained by the fact that learning graph assignments effectively via GNN outputs is challenging.

## 7 CONCLUSION AND OUTLOOK

This paper proposed a general NN framework for angular synchronization and a heterogeneous extension. As the current framework is limited to SO(2), we believe that extending our GNN-based framework to the setting of other more general groups is an exciting research direction to pursue, and constitutes ongoing work (for instance, for doing synchronization over the full Euclidean group $\text{Euc}(2) = \mathbb{Z}_2 \times \text{SO}(2) \times \mathbb{R}^2$). We also plan to optimize the loss functions under constraints, train our framework with supervision of ground-truth angles (anchor information), and explore the interplay with low-rank matrix completion. Another interesting direction is to extend our SNL example to explore the graph realization problem, of recovering point clouds from a sparse noisy set of pairwise Euclidean distances.

## ACKNOWLEDGEMENTS

Y.H. is supported by a Clarendon scholarship. G.R. is supported in part by EPSRC grants EP/T018445/1, EP/W037211/1 and EP/R018472/1. M.C. acknowledges support from the EPSRC grants EP/N510129/1 and EP/W037211/1 at The Alan Turing Institute.

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

## A ANALYTICAL DISCUSSIONS

### A.1 PROPERTIES OF THE LOSS FUNCTIONS

In classical convex optimization of the type $\inf_{\mathbf{x}} g(x)$, optimal values are achieved at stationary points; when the function $g$ is differentiable, then following Fermat's rule, stationary points are points at which the gradient of $g$ vanishes. Such points are typically found via gradient descent methods. When the function $g$ is not differentiable, then there are weaker variants for differentiablity available such as the directional derivative. First we recall the notion of a directional derivative and a directional stationary point from Li et al. (2020). The *directional derivative* of a function $f$ at point $\mathbf{x} \in \mathbb{R}^m$ in the direction $\mathbf{d} \in \mathbb{R}^m$ is defined by

$$f'(\mathbf{x}, \mathbf{d}) = \lim_{t \searrow 0} \frac{f(\mathbf{x} + t\mathbf{d}) - f(\mathbf{x})}{t}.$$

A *directional stationary* point $\mathbf{x} \in \mathbb{R}^{\mathbf{n}}$ of the problem $\inf_{\mathbf{x} \in C} g(x)$ for $C \subset \mathbb{R}^n$ and $g : \mathbb{R}^n \to \mathbb{R}$ is a point such that the directional derivatives in any direction $\mathbf{d} \in \mathbb{R}^n$ satisfy $(g + \mathbb{1}_C)'(\mathbf{x}, \mathbf{d}) \geqslant 0$. This notion is broad enough to include functions such as the maximum which is not everywhere differentiable.

Moreover we say that a function $f : \mathbb{R}^m \to \mathbb{R}$ is *locally Lipschitz* if for any bounded set $\mathcal{S} \subset \mathbb{R}^m$, there exists a constant $L > 0$ such that

$$|f(\mathbf{x} - \mathbf{y})| \leqslant L \parallel \mathbf{x} - \mathbf{y} \parallel_2$$

for all $\mathbf{x}, \mathbf{y} \in \mathcal{S}$. Note that a locally Lipschitz function $f$ is differentiable almost everywhere, see for example (Rockafellar & Wets, 2009, Theorem 9.60) where also more background on directional derivatives and subdifferentials can be found.

**Proof of Proposition 1** Here we prove Proposition 1 from the main text; for convenience, we repeat it here.

**Proposition 3.** *Every local minimum of eq. (1) is a directional stationary point of eq. (2).*

*Proof.* eq. (2) gives that

$$
\begin{aligned}
\mathcal{L}_{\text{upset}} &= \parallel \mathbf{M} \parallel_F / t \\
&= \frac{1}{t} \sqrt{\sum_{i,j} \mathbb{1}(\mathbf{T}_{i,j} - \mathbf{A}_{i,j} \neq 0, 2\pi) \min(\mathbf{T}_{i,j} - \mathbf{A}_{i,j} \bmod 2\pi, \mathbf{A}_{i,j} - \mathbf{T}_{i,j} \bmod 2\pi)^2} \\
&= \frac{1}{t} \sqrt{\sum_{i,j} \mathbb{1}(\mathbf{T}_{i,j} - \mathbf{A}_{i,j} \neq 0, 2\pi) \min\{\mathbf{T}_{i,j} - \mathbf{A}_{i,j} \bmod 2\pi, 2\pi - (\mathbf{T}_{i,j} - \mathbf{A}_{i,j} \bmod 2\pi)\}^2}
\end{aligned}
$$

where $t$ is the number of nonzero elements in $\mathbf{A}$. The function $f : (0, 2\pi) \mapsto (0, 2\pi)$ given by $f(x) = \min(x^2, (2\pi - x)^2)$ is differentiable with derivative uniformly bounded by $4\pi$ (and is hence locally Lipschitz) except at the point $x = \pi$ where it takes on its maximum, $\pi^2$. Thus,

this function is a directionally differentiable Lipschitz function (which can be seen by writing $\min(a, b) = \frac{1}{2}(a + b) - \frac{1}{2}|a - b|$ and noting that $f(x) = -|x|$ is a directionally differentiable Lipschitz function). Moreover we can phrase the optimization problem for the upset loss as an optimization problem over the closed set $C = [0, 2\pi]^n$ representing sets $\{r_i, i = 1, \ldots, n\}$ which are then used to obtain matrices $T$ with entries $T_{ij} = r_i - r_j \mod 2\pi$. Fact 6 in Li et al. (2020) then guarantees that every local minimum is a directional stationary point of eq. (2). $\qquad\square$

**Discussion of the case of general $k$** For general $k$-synchronization, we only require $\mathbf{M}_{i,j}^{(l)}$ to be close to zero for one $l$ instead of all because each edge is assumed to belong to exactly one graph $\mathbf{G}_l$. Therefore in an ideal setting, for each edge $(i, j) \in \mathcal{E}$, exactly one of the entries $\mathbf{M}_{i,j}^{(l)}, l = 1, \ldots, k$ is zero. If all entries are large then this indicates that the information for $(i, j)$ is very noisy. Subsequently, the entry for $(i, j)$ is downweighted in the updated graph for the cycle loss function. The rationale is that when edge information is very noisy, the cycle consistency will often be violated; violations for edge information that is not so noisy are more important for angular synchronization as they should contain a stronger signal.

In terms of the cycle loss function itself, the confidence matrix $\tilde{\mathbf{C}}$ for edges in $\mathcal{G}$ arises. First consider the optimization problem in which $\tilde{\mathbf{C}}$ is omitted; taking $\tilde{\mathbf{A}} = (\mathbf{A} - \mathbf{A}^\top) \mod 2\pi$ and we optimize the upset loss, the cycle loss, or both. For the upset loss, eq. (5) itself when considering fixed $\tilde{\mathbf{A}}^{(l)}$ terms, can be analyzed similarly to the analysis of $\mathcal{L}_{\text{upset}}$ in Proposition 1, using that the minimum as appearing in $\mathbf{M}_{i,j} = \min_{l \in \{1,\ldots,k\}} \mathbf{M}_{i,j}^{(l)}$ is a directionally differentiable Lipschitz function. For the cycle loss function, the expression of $\mathcal{L}_{\text{cycle}}^{(l)}$ takes a constant (when we regard $\tilde{\mathbf{A}}^{(l)}$ as fixed) away from the minimum of $S_{i,j,q}^{(l)} \mod 2\pi$ and $(-S_{i,j,q}^{(l)}) \mod 2\pi$. This minimum is equivalent to $|\pi - (S_{i,j,q}^{(l)} \mod 2\pi)|$, which is again a directionally differentiable Lipschitz function. Arguing as for Proposition 1 thus shows that the statement of this proposition extends to this special treatment of the $k$-synchronization problem.

In our general treatment of the $k$-synchronization problem, the confidence matrix $\tilde{\mathbf{C}}$ depends on the maximum $\mathbf{M}$ of the residual matrices and involves the expression $\frac{1}{1+\mathbf{M}_{i,j}} \mathbb{1}(\mathbf{A}_{i,j} \neq 0)$. While the function $f(x) = \frac{1}{1+x}$ is differentiable for $x > 0$, its composition with a function such as $\mathbf{M}$ may not be differentiable, as $\mathbf{M}$ only possesses a very weak notion of differential, called a *limiting subdifferential* in Li et al. (2020), to which the chain rule does not apply. This complex dependence hinders a more rigorous analysis of the general treatment of the $k$-synchronization problem, where even the chain rule is not guaranteed to hold.

**Non-differentiable points of the loss function** Although the Frobenius norm, the min function, and modulo have non-differentiable points, these points have measure zero. Moreover, as we use PyTorch autograd [1] for gradient calculation, even in the presence of non-differentiable points, backpropagation can be carried out whenever an approximate gradient can be constructed. Note that the absolute value function is convex, and hence autograd will apply the sub-gradient of the minimum norm. There also exist differentiable approximations for the modulo, and hence backpropagation can still be executed. Finally, in our experiments, we do not empirically observe any issue of convergence.

**Novelty** While the design of the upset loss in isolation may be relatively straightforward for the $k = 1$ case, we provide theoretical support as well as a less obvious loss function extension to handle broader $k \geq 2$ cases that rely on assigning edges to different graphs. The design of the cycle loss is not trivial and based on problem-specific insights.

## A.2 ROBUSTNESS OF GNNSYNC

First we review DIMPA (Directed Mixed Path Aggregation) from He et al. (2022b) for obtaining a network embedding. DIMPA captures local network information by taking a weighted average of information from neighbors within $h$ hops. Here we use $h = 2$ hops throughout the paper. Let $\mathbf{A} \in \mathbb{R}^{n \times n}$ be an adjacency and $\mathbf{A}_s$ its row-normalization. A weighted self-loop is added to each

---

[1] https://pytorch.org/docs/stable/notes/autograd.html

node; then we normalize by setting $\mathbf{A}_s = (\tilde{\mathbf{D}}^s)^{-1}\tilde{\mathbf{A}}^s$, where $\tilde{\mathbf{A}}^s = \mathbf{A} + \tau\mathbf{I}_n$, with $\tilde{\mathbf{D}}^s$ the diagonal matrix with entries $\tilde{\mathbf{D}}^s_{i,i} = \sum_j \tilde{\mathbf{A}}^s_{i,j}$, $\mathbf{I}_n$ the $n \times n$ identity matrix, and $\tau$ is a small value; as in He et al. (2022b) we take $\tau = 0.5$.

The $h$-hop **source** matrix is given by $(\mathbf{A}_s)^h$. The set of *up-to-$h$-hop* source neighborhood matrices is denoted as $\mathbb{A}^{s,h} = \{\mathbf{I}_n, \mathbf{A}_s, \ldots, (\mathbf{A}_s)^h\}$. Similarly, for aggregating information when each node is viewed as a **target** node of a link, we carry out the same procedure for the transpose $\mathbf{A}^\top$. The set of up-to-$h$-hop target neighborhood matrices is denotes as $\mathbb{A}^{t,h} = \{\mathbf{I}_n, \mathbf{A}_t, \ldots, (\mathbf{A}_t)^h\}$, where $\mathbf{A}_t$ is the row-normalized target adjacency matrix calculated from $\mathbf{A}^\top$.

Let the input feature matrix be denoted by $\mathbf{X} \in \mathbb{R}^{n \times d_{\text{in}}}$. The source embedding is given by

$$\mathbf{Z}_s = \left(\sum_{\mathbf{N} \in \mathbb{A}^{s,h}} \omega^s_{\mathbf{N}} \cdot \mathbf{N}\right) \cdot \mathbf{Q}^s \in \mathbb{R}^{n \times d}, \tag{7}$$

where for each $\mathbf{N}$, $\omega^s_{\mathbf{N}}$ is a learnable scalar, $d$ is the dimension of this embedding, and $\mathbf{Q}^s = \text{MLP}^{(s,L)}(\mathbf{X})$. Here, the hyperparameter $L$ controls the number of layers in the multilayer perceptron (MLP) with ReLU activation but without the bias terms; as in He et al. (2022b) we fix $L = 2$ throughout. Each layer of the MLP has the same number $d$ of hidden units. The target embedding $\mathbf{Z}_t$ is defined similarly, with $s$ replaced by $t$ ineq. (7). After these two decoupled aggregations, the embeddings are concatenated to obtain the final node embedding as a $n \times (2d)$ matrix $\mathbf{Z} = \text{CONCAT}(\mathbf{Z}_s, \mathbf{Z}_t)$.

**Proof of Proposition 2**  Here we prove Proposition 2 from the main text; for convenience, we repeat it here.

**Proposition 4.** *For adjacency matrices* $\mathbf{A}, \hat{\mathbf{A}}$, *assume their row-normalized variants* $\mathbf{A}_s, \hat{\mathbf{A}}_s, \mathbf{A}_t, \hat{\mathbf{A}}_t$ *satisfy* $\left\|\mathbf{A}_s - \hat{\mathbf{A}}_s\right\|_F < \epsilon_s$ *and* $\left\|\mathbf{A}_t - \hat{\mathbf{A}}_t\right\|_F < \epsilon_t$, *where subscripts* $s, t$ *denote source and target, resp. Assume further their input feature matrices* $\mathbf{X}, \hat{\mathbf{X}}$ *satisfy* $\left\|\mathbf{X} - \hat{\mathbf{X}}\right\|_F < \epsilon_f$. *Then their initial angles* $\mathbf{r}^{(0)}, \hat{\mathbf{r}}^{(0)}$ *from a trained GNNSync using DIMPA satisfy* $\left\|\mathbf{r}^{(0)} - \hat{\mathbf{r}}^{(0)}\right\|_F < B_s\epsilon_s + B_t\epsilon_s + B_f\epsilon_f$, *for values* $B_s, B_t, B_f$ *that can be bounded by imposing constraints on model parameters and input.*

*Proof.*  Let us assume the input feature matrices are $\mathbf{X}, \hat{\mathbf{X}} \in \mathbb{R}^{n \times d_{\text{in}}}$ for $\mathbf{A}$ and $\hat{\mathbf{A}}$, respectively.

The DIMPA procedures for the input row-normalized adjacency matrices $\mathbf{A}_s, \mathbf{A}_t$ with 2 hops and hidden dimension $d$ can be written as a concatenation of the source and target node embeddings $\mathbf{Z}_s$ and $\mathbf{Z}_t$, where

$$\begin{aligned}\mathbf{Z}_s &= (\mathbf{I}_n + a_{s1}\mathbf{A}_s + a_{s2}\mathbf{A}_s^2)\text{ReLU}(\mathbf{X}\mathbf{W}_{s0})\mathbf{W}_{s1}, \\ \mathbf{Z}_t &= (\mathbf{I}_n + a_{t1}\mathbf{A}_t + a_{t2}\mathbf{A}_t^2)\text{ReLU}(\mathbf{X}\mathbf{W}_{t0})\mathbf{W}_{t1}.\end{aligned} \tag{8}$$

Here $\mathbf{I}_n \in \mathbb{R}^{n \times n}$ is the identity matrix, $a_{s1}, a_{s2}, a_{t1}, a_{t2} \in \mathbb{R}$, $\mathbf{W}_{s0}, \mathbf{W}_{t0} \in \mathbb{R}^{d_{\text{in}} \times d}$ where $d$ is the hidden dimension, and $\mathbf{W}_{s1}, \mathbf{W}_{t1} \in \mathbb{R}^{h \times h}$. Similarly, we have for $\hat{\mathbf{A}}_s$ and $\hat{\mathbf{A}}_t$

$$\begin{aligned}\hat{\mathbf{Z}}_s &= (\mathbf{I}_n + a_{s1}\hat{\mathbf{A}}_s + a_{s2}\hat{\mathbf{A}}_s^2)\text{ReLU}(\hat{\mathbf{X}}\mathbf{W}_{s0})\mathbf{W}_{s1}, \\ \hat{\mathbf{Z}}_t &= (\mathbf{I}_n + a_{t1}\hat{\mathbf{A}}_t + a_{t2}\hat{\mathbf{A}}_t^2)\text{ReLU}(\hat{\mathbf{X}}\mathbf{W}_{t0})\mathbf{W}_{t1}.\end{aligned} \tag{9}$$

After DIMPA, we carry out the innerproduct procedure and sigmoid rescaling, to obtain for $k = 1$

$$\mathbf{r}^{(0)} = 2\pi\text{sigmoid}(\mathbf{Z}_s\mathbf{a}_s + \mathbf{Z}_t\mathbf{a}_t + b), \quad \hat{\mathbf{r}}^{(0)} = 2\pi\text{sigmoid}(\hat{\mathbf{Z}}_s\mathbf{a}_s + \hat{\mathbf{Z}}_t\mathbf{a}_t + b), \tag{10}$$

where $\mathbf{a}_s, \mathbf{a}_t \in \mathbb{R}^{d \times 1}$ and $b$ is a trained scalar.

For $k > 1$, we have (before reshaping $\mathbf{r}^{(0)}$ and $\hat{\mathbf{r}}^{(0)}$ from shape $nk \times 1$ to shape $n \times k$ which does not change the Frobenius norm)

$$\mathbf{r}^{(0)} = 2\pi\text{sigmoid}(\mathbf{Z}_s\mathbf{a}_s + \mathbf{Z}_t\mathbf{a}_t + \mathbf{b}), \quad \hat{\mathbf{r}}^{(0)} = 2\pi\text{sigmoid}(\hat{\mathbf{Z}}_s\mathbf{a}_s + \hat{\mathbf{Z}}_t\mathbf{a}_t + \mathbf{b}), \tag{11}$$

where $\mathbf{a}_s, \mathbf{a}_t \in \mathbb{R}^{dk}$ and $\mathbf{b} \in \mathbb{R}^k$. Indeed, we could view the scalar $b$ as a 1D vector, and consider eq. (10) as a special case of eq. (11).

Using eq. (8) and eq. (9), along with the triangle inequality, we have that

$$
\begin{aligned}
&\left\|\mathbf{Z}_s - \hat{\mathbf{Z}}_s\right\|_F \\
&= \left\|(\mathbf{I}_n + a_{s1}\mathbf{A}_s + a_{s2}\mathbf{A}_s^2)\text{ReLU}(\mathbf{X}\mathbf{W}_{s0})\mathbf{W}_{s1} - (\mathbf{I}_n + a_{s1}\hat{\mathbf{A}}_s + a_{s2}\hat{\mathbf{A}}_s^2)\text{ReLU}(\hat{\mathbf{X}}\mathbf{W}_{s0})\mathbf{W}_{s1}\right\|_F \\
&\leqslant \left\|(\mathbf{I}_n + a_{s1}\mathbf{A}_s + a_{s2}\mathbf{A}_s^2)\text{ReLU}(\mathbf{X}\mathbf{W}_{s0})\mathbf{W}_{s1} - (\mathbf{I}_n + a_{s1}\hat{\mathbf{A}}_s + a_{s2}\hat{\mathbf{A}}_s^2)\text{ReLU}(\mathbf{X}\mathbf{W}_{s0})\mathbf{W}_{s1}\right\|_F + \\
&\quad \left\|(\mathbf{I}_n + a_{s1}\hat{\mathbf{A}}_s + a_{s2}\hat{\mathbf{A}}_s^2)\text{ReLU}(\mathbf{X}\mathbf{W}_{s0})\mathbf{W}_{s1} - (\mathbf{I}_n + a_{s1}\hat{\mathbf{A}}_s + a_{s2}\hat{\mathbf{A}}_s^2)\text{ReLU}(\hat{\mathbf{X}}\mathbf{W}_{s0})\mathbf{W}_{s1}\right\|_F \\
&\leqslant \left\|[a_{s1}(\mathbf{A}_s - \hat{\mathbf{A}}_s) + a_{s2}(\mathbf{A}_s^2 - \hat{\mathbf{A}}_s^2)]\text{ReLU}(\mathbf{X}\mathbf{W}_{s0})\mathbf{W}_{s1}\right\|_F + \\
&\quad \left\|\mathbf{I}_n + a_{s1}\hat{\mathbf{A}}_s + a_{s2}\hat{\mathbf{A}}_s^2\right\|_F \|\mathbf{W}_{s1}\|_F \left\|\text{ReLU}(\mathbf{X}\mathbf{W}_{s0}) - \text{ReLU}(\hat{\mathbf{X}}\mathbf{W}_{s0})\right\|_F \\
&\leqslant \left\|\mathbf{A}_s - \hat{\mathbf{A}}_s\right\|_F \left\|[a_{s1} + a_{s2}(\mathbf{A}_s + \hat{\mathbf{A}}_s)]\text{ReLU}(\mathbf{X}\mathbf{W}_{s0})\mathbf{W}_{s1}\right\|_F + \\
&\quad \left\|\mathbf{I}_n + a_{s1}\hat{\mathbf{A}}_s + a_{s2}\hat{\mathbf{A}}_s^2\right\|_F \|\mathbf{W}_{s1}\|_F \|\mathbf{W}_{s0}\|_F \left\|\mathbf{X} - \hat{\mathbf{X}}\right\|_F \\
&< \epsilon_s B_{s0} + \epsilon_f B_{fs},
\end{aligned}
\tag{12}
$$

where $B_{s0} = \left\|[a_{s1} + a_{s2}(\mathbf{A}_s + \hat{\mathbf{A}}_s)]\text{ReLU}(\mathbf{X}\mathbf{W}_{s0})\mathbf{W}_{s1}\right\|_F$ and
$B_{fs} = \left\|\mathbf{I}_n + a_{s1}\hat{\mathbf{A}}_s + a_{s2}\hat{\mathbf{A}}_s^2\right\|_F \|\mathbf{W}_{s1}\|_F \|\mathbf{W}_{s0}\|_F$. Note that we also use the fact that the ReLU function is Lipschitz with Lipschitz constant 1.

Likewise, we have

$$
\left\|\mathbf{Z}_t - \hat{\mathbf{Z}}_t\right\|_F < \epsilon_t B_{t0} + \epsilon_f B_{ft},
\tag{13}
$$

where $B_{t0} = \left\|[a_{t1} + a_{t2}(\mathbf{A}_t + \hat{\mathbf{A}}_t)]\text{ReLU}(\mathbf{X}\mathbf{W}_{t0})\mathbf{W}_{t1}\right\|_F$ and
$B_{ft} = \left\|\mathbf{I}_n + a_{t1}\hat{\mathbf{A}}_t + a_{t2}\hat{\mathbf{A}}_t^2\right\|_F \|\mathbf{W}_{t1}\|_F \|\mathbf{W}_{t0}\|_F$.

With eq. (12) and eq. (13), noting that the sigmoid function is Lipschitz with Lipschitz constant 1, we employ eq. (11) to obtain

$$
\begin{aligned}
&\left\|\mathbf{r}^{(0)} - \hat{\mathbf{r}}^{(0)}\right\|_F \\
&= \left\|2\pi\text{sigmoid}(\mathbf{Z}_s\mathbf{a}_s + \mathbf{Z}_t\mathbf{a}_t + \mathbf{b}) - 2\pi\text{sigmoid}(\hat{\mathbf{Z}}_s\mathbf{a}_s + \hat{\mathbf{Z}}_t\mathbf{a}_t + \mathbf{b})\right\|_F \\
&\leqslant 2\pi \left\|(\mathbf{Z}_s\mathbf{a}_s + \mathbf{Z}_t\mathbf{a}_t + \mathbf{b}) - (\hat{\mathbf{Z}}_s\mathbf{a}_s + \hat{\mathbf{Z}}_t\mathbf{a}_t + \mathbf{b})\right\|_F \\
&= 2\pi \left\|(\mathbf{Z}_s - \hat{\mathbf{Z}}_s)\mathbf{a}_s + (\mathbf{Z}_t - \hat{\mathbf{Z}}_t)\mathbf{a}_t\right\|_F \\
&\leqslant 2\pi \left[\left\|(\mathbf{Z}_s - \hat{\mathbf{Z}}_s)\mathbf{a}_s\right\|_F + \left\|(\mathbf{Z}_t - \hat{\mathbf{Z}}_t)\mathbf{a}_t\right\|_F\right] \\
&= 2\pi \left(\|\mathbf{a}_s\|_F \left\|\mathbf{Z}_s - \hat{\mathbf{Z}}_s\right\|_F + \|\mathbf{a}_t\|_F \left\|\mathbf{Z}_t - \hat{\mathbf{Z}}_t\right\|_F\right) \\
&< 2\pi(\epsilon_s B_{s0} + \epsilon_f B_{fs} + \epsilon_t B_{t0} + \epsilon_f B_{ft}) \\
&= \epsilon_s B_s + \epsilon_t B_t + \epsilon_f B_f,
\end{aligned}
$$

with values

$$B_s = 2\pi B_{s0} = 2\pi \left\| [a_{s1} + a_{s2}(\mathbf{A}_s + \hat{\mathbf{A}}_s)]\text{ReLU}(\mathbf{X}\mathbf{W}_{s0})\mathbf{W}_{s1} \right\|_F,$$

$$B_t = 2\pi B_{t0} = 2\pi \left\| [a_{t1} + a_{t2}(\mathbf{A}_t + \hat{\mathbf{A}}_t)]\text{ReLU}(\mathbf{X}\mathbf{W}_{t0})\mathbf{W}_{t1} \right\|_F,$$

$$B_f = 2\pi(B_{fs} + B_{ft}) \tag{14}$$

$$= 2\pi \left\| \mathbf{I}_n + a_{s1}\hat{\mathbf{A}}_s + a_{s2}\hat{\mathbf{A}}_s^2 \right\|_F \|\mathbf{W}_{s1}\|_F \|\mathbf{W}_{s0}\|_F$$

$$+ 2\pi \left\| \mathbf{I}_n + a_{t1}\hat{\mathbf{A}}_t + a_{t2}\hat{\mathbf{A}}_t^2 \right\|_F \|\mathbf{W}_{t1}\|_F \|\mathbf{W}_{t0}\|_F.$$

If we in addition have

$$\|\mathbf{W}_{s0}\|_F \leqslant 1, \quad \|\mathbf{W}_{s1}\|_F \leqslant 1, \quad \|\mathbf{W}_{t0}\|_F \leqslant 1, \quad \|\mathbf{W}_{t1}\|_F \leqslant 1,$$

$$\left\| a_{s1} + a_{s2}(\mathbf{A}_s + \hat{\mathbf{A}}_s) \right\|_F \leqslant 1, \quad \left\| a_{t1} + a_{t2}(\mathbf{A}_t + \hat{\mathbf{A}}_t) \right\|_F \leqslant 1,$$

$$\|\mathbf{X}\mathbf{W}_{s0}\|_F \leqslant 1, \quad \left\| \mathbf{I}_n + a_{s1}\hat{\mathbf{A}}_s + a_{s2}\hat{\mathbf{A}}_s^2 \right\|_F \leqslant 1, \quad \left\| \mathbf{I}_n + a_{t1}\hat{\mathbf{A}}_t + a_{t2}\hat{\mathbf{A}}_t^2 \right\|_F \leqslant 1,$$

then the bound becomes $\left\| \mathbf{r}^{(0)} - \hat{\mathbf{r}}^{(0)} \right\|_F < 2\pi(\epsilon_s + \epsilon_t + 2\epsilon_f)$, as

$$\left\| [a_{s1} + a_{s2}(\mathbf{A}_s + \hat{\mathbf{A}}_s)]\text{ReLU}(\mathbf{X}\mathbf{W}_{s0})\mathbf{W}_{s1} \right\|_F$$

$$\leqslant \left\| a_{s1} + a_{s2}(\mathbf{A}_s + \hat{\mathbf{A}}_s) \right\|_F \|\text{ReLU}(\mathbf{X}\mathbf{W}_{s0})\|_F \|\mathbf{W}_{s1}\|_F$$

$$\leqslant \left\| a_{s1} + a_{s2}(\mathbf{A}_s + \hat{\mathbf{A}}_s) \right\|_F \|\mathbf{X}\mathbf{W}_{s0}\|_F \|\mathbf{W}_{s1}\|_F,$$

and similarly for $B_t$.

This completes the proof. $\square$

**Discussion on GNNSync's robustness to noise**   With the above proposition, we could consider $\hat{\mathbf{A}}$ as the ground-truth noiseless adjacency matrix whose nonzero entries encode $(\theta_i - \theta_j) \bmod 2\pi$, and $\mathbf{A}$ as the actual noisy observed input graph. We then execute row normalization to obtain the source and target matrices $\hat{\mathbf{A}}_s$ and $\hat{\mathbf{A}}_t$ for the ground-truth and $\mathbf{A}_s$ and $\mathbf{A}_t$ for the observation, respectively. In a favourable noise regime, $\epsilon_s$ and $\epsilon_t$ would be small. The value $\epsilon_f$ comes from the feature generation method. For Spectral_RN baseline as input feature generation method for example, this involves some eigensolver corresponding to complex Hermitian matrices, and hence the Davis-Kahan Theorem (Davis & Kahan, 1970) or one of its variants (Li, 1998; Yu et al., 2015) could be applied to upper-bound $\epsilon_f$. As for the values $B_s, B_t$, and $B_f$, we could bound them by adding constraints to GNNSync's model parameters. Employing a backpropagation procedure with our novel loss functions could further boost the robustness of GNNSync, with learnable procedures, as shown for example in Ruiz et al. (2021).

# B   DATA SETS

## B.1   RANDOM GRAPH OUTLIER MODELS

The detailed synthetic data generation process is as follows:

1. Given the number of nodes $n$, generate $k$ group(s) of ground-truth angles $\{\theta_{i,l} : i \in \{1, \ldots, n\}\}$ for $l \in \{1, \ldots, k\}$. One option is to generate each $\theta_{i,l}$ from the same Gamma distribution with shape 0.5 and scale $2\pi$. We denote this option with subscript "1". Since angles could be highly correlated in practical scenarios, we introduce a more realistic but challenging option "2", with multivariate normal ground-truth angles. For example, in the SNL application, angles correspond to patch rotations, and may well be that patches in similar geographic regions have corresponding rotations. The mean of the ground-truth angles is $\pi$, with covariance matrix for each $l \in \{1, \ldots, k\}$ defined by $\mathbf{w}\mathbf{w}^\top$, where entries in $\mathbf{w}$ are generated independently from a standard normal distribution. We then apply mod $2\pi$ to all angles to ensure that they lie in $[0, 2\pi)$. We thus obtain the ground-truth adjacency matrix (matrices) $\mathbf{A}_{\text{GT}}^l \in \mathbb{R}^{n \times n}$, whose $(i, j)$ element is given by $(\theta_{i,l} - \theta_{j,l}) \bmod 2\pi$.

2. Generate a noisy background adjacency matrix $\mathbf{A}_{\text{noise}} \in \mathbb{R}^{n \times n}$ whose entries are independently generated from a uniform distribution over $[0, 2\pi)$.

3. Generate a selection matrix $\mathbf{A}_{\text{sel}} \in \mathbb{R}^{n \times n}$ whose entries are independently drawn from a Uniform(0,1) distribution. The $(i, j)$ entry of this selection matrix is used to assign whether or not the observation is noisy, and if not noisy, to which graph it is assigned, using for $l = 1, \ldots, k$

$$B_{\text{noise}}(i, j; l) = \mathbb{1}\big((1 - \eta)(l - 1)/k \leqslant \mathbf{A}_{\text{sel}}(i, j) < (1 - \eta)l/k\big)$$

and $B_{\text{noise}}(i, j; \infty) = \mathbb{1}\big(\mathbf{A}_{\text{sel}}(i, j) \geqslant (1 - \eta)\big)$, where $\mathbb{1}(\cdot)$ is the indicator function.

4. Construct a complete (without self-loops) weighted adjacency matrix $\mathbf{A}_{\text{complete}} \in \mathbb{R}^{n \times n}$ by $\mathbf{A}_{\text{complete}}(i, j) = \sum_{l \in \{1, \ldots, k\}} \mathbf{A}_{\text{GT}}^{(l)}(i, j) B_{\text{noise}}(i, j; l) + \mathbf{A}_{\text{noise}}(i, j) B_{\text{noise}}(i, j; \infty)$.

5. Generate a measurement graph $\bar{\mathcal{G}}$ with adjacency matrix $\bar{\mathbf{G}}$ using a standard random graph model, as introduced by Sec. 6.1.

6. The edges in $\bar{\mathcal{G}}$ are the edges on which we observe the noisy version $\mathbf{A}_{\text{complete}}$ of the ground-truth adjacency matrix (matrices) $\mathbf{A}_{\text{GT}}^l$, to obtain the temporary adjacency matrix $\mathbf{T}_1$ by $\mathbf{T}_1(i, j) = \mathbf{A}_{\text{complete}}(i, j) \mathbb{1}(\bar{\mathbf{G}}(i, j) \neq 0)$.

7. The true angle differences would yield a skew-symmetric matrix before taking the entries mod $2\pi$. We therefore construct a skew-symmetric matrix $\mathbf{T}_2$ by setting $\mathbf{T}_2(i, i) = 0$ for all $i$, and for $i \neq j$ setting $\mathbf{T}_2(i, j) = \mathbf{T}_1(i, j)\mathbb{1}(i < j) - \mathbf{T}_1(j, i)\mathbb{1}(i > j)$.

8. In the skew-symmetric matrix, each entry appears twice, with different signs. For computational reasons, for the final adjacency matrix, we only keep the non-negative entries, except for evaluation. We obtain the final adjacency matrix $\mathbf{A}$ by $\mathbf{A}_{i,j} = \mathbf{A}(i, j) = \mathbf{T}_2(i, j)\mathbb{1}(\mathbf{T}_2(i, j) \geqslant 0) \bmod 2\pi$.

In addition to the two options introduced in Sec. 6.1, we introduce two more options for the ground-truth angle generation process here, which are both multivariate normal distributions, but with different covariance matrices. For option "3", the covariance matrix is just the identity matrix. For option "4", we have a block-diagonal covariance matrix, with six blocks, each of which is generated independently according to option "2" as stated in Sec. 6.1 in the main text.

As methods could be applied to different connected components of disconnected graphs separately, we focus on weakly connected networks. We have checked that all generated networks in our experiments are weakly connected.

The reason behind the naming convention "outlier" stems from the noisy offset entries in the adjacency matrix.

Steps 7 and 8 are to ensure that there does not exist an edge $(i, j)$ such that $(\mathbf{A}_{i,j} + \mathbf{A}_{j,i}) \bmod 2\pi \neq 0$, as this would be confusing. In principle, we could work with the upper-triangular part or the lower-triangular part of the adjacency matrix first, then obtain the skew-symmetric adjacency matrix $\mathbf{T}_2$ and apply step 8 again. The procedures mentioned in Sec. 6.1 are what we implement in practice, which should take no more than twice the computational cost compared to working with half of the adjacency matrix at the beginning. Note that data generation only happens once before running the actual experiments.

Our synthetic data settings are similar to those in previous angular synchronization papers, such as Singer (2011); Lerman & Shi (2022); Cucuringu & Tyagi (2022), to generate noisy samples from an outlier model (where each outlier measurement is generated uniformly at random), instead of using additive Gaussian noise in the so-called spike models. The choice of the number of nodes 360 could be changed to other numbers; we chose it to relate to 360 possible integer degrees of an angle. Note that the initial work of Singer (2011) considered $n$ random rotation angles uniformly distributed in $[0, 2\pi)$. We do not observe a large difference in the performance of other sizes (we have also tried 300 and 500, for example).

The choice of synthetic data set construction is inspired by Singer (2011) and Cucuringu & Tyagi (2022). They are noisy versions of standard random graph models. These random graph models were chosen as they can be used for comparison. Indeed, some previous works have only used ER measurement graphs as in Lerman & Shi (2022), and Singer (2011) theoretically analyzed and

experimented with both sparse ER and complete measurement graphs; we already have a more thorough setup in our experiments. Furthermore, the addition of the RGG model stems from the very fact that this model is perhaps the most representative one, given the applications that have motivated the development of the group synchronization problem over the last decade. Indeed, in sensor network localization or the *molecule problem* in NMR structural biology, pairwise Euclidean distance information is only available between nearby sensors or atoms (e.g., certain sensors/atoms are connected if at most 6 miles/angstrom apart), hence leading to an RGG (disc graph) model. In this setup, in order to recover the latent coordinates, the state-of-the-art methods rely on divide-and-conquer approaches that first divide the graph into overlapping subgraphs /patches, embed locally to take advantage of the higher edge density locally, and finally aim to **stitch globally**, which is where group synchronization comes into play. Therefore, any patch-based reconstruction method that leverages the local geometry is only able to pairwise align only nearby patches that have enough points in common; far away patches that do not overlap simply cannot be aligned. Thus, the choice of RGG resembles best the real-world applications. The ER model has been predominantly used in the literature as it is easier to analyze theoretically compared to RGG, in light of available tools from the random matrix theory literature.

## B.2 SENSOR NETWORK LOCALIZATION

Previous works in the field of angular synchronization typically only consider synthetic data sets in their experiments, and those applying synchronization to real-world data do not typically publish the data sets. Concrete examples for such works include tracking the trajectory of a moving object using directional sensors (Plarre & Kumar, 2005), and habitat monitoring in an infrastructure-less environment in which radios are turned on at designated times to save power on Great Duck Island (Mainwaring et al., 2002).

In this paper, we adapt the task on group synchronization over the Euclidean group of rigid motions $Euc(2) = \mathbb{Z}_2 \times SO(2) \times \mathbb{R}^2$ to a real-world task, by focusing on the angular synchronization $SO(2)$ component. This task on a real-world data set is a special case of the sensor network localization (SNL) task on the plane ($\mathbb{R}^2$) mentioned in Cucuringu et al. (2012a), but we focus on synchronization over $SO(2)$ only, as we do not consider any translations or reflections. Though we do not have purely real-world data sets that are employed in practice, we mimic the practical task of sensor network localization (with a focus on rotation only) and conduct the localization task on the U.S. map as well as a PACM point cloud. In detail, the task is conducted as follows, where Fig. 5 provides an overview of the pipeline on the U.S. map with an illustrative example:

1. Starting with the ground-truth locations of U.S. cities (we have $n = 1097$ cities, see red dots in Fig. 5), we construct patches using each city as a central node and add its $k_{\text{patch}} = 50$ nearest neighbors to the corresponding patch (see Fig. 5(a)). We then obtain $m = n = 1097$ patches (see Fig. 5(b) for a two-patch example). This is to represent sensor patches in the real world.

2. For each patch, we add noise to each node's coordinates using independent normal distributions for x and y coordinates respectively, with mean zero and standard deviation $\eta$ times of x and y coordinates' standard deviation, respectively (see Fig. 5(c)). Note that the noise added to the same node is independent for different patches. This is to represent noisy observations due to the lack of use of the expensive GPS service to estimate sensor coordinates.

3. We then rotate the patches based on some ground-truth rotation angles $\theta_1, \ldots, \theta_n$ (see Fig. 5(d)). Here we generate the angles using one of the options introduced in Sec. 6.1 and Sec. B.1. This again is to represent noisy observations in the real world.

4. Then for each pair of the patches that have at least $k_{\text{thres}} = 6$ overlapping nodes, we apply Procrustes alignment (Gower, 1975) to estimate the rotation angle based on these overlapping nodes (but with noisy coordinates) and add an edge with the weight the estimated rotation angle to the observed (measurement) adjacency matrix $\mathbf{A}$. In other words, if two patches $P_i, P_j$ that have at least $k_{\text{thres}} = 6$ overlapping nodes, we have $\mathbf{A}_{i,j}$ the estimated rotation angle from Procrustes alignment to rotate $P_j$ to align with $P_i$. This angle is an estimation of $\theta_i - \theta_j$. The threshold is set to represent the real-world scenario where only nearby sensors may communicate with each other.

5. After that, we perform angular synchronization on the sparse adjacency matrix $\mathbf{A}$ (retaining only the upper triangular entries) to obtain the initial estimated angles $r_1^{(0)}, \ldots, r_n^{(0)}$ for each patch.

6. We then update the estimated angles by shifting by the average of pairwise differences between the estimated and ground-truth angles, in order to mod out the global degree of freedom from the synchronization step (see Fig. 5(e)). That is, we first calculate the average of pairwise differences by $\delta_{\text{pairwise}} = \frac{1}{n} \sum_{i=1}^{n} [(r_i^{(0)} - \theta_i) \bmod 2\pi]$, then set $r_i = (r_i^{(0)} - \delta_{\text{pairwise}}) \bmod 2\pi, i = 1, \ldots, n$.

7. Next, we apply the estimated rotations to the noisy patches.

8. Finally, we estimate the city coordinates by averaging the estimated locations for each city (node) across patches that contain this city (node) (see Fig. 5(f)).

Note that the noise in the observed adjacency matrix originates from the error by Procrustes alignment, with possible noise added to nodes' coordinates. Therefore, even when $\eta = 0$, the observed adjacency matrix may not align perfectly with ground-truth pairwise angular offsets. Besides, the observed adjacency matrix is sparse instead of complete due to the thresholding set up to only connect two nodes in the graph if the patches have enough overlapping nodes. In our experiments, we vary $\eta$ from $[0, 0.05, 0.1, 0.15, 0.2, 0.25]$.

Our current experiment is focused on group synchronization over the group SO(2), and in future work we plan to explore synchronization over the full Euclidean group $\text{Euc}(2) = \mathbb{Z}_2 \times \text{SO}(2) \times \mathbb{R}^2$, similar to Cucuringu et al. (2012a), where in addition to rotations, both reflections and translations are considered and synchronized over.

## C    IMPLEMENTATION DETAILS

### C.1    SETUP

We use the whole graph for training for at most 1000 epochs, and stop early if the loss value does not decrease for 200 epochs. We use Stochastic Gradient Descend (SGD) as the optimizer and $\ell_2$ regularization with weight decay $5 \cdot 10^{-4}$ to avoid overfitting. We use as learning rate 0.005 throughout.

For each synthetic data set, we generate 5 synthetic networks under the same setting, each with 2 repeated runs.

The DIMPA model is inherited from He et al. (2022b). Indeed, other directed graph embedding neural network methods such as Tong et al. (2020) and Zhang et al. (2021) could be employed, and we pick DIMPA just for simplicity. In our experiments, we did try out Tong et al. (2020), and we do not observe much difference in the performance as long as some directed graph embeddings could be produced.

### C.2    CODES, DATA AND HARDWARE

To fully reproduce our results, anonymized code is available at `https://github.com/SherylHYX/GNN_Sync`. Experiments were conducted on two compute nodes, each with 8 Nvidia Tesla T4, 96 Intel Xeon Platinum 8259CL CPUs @ 2.50GHz and 378GB RAM. All experiments can be completed within several days, including all variants.

The data sets considered here are relatively small and the same applies to GNNSync's competitive papers. Although each individual task does not require many resources (often < 5min/run), for the

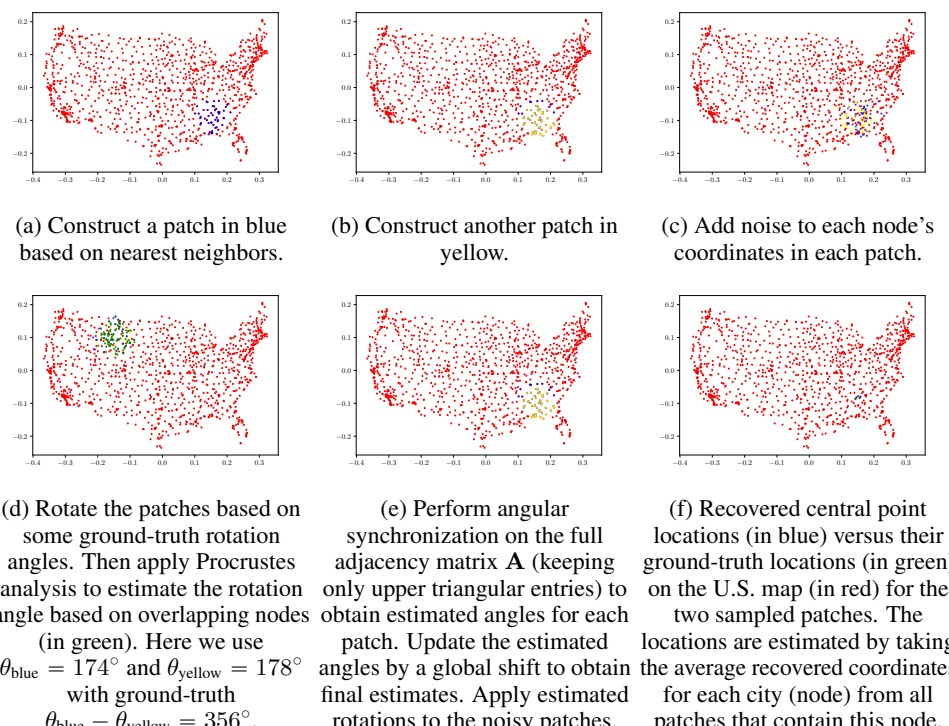

(a) Construct a patch in blue based on nearest neighbors.

(b) Construct another patch in yellow.

(c) Add noise to each node's coordinates in each patch.

(d) Rotate the patches based on some ground-truth rotation angles. Then apply Procrustes analysis to estimate the rotation angle based on overlapping nodes (in green). Here we use $\theta_{\text{blue}} = 174°$ and $\theta_{\text{yellow}} = 178°$ with ground-truth $\theta_{\text{blue}} - \theta_{\text{yellow}} = 356°$.

(e) Perform angular synchronization on the full adjacency matrix $\mathbf{A}$ (keeping only upper triangular entries) to obtain estimated angles for each patch. Update the estimated angles by a global shift to obtain final estimates. Apply estimated rotations to the noisy patches.

(f) Recovered central point locations (in blue) versus their ground-truth locations (in green) on the U.S. map (in red) for the two sampled patches. The locations are estimated by taking the average recovered coordinates for each city (node) from all patches that contain this node.

Figure 5: U.S. city patch localization pipeline with two patches as an example: Starting with the ground-truth locations of U.S. cities, we construct patches using each city as a central node and add its $k_{\text{patch}} = 50$ nearest neighbors to the corresponding patch. We then add noise to each node's coordinates using independent normal distributions for x and y coordinates respectively, with mean zero and standard deviation $\eta = 0.05$ times of x and y coordinates' standard deviation, respectively. We then rotate the patches based on some ground-truth rotation angles from option "2" introduced in Sec. 4.1. Here we use $\theta_{\text{blue}} = 174°$ and $\theta_{\text{yellow}} = 178°$ with ground-truth $\theta_{\text{blue}} - \theta_{\text{yellow}} = 356°$. The estimated rotation angle from the blue patch to the yellow one is $\mathbf{A}_{\text{blue, yellow}} = 6.25$ (i.e., $358°$). Then we apply Procrustes analysis to estimate the rotation angle based on overlapping nodes (but with noisy coordinates). After that, we perform angular synchronization on the full adjacency matrix $\mathbf{A}$ (keeping only upper triangular entries) to obtain estimated angles for each patch. We then update the estimated angles by shifting by the average of pairwise differences between the estimated and ground-truth angles. Here we have estimates $r_{\text{blue}} = 173°$ and $r_{\text{yellow}} = 175°$ with $r_{\text{blue}} - r_{\text{yellow}} = 358°$. Then we apply estimated rotations to the noisy patches. Finally, we obtain recovered central point locations for the two sampled patches. The locations are estimated by taking the average recovered coordinates for each city (node) from all patches that contain this node. The recovered points are colored in blue, while their ground-truth locations are colored in green.

synthetic data sets in this paper we have

$$
\begin{aligned}
& 3(\text{measurement graph styles for } k = 1) \cdot 10(\text{noise levels}) \\
& \quad \cdot 3(\text{sparsity levels}) \cdot 4(\text{ground-truth options}) \\
+ \quad & 3(\text{number of larger } k \text{ values}) \cdot 6(\text{measurement graph} \\
& \text{options for } k > 1) \cdot 8(\text{noise levels}) \\
& \quad \cdot 3(\text{sparsity levels}) \cdot 4(\text{ground-truth options}) \\
= \quad & 360 + 1,728 = 2,088
\end{aligned}
$$

synthetic data sets. Each data set requires 10 runs for each of the

$$
\begin{aligned}
& 1(\text{main results}) \\
& \quad +6(\text{different baselines as input features}) \\
& \quad +1(\text{no Projected Gradient Steps}) \\
& \quad +1(\text{trainable } \alpha) \\
& = \quad 9
\end{aligned}
$$

variants for the regular angular synchronization ($k = 1$) and

$$
\begin{aligned}
& 3(\text{main results}) \\
& \quad +2(\text{different baselines as input features}) \\
& \quad +2(\text{different baseline}) \\
& \quad +2(\text{trainable } \{\alpha_\gamma\}) \\
& \quad +2(\text{no Projected Gradient Steps}) \\
& \quad +2(\text{separate } \mathbf{H}^{(l)}) \\
& \quad +5(\text{other linear combinations of the loss function}) \\
& = \quad 18
\end{aligned}
$$

variants for general $k$-synchronization with $k \in \{2, 3, 4\}$. Therefore, the set of tasks in this paper requires a total of $360 \cdot 10 \cdot 9 + 1,728 \cdot 10 \cdot 18 = 32,400 + 311,040 = 343,440$ runs.

The baselines are typically faster, as they do not involve training, but GNNSync is also pretty computationally friendly, not at a significantly higher computational expense. Indeed, the set of all cycles could be pre-computed before training. For $k$-synchronization, we only need to verify whether all edges in a cycle are contained in an estimated graph, and to keep only these cycles for computation. There is a loop that repeats $k$ times, but for each loop, only matrix operations are involved, which can be done in parallel for all possible cycles. This would not be too expensive, as validated by our experiments. Besides, for the MSE function, we do not use it for training, so it is not a loss function in the first place. For evaluation, it does require computing all permutations of $k$ but the evaluation is only conducted once. Therefore, these computationally expensive operations (locating all possible cycles and permutations of $k$ in the MSE computation) are not involved in training, but only before or after training, and hence our method is still scalable with $n$ and $k$.

## C.3 BASELINE IMPLEMENTATION

For CEMP_GCW and CEMP_MST, we adapt the MatLab code from `https://github.com/yunpeng-shi/CEMP/tree/main/SO2` to Python. For other baselines, we implement the approaches based on equations from the original papers. For TAS, we transform the MatLab codes from the authors of Maunu & Lerman (2023) to Python. We set the number of epochs to 50, and set the trimming parameter to zero due to the high sparsity level of our synthetic networks, as otherwise, almost all predictions would be the same, just like the Trivial solution.

Besides, we do not compare GNNSync against the method in Gao & Zhao (2019) in our experiments, as there is no code available. Also, their algorithm involves integration and angular argmax in each iteration, which seems to be computationally expensive.

Finally, we are aware of Semi-Definite Programming (SDP) baselines but have found them too time-consuming or space-inefficient. Also, from Singer (2011), we know that SDP and spectral methods have comparable performance. Therefore, in our experiments, we omit the SDP results.

## C.4 MSE CALCULATION

As stated in eq. (2), the MSE function calculates the mean square error using a global angular rotation that minimizes the MSE value. The implementation of the MSE function, however, does not explicitly search for the lowest MSE value through grid search or gradient descent. Inspired by the implementation of the MSE in Singer & Shkolnisky (2011), we first map each of the predicted angles $\mathbf{r}$ and the ground-truth angles $\mathbf{R}$ to rotation matrices by the mapping function

$$
rot(\theta) = \begin{bmatrix} \cos(\theta) & -\sin(\theta) \\ \sin(\theta) & \cos(\theta) \end{bmatrix}.
$$

Table 1: Average MSE values (plus/minus one standard deviation) for the real-world experiments on the U.S. map over ten runs. The best is marked in **bold red** while the second best is in underline blue.

| $\eta$ | option | GNNSync | Spectral | Spectral_RN | GPM | TranSync | CEMP_GCW | CEMP_MST | TAS | Trivial |
|---|---|---|---|---|---|---|---|---|---|---|
| 0 | 1 | 0.010±0.006 | **0.000±0.000** | **0.000±0.000** | **0.000±0.000** | **0.000±0.000** | **0.000±0.000** | **0.000±0.000** | 2.358±0.075 | 2.442±0.069 |
| 0 | 2 | 0.004±0.001 | **0.000±0.000** | **0.000±0.000** | **0.000±0.000** | **0.000±0.000** | **0.000±0.000** | **0.000±0.000** | 1.567±0.035 | 1.566±0.037 |
| 0 | 3 | **0.000±0.000** | **-0.000±0.000** | **-0.000±0.000** | **-0.000±0.000** | **-0.000±0.000** | **-0.000±0.000** | **-0.000±0.000** | 0.138±0.174 | 0.138±0.174 |
| 0 | 4 | 0.002±0.001 | **0.000±0.000** | **0.000±0.000** | **0.000±0.000** | **0.000±0.000** | **0.000±0.000** | **0.000±0.000** | 0.651±0.237 | 0.663±0.238 |
| 0.05 | 1 | 0.014±0.006 | 0.007±0.001 | **0.006±0.001** | **0.006±0.001** | 0.036±0.027 | 0.082±0.040 | 1.162±0.438 | 2.340±0.098 | 2.411±0.094 |
| 0.05 | 2 | 0.010±0.002 | **0.006±0.000** | **0.006±0.000** | **0.006±0.000** | 0.026±0.006 | 0.072±0.006 | 1.353±0.461 | 1.559±0.032 | 1.559±0.033 |
| 0.05 | 3 | **0.006±0.002** | 0.007±0.001 | **0.006±0.001** | **0.006±0.001** | 0.027±0.009 | 0.816±0.494 | 2.827±0.980 | 0.293±0.481 | 0.294±0.482 |
| 0.05 | 4 | 0.007±0.002 | **0.006±0.001** | **0.006±0.000** | **0.006±0.001** | 0.024±0.007 | 0.319±0.251 | 1.826±0.918 | 0.750±0.373 | 0.759±0.372 |
| 0.1 | 1 | 0.030±0.005 | 0.030±0.007 | 0.027±0.006 | **0.025±0.005** | 0.147±0.026 | 0.528±0.077 | 2.819±0.292 | 2.318±0.104 | 2.368±0.101 |
| 0.1 | 2 | **0.025±0.002** | 0.031±0.002 | 0.028±0.002 | 0.027±0.001 | 0.188±0.053 | 0.397±0.048 | 2.874±0.315 | 1.558±0.030 | 1.564±0.028 |
| 0.1 | 3 | **0.019±0.003** | 0.027±0.005 | 0.024±0.004 | 0.025±0.004 | 0.129±0.048 | 1.775±1.104 | 3.583±0.298 | 0.138±0.175 | 0.138±0.174 |
| 0.1 | 4 | **0.021±0.006** | 0.026±0.003 | 0.023±0.002 | 0.023±0.002 | 0.127±0.033 | 1.055±0.616 | 3.102±0.205 | 0.656±0.238 | 0.663±0.238 |
| 0.15 | 1 | **0.059±0.008** | 0.075±0.012 | 0.072±0.013 | 0.062±0.008 | 0.483±0.279 | 1.496±0.473 | 3.698±0.092 | 2.363±0.117 | 2.396±0.118 |
| 0.15 | 2 | **0.057±0.005** | 0.082±0.005 | 0.076±0.007 | 0.067±0.003 | 0.492±0.122 | 1.193±0.238 | 3.501±0.301 | 1.566±0.037 | 1.566±0.037 |
| 0.15 | 3 | **0.037±0.006** | 0.052±0.011 | 0.048±0.009 | 0.046±0.010 | 0.277±0.064 | 1.745±0.540 | 3.671±0.229 | 0.138±0.175 | 0.138±0.174 |
| 0.15 | 4 | **0.043±0.007** | 0.055±0.008 | 0.052±0.006 | 0.046±0.005 | 0.591±0.384 | 1.317±0.174 | 3.519±0.152 | 0.658±0.239 | 0.663±0.238 |
| 0.2 | 1 | **0.101±0.007** | 0.148±0.024 | 0.151±0.019 | 0.107±0.009 | 1.000±0.257 | 2.568±0.906 | 3.711±0.122 | 2.377±0.095 | 2.399±0.093 |
| 0.2 | 2 | **0.101±0.006** | 0.163±0.017 | 0.159±0.018 | 0.122±0.009 | 1.054±0.245 | 2.082±0.263 | 3.717±0.109 | 1.564±0.037 | 1.566±0.037 |
| 0.2 | 3 | **0.065±0.015** | 0.095±0.018 | 0.092±0.019 | 0.078±0.015 | 0.756±0.222 | 2.239±0.556 | 3.699±0.111 | 0.335±0.377 | 0.336±0.380 |
| 0.2 | 4 | **0.066±0.008** | 0.096±0.017 | 0.095±0.019 | 0.072±0.010 | 0.717±0.258 | 2.040±0.334 | 3.751±0.089 | 0.660±0.239 | 0.663±0.238 |
| 0.25 | 1 | **0.158±0.008** | 0.244±0.038 | 0.263±0.039 | 0.164±0.011 | 1.690±0.569 | 2.888±0.240 | 3.791±0.096 | 2.427±0.069 | 2.442±0.069 |
| 0.25 | 2 | **0.163±0.013** | 0.291±0.038 | 0.294±0.042 | 0.193±0.018 | 1.888±0.737 | 2.633±0.294 | 3.782±0.108 | 1.564±0.037 | 1.566±0.037 |
| 0.25 | 3 | 0.105±0.065 | 0.143±0.018 | 0.242±0.105 | **0.103±0.021** | 1.265±0.583 | 3.050±0.552 | 3.765±0.081 | 0.138±0.174 | 0.138±0.174 |
| 0.25 | 4 | 0.128±0.074 | 0.170±0.029 | 0.176±0.040 | **0.107±0.017** | 1.296±0.320 | 2.787±0.477 | 3.787±0.070 | 0.660±0.238 | 0.663±0.238 |

We then calculate the matrix

$$\mathbf{Q} = \frac{1}{n} \sum_{i=1}^{n} rot(\mathbf{R}_i)^{\top} \cdot rot(\mathbf{r}_i).$$

The MSE value is given by

$$4 - 2 \sum_{i=1}^{n} sing_i(\mathbf{Q}),$$

where $sing_i(\mathbf{Q})$ is the $i$-th singular value of $\mathbf{Q}$ during Singular Value Decomposition (SVD).

## D  EXTENDED EXPERIMENTAL RESULTS

This section reports extended experimental results mentioned in the main text.

### D.1  EXTENDED MAIN RESULTS

Full main synthetic experimental results are shown in Fig. 6 to 17. Results on real-world data sets are shown in Fig. 18 to 27, while other PACM results are omitted but with the same conclusion. To accommodate potential variability in different runs, we report the mean and standard deviation of ten runs (two repeated runs on five different sets of ground-truth angles) in Tab. 1 and 3. We also compute the Average Normalized Error (ANE) for coordinate recovery similar to eq. (44) of Cucuringu et al. (2012a), and report mean and one standard deviation of the results in Tab. 2 and 4. Specifically, denote $(x_i, y_i)$ as the ground-truth coordinate for node $i$ where $i = 1, \ldots, n$, and $(\hat{x}_i, \hat{y}_i)$ as the predicted coordinate, we define the Average Normalized Error (ANE) as

$$ANE = \frac{\sqrt{\sum_{i=1}^{n}[(x_i - \hat{x}_i)^2 + (y_i - \hat{y}_i)^2]}}{\sqrt{\sum_{i=1}^{n}[(x_i - x_0)^2 + (y_i - y_0)^2]}}, \tag{15}$$

where $(x_0, y_0) = (\frac{1}{n} \sum_{i=1}^{n} x_i, \frac{1}{n} \sum_{i=1}^{n} y_i) = (0, 0)$ is the center of mass of the true coordinates. We conclude that GNNSync is able to effectively recover coordinates. We also observe that GNNSync is more robust to the noise of patch coordinates. We omit the visual plots for the "Trivial" baseline but report its performance in Tab. 1, 2, 3, and 4.

Table 2: Average ANE values (plus/minus one standard deviation) for the real-world experiments on the U.S. map over ten runs. The best is marked in **bold red** while the second best is in _underline blue_.

| η | option | GNNSync | Spectral | Spectral_RN | GPM | TranSync | CEMP_GCW | CEMP_MST | TAS | Trivial |
|---|---|---|---|---|---|---|---|---|---|---|
| 0 | 1 | 0.075±0.028 | **0.000±0.000** | **0.000±0.000** | **0.000±0.000** | **0.000±0.000** | **0.000±0.000** | **0.000±0.000** | 0.740±0.021 | 0.973±0.263 |
| 0 | 2 | 0.047±0.010 | **0.000±0.000** | **0.000±0.000** | **0.000±0.000** | **0.000±0.000** | **0.000±0.000** | **0.000±0.000** | 0.425±0.015 | 0.423±0.014 |
| 0 | 3 | 0.011±0.009 | **0.000±0.000** | **0.000±0.000** | **0.000±0.000** | **0.000±0.000** | **0.000±0.000** | **0.000±0.000** | 0.050±0.049 | 0.049±0.049 |
| 0 | 4 | 0.030±0.016 | **0.000±0.000** | **0.000±0.000** | **0.000±0.000** | **0.000±0.000** | **0.000±0.000** | **0.000±0.000** | 0.213±0.078 | 0.219±0.078 |
| 0.05 | 1 | 0.074±0.027 | **0.032±0.010** | 0.032±0.011 | 0.032±0.009 | 0.360±0.591 | 0.121±0.069 | 0.709±0.495 | 0.735±0.025 | 0.984±0.277 |
| 0.05 | 2 | _0.054±0.013_ | 0.284±0.508 | **0.030±0.002** | 0.159±0.258 | 0.092±0.020 | 0.146±0.035 | 0.638±0.238 | 0.421±0.015 | 0.419±0.013 |
| 0.05 | 3 | **0.030±0.017** | 0.118±0.160 | _0.035±0.005_ | 0.039±0.008 | 0.469±0.763 | 0.708±0.338 | 0.900±0.178 | 0.089±0.124 | 0.089±0.124 |
| 0.05 | 4 | 0.229±0.569 | 0.404±0.744 | **0.031±0.007** | _0.032±0.008_ | 0.058±0.009 | 0.494±0.595 | 0.750±0.369 | 0.243±0.121 | 0.250±0.122 |
| 0.1 | 1 | _0.078±0.014_ | **0.070±0.024** | 0.204±0.152 | 0.160±0.181 | 0.164±0.050 | 0.568±0.409 | 1.068±0.135 | 0.733±0.028 | 0.987±0.280 |
| 0.1 | 2 | **0.057±0.017** | 0.077±0.007 | _0.076±0.005_ | 0.259±0.355 | 0.270±0.042 | 0.345±0.042 | 0.992±0.161 | 0.415±0.016 | 0.414±0.016 |
| 0.1 | 3 | **0.046±0.015** | 0.085±0.017 | 0.079±0.018 | 0.324±0.466 | 0.503±0.503 | 0.920±0.298 | 1.019±0.109 | _0.053±0.047_ | _0.053±0.047_ |
| 0.1 | 4 | 0.247±0.583 | _0.075±0.019_ | **0.069±0.016** | 0.124±0.119 | 0.226±0.135 | 0.828±0.382 | 0.964±0.146 | 0.215±0.079 | 0.219±0.078 |
| 0.15 | 1 | 0.313±0.449 | _0.216±0.153_ | 0.602±0.713 | **0.104±0.022** | 0.313±0.027 | 0.859±0.422 | 1.019±0.071 | 0.755±0.032 | 1.101±0.257 |
| 0.15 | 2 | **0.076±0.019** | 0.573±0.679 | 0.151±0.062 | _0.136±0.026_ | 0.460±0.304 | 0.983±0.460 | 0.964±0.065 | 0.425±0.015 | 0.424±0.014 |
| 0.15 | 3 | 0.181±0.347 | 0.189±0.165 | 0.102±0.024 | 0.107±0.025 | 0.524±0.527 | 0.787±0.145 | 0.979±0.105 | **0.057±0.045** | **0.057±0.045** |
| 0.15 | 4 | **0.075±0.032** | 0.104±0.030 | 0.096±0.020 | _0.088±0.024_ | 0.929±0.522 | 0.947±0.460 | 1.019±0.060 | 0.215±0.077 | 0.220±0.078 |
| 0.2 | 1 | _0.131±0.026_ | 0.247±0.203 | 0.612±0.375 | **0.125±0.034** | 0.835±0.435 | 0.876±0.140 | 1.087±0.058 | 0.749±0.027 | 0.976±0.267 |
| 0.2 | 2 | **0.094±0.018** | 0.451±0.332 | _0.225±0.109_ | 0.433±0.506 | 0.609±0.234 | 0.967±0.272 | 1.005±0.038 | 0.424±0.017 | 0.424±0.014 |
| 0.2 | 3 | **0.087±0.037** | 0.133±0.025 | _0.130±0.028_ | 0.132±0.030 | 0.964±0.649 | 0.889±0.211 | 1.020±0.053 | _0.109±0.092_ | 0.110±0.093 |
| 0.2 | 4 | **0.077±0.023** | 0.123±0.029 | 0.126±0.025 | _0.103±0.027_ | 0.724±0.484 | 0.911±0.217 | 0.999±0.031 | 0.215±0.076 | 0.221±0.077 |
| 0.25 | 1 | **0.197±0.066** | _0.223±0.101_ | 0.337±0.255 | 0.478±0.418 | 0.656±0.188 | 0.978±0.133 | 1.008±0.032 | 0.760±0.022 | 0.974±0.263 |
| 0.25 | 2 | **0.122±0.034** | 0.494±0.399 | _0.393±0.219_ | 0.546±0.701 | 0.912±0.215 | 0.895±0.061 | 1.030±0.025 | 0.425±0.017 | 0.425±0.014 |
| 0.25 | 3 | 0.260±0.397 | 0.281±0.215 | 0.230±0.065 | 0.157±0.038 | 1.193±0.429 | 1.085±0.153 | 1.009±0.052 | **0.067±0.041** | **0.067±0.041** |
| 0.25 | 4 | **0.134±0.109** | 0.513±0.662 | _0.188±0.047_ | 0.288±0.343 | 0.612±0.174 | 0.958±0.147 | 1.042±0.048 | 0.217±0.074 | 0.222±0.076 |

Table 3: Average MSE values (plus/minus one standard deviation) for the real-world experiments on the PACM point cloud over ten runs. The best is marked in **bold red** while the second best is in _underline blue_.

| η | option | GNNSync | Spectral | Spectral_RN | GPM | TranSync | CEMP_GCW | CEMP_MST | TAS | Trivial |
|---|---|---|---|---|---|---|---|---|---|---|
| 0 | 1 | 0.010±0.013 | **-0.000±0.000** | **-0.000±0.000** | **-0.000±0.000** | **-0.000±0.000** | **-0.000±0.000** | **-0.000±0.000** | 2.249±0.128 | 2.468±0.139 |
| 0 | 2 | 0.001±0.001 | **0.000±0.000** | **0.000±0.000** | **0.000±0.000** | **0.000±0.000** | **0.000±0.000** | **0.000±0.000** | 1.640±0.041 | 1.565±0.042 |
| 0 | 3 | 0.002±0.002 | **0.000±0.000** | **0.000±0.000** | **0.000±0.000** | **0.000±0.000** | **0.000±0.000** | **0.000±0.000** | 1.221±0.779 | 1.160±0.783 |
| 0 | 4 | 0.001±0.001 | **0.000±0.000** | **0.000±0.000** | **0.000±0.000** | **0.000±0.000** | **0.000±0.000** | **0.000±0.000** | 1.196±0.430 | 1.176±0.426 |
| 0.05 | 1 | 0.015±0.011 | **0.003±0.000** | 0.003±0.001 | 0.003±0.000 | 0.010±0.004 | 0.039±0.008 | 0.202±0.093 | 2.246±0.129 | 2.468±0.139 |
| 0.05 | 2 | 0.004±0.001 | 0.003±0.000 | **0.002±0.000** | **0.002±0.000** | 0.010±0.003 | 0.024±0.016 | 0.885±1.157 | 1.611±0.057 | 1.543±0.051 |
| 0.05 | 3 | 0.004±0.002 | **0.003±0.000** | 0.003±0.000 | 0.003±0.000 | 0.009±0.004 | 0.057±0.058 | 0.528±0.496 | 1.218±0.777 | 1.160±0.783 |
| 0.05 | 4 | **0.003±0.001** | **0.003±0.000** | 0.003±0.000 | 0.003±0.000 | 0.011±0.012 | 0.177±0.206 | 0.718±0.292 | 1.006±0.246 | 0.987±0.253 |
| 0.1 | 1 | 0.015±0.005 | **0.012±0.004** | **0.012±0.004** | **0.012±0.004** | 0.049±0.014 | 0.187±0.107 | 1.580±0.642 | 2.122±0.150 | 2.346±0.135 |
| 0.1 | 2 | **0.010±0.001** | 0.011±0.001 | **0.010±0.001** | **0.010±0.001** | 0.044±0.011 | 0.145±0.031 | 1.599±0.459 | 1.631±0.036 | 1.565±0.042 |
| 0.1 | 3 | 0.012±0.003 | **0.010±0.001** | **0.010±0.001** | **0.010±0.001** | 0.055±0.025 | 0.298±0.377 | 1.394±0.659 | 1.213±0.775 | 1.160±0.783 |
| 0.1 | 4 | **0.010±0.001** | 0.011±0.001 | **0.010±0.001** | **0.010±0.001** | 0.067±0.052 | 0.271±0.148 | 1.934±0.656 | 1.191±0.431 | 1.176±0.426 |
| 0.15 | 1 | 0.032±0.009 | 0.029±0.012 | **0.028±0.011** | _0.028±0.012_ | 0.136±0.088 | 0.380±0.192 | 2.009±0.657 | 2.262±0.136 | 2.468±0.139 |
| 0.15 | 2 | 0.022±0.002 | 0.022±0.002 | **0.020±0.002** | _0.021±0.002_ | 0.109±0.034 | 0.522±0.528 | 2.927±0.254 | 1.587±0.064 | 1.530±0.057 |
| 0.15 | 3 | 0.021±0.005 | 0.020±0.004 | **0.019±0.004** | **0.019±0.004** | 0.107±0.035 | 0.523±0.416 | 2.970±0.502 | 0.570±0.468 | 0.524±0.446 |
| 0.15 | 4 | **0.020±0.002** | 0.022±0.001 | **0.020±0.001** | **0.020±0.001** | 0.091±0.038 | 0.567±0.324 | 2.869±0.430 | 1.185±0.427 | 1.176±0.426 |
| 0.2 | 1 | **0.050±0.012** | 0.053±0.019 | _0.051±0.018_ | _0.051±0.019_ | 0.236±0.117 | 0.546±0.167 | 2.894±0.312 | 2.288±0.139 | 2.468±0.139 |
| 0.2 | 2 | **0.037±0.003** | 0.039±0.006 | **0.037±0.005** | 0.038±0.006 | 0.250±0.109 | 0.826±0.410 | 2.982±0.595 | 1.610±0.041 | 1.565±0.042 |
| 0.2 | 3 | 0.032±0.006 | 0.032±0.004 | **0.030±0.003** | **0.030±0.004** | 0.157±0.045 | 0.872±0.368 | 3.330±0.284 | 1.180±1.161 | 1.166±1.170 |
| 0.2 | 4 | **0.030±0.002** | 0.034±0.002 | 0.032±0.002 | _0.031±0.002_ | 0.191±0.084 | 0.894±0.220 | 3.312±0.295 | 0.998±0.246 | 0.987±0.253 |
| 0.25 | 1 | **0.069±0.019** | 0.082±0.026 | _0.078±0.025_ | 0.081±0.028 | 0.389±0.185 | 1.028±0.224 | 3.513±0.350 | 2.318±0.140 | 2.468±0.139 |
| 0.25 | 2 | **0.054±0.005** | 0.059±0.010 | _0.056±0.009_ | 0.057±0.011 | 0.454±0.386 | 0.955±0.126 | 3.586±0.224 | 1.605±0.040 | 1.565±0.042 |
| 0.25 | 3 | 0.050±0.013 | 0.051±0.009 | **0.047±0.008** | _0.049±0.009_ | 0.341±0.105 | 0.942±0.049 | 3.469±0.164 | 1.475±1.071 | 1.460±1.084 |
| 0.25 | 4 | **0.047±0.005** | 0.053±0.003 | _0.050±0.004_ | _0.050±0.003_ | 0.359±0.176 | 1.179±0.442 | 3.238±0.333 | 1.181±0.425 | 1.176±0.426 |

Table 4: Average ANE values (plus/minus one standard deviation) for the real-world experiments on the PACM point cloud over ten runs. The best is marked in **bold red** while the second best is in underline blue.

| $\eta$ | option | GNNSync | Spectral | Spectral_RN | GPM | TranSync | CEMP_GCW | CEMP_MST | TAS | Trivial |
|---|---|---|---|---|---|---|---|---|---|---|
| 0 | 1 | 0.118±0.093 | **0.000±0.000** | **0.000±0.000** | **0.000±0.000** | **0.000±0.000** | **0.000±0.000** | **0.000±0.000** | 1.357±0.080 | 1.655±0.426 |
| 0 | 2 | 0.051±0.031 | **0.000±0.000** | **0.000±0.000** | **0.000±0.000** | **0.000±0.000** | **0.000±0.000** | **0.000±0.000** | 0.870±0.029 | 0.785±0.020 |
| 0 | 3 | 0.048±0.034 | **0.000±0.000** | **0.000±0.000** | **0.000±0.000** | **0.000±0.000** | **0.000±0.000** | **0.000±0.000** | 0.645±0.396 | 0.597±0.390 |
| 0 | 4 | 0.038±0.024 | **0.000±0.000** | **0.000±0.000** | **0.000±0.000** | **0.000±0.000** | **0.000±0.000** | **0.000±0.000** | 0.687±0.236 | 0.631±0.223 |
| 0.05 | 1 | 0.139±0.079 | 0.722±1.382 | 0.088±0.120 | **0.029±0.009** | 0.093±0.042 | 0.154±0.066 | 0.964±1.353 | 1.347±0.082 | 1.655±0.426 |
| 0.05 | 2 | 0.055±0.019 | 0.671±1.288 | **0.022±0.003** | 0.024±0.005 | 0.102±0.048 | 0.121±0.093 | 0.692±0.687 | 0.869±0.043 | 0.805±0.045 |
| 0.05 | 3 | 0.061±0.031 | 0.029±0.007 | **0.026±0.007** | 0.028±0.007 | 0.062±0.017 | 0.823±1.317 | 0.588±0.310 | 0.640±0.393 | 0.597±0.389 |
| 0.05 | 4 | 0.046±0.021 | 0.032±0.007 | **0.030±0.006** | 0.031±0.007 | 0.068±0.039 | 0.291±0.253 | 0.826±0.425 | 0.699±0.256 | 0.671±0.279 |
| 0.1 | 1 | 0.116±0.037 | **0.072±0.039** | 0.074±0.039 | 0.661±1.159 | 0.220±0.066 | 0.896±1.294 | 1.326±0.710 | 1.320±0.094 | 1.654±0.433 |
| 0.1 | 2 | 0.068±0.021 | 0.197±0.275 | **0.053±0.009** | 0.088±0.063 | 0.210±0.041 | 1.057±1.368 | 1.499±0.537 | 0.851±0.018 | 0.785±0.020 |
| 0.1 | 3 | 0.077±0.039 | 0.144±0.168 | 0.158±0.216 | **0.055±0.016** | 0.196±0.070 | 0.401±0.320 | 1.470±0.907 | 0.636±0.391 | 0.597±0.389 |
| 0.1 | 4 | 0.057±0.014 | 0.063±0.011 | **0.056±0.014** | 0.060±0.013 | 0.270±0.185 | 0.639±0.383 | 1.419±0.437 | 0.683±0.245 | 0.632±0.223 |
| 0.15 | 1 | 0.163±0.063 | 0.228±0.178 | 0.159±0.064 | **0.138±0.060** | 0.301±0.235 | 0.511±0.346 | 1.313±0.653 | 1.357±0.084 | 1.655±0.426 |
| 0.15 | 2 | 0.085±0.020 | 0.078±0.017 | **0.076±0.015** | 0.349±0.382 | 0.232±0.078 | 0.739±0.438 | 2.096±0.323 | 0.844±0.043 | 0.768±0.032 |
| 0.15 | 3 | 0.095±0.039 | 0.738±1.340 | **0.076±0.031** | 0.079±0.032 | 0.239±0.061 | 0.903±1.036 | 1.803±0.217 | 0.322±0.241 | 0.298±0.226 |
| 0.15 | 4 | **0.079±0.020** | 0.094±0.016 | 0.086±0.019 | 0.090±0.017 | 0.366±0.240 | 1.125±1.250 | 1.775±0.338 | 0.670±0.237 | 0.632±0.223 |
| 0.2 | 1 | **0.189±0.063** | 0.196±0.068 | 0.199±0.069 | 0.193±0.071 | 0.937±1.219 | 1.528±0.954 | 2.331±0.262 | 1.378±0.085 | 1.655±0.426 |
| 0.2 | 2 | **0.102±0.029** | 0.111±0.028 | 0.110±0.031 | 0.265±0.313 | 0.402±0.239 | 1.202±0.789 | 1.909±0.278 | 0.840±0.027 | 0.785±0.020 |
| 0.2 | 3 | **0.097±0.036** | 0.846±1.507 | 0.526±0.873 | 0.169±0.173 | 0.336±0.124 | 0.829±0.338 | 1.852±0.125 | 0.633±0.591 | 0.635±0.610 |
| 0.2 | 4 | **0.096±0.017** | 0.116±0.020 | 0.807±0.855 | 0.104±0.022 | 0.398±0.247 | 1.059±0.303 | 1.952±0.435 | 0.690±0.270 | 0.672±0.279 |
| 0.25 | 1 | 0.290±0.277 | 0.254±0.074 | 0.252±0.075 | **0.247±0.081** | 0.465±0.140 | 1.505±0.824 | 2.016±0.194 | 1.391±0.093 | 1.655±0.426 |
| 0.25 | 2 | **0.109±0.029** | 0.142±0.051 | 0.129±0.040 | 0.372±0.494 | 0.907±1.221 | 1.312±0.891 | 2.087±0.138 | 0.838±0.028 | 0.785±0.020 |
| 0.25 | 3 | **0.311±0.615** | 0.557±0.872 | 0.407±0.586 | 0.313±0.268 | 0.871±0.806 | 1.713±0.728 | 1.941±0.078 | 0.779±0.534 | 0.782±0.560 |
| 0.25 | 4 | **0.109±0.022** | 0.140±0.035 | 0.888±0.965 | 0.131±0.035 | 0.427±0.117 | 1.248±0.622 | 1.923±0.205 | 0.648±0.235 | 0.633±0.222 |

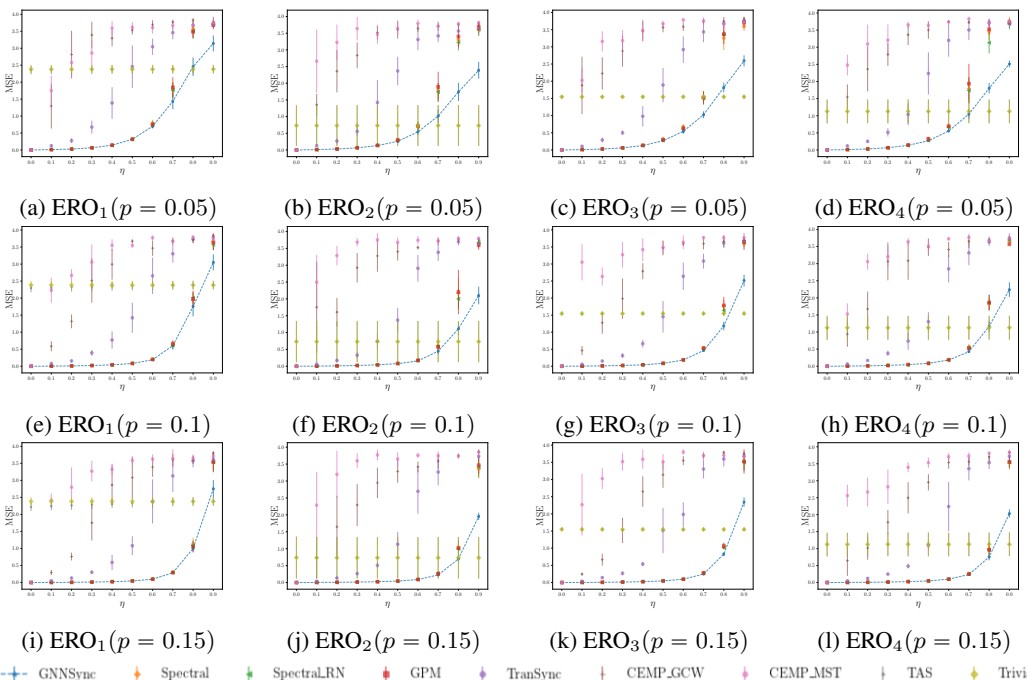

(a) $\mathrm{ERO}_1(p = 0.05)$  (b) $\mathrm{ERO}_2(p = 0.05)$  (c) $\mathrm{ERO}_3(p = 0.05)$  (d) $\mathrm{ERO}_4(p = 0.05)$

(e) $\mathrm{ERO}_1(p = 0.1)$  (f) $\mathrm{ERO}_2(p = 0.1)$  (g) $\mathrm{ERO}_3(p = 0.1)$  (h) $\mathrm{ERO}_4(p = 0.1)$

(i) $\mathrm{ERO}_1(p = 0.15)$  (j) $\mathrm{ERO}_2(p = 0.15)$  (k) $\mathrm{ERO}_3(p = 0.15)$  (l) $\mathrm{ERO}_4(p = 0.15)$

GNNSync   Spectral   SpectralRN   GPM   TranSync   CEMP_GCW   CEMP_MST   TAS   Trivial

Figure 6: MSE performance comparison on GNNSync against baselines on angular synchronization ($k = 1$) for ERO models. $p$ is the network density and $\eta$ is the noise level. Error bars indicate one standard deviation.

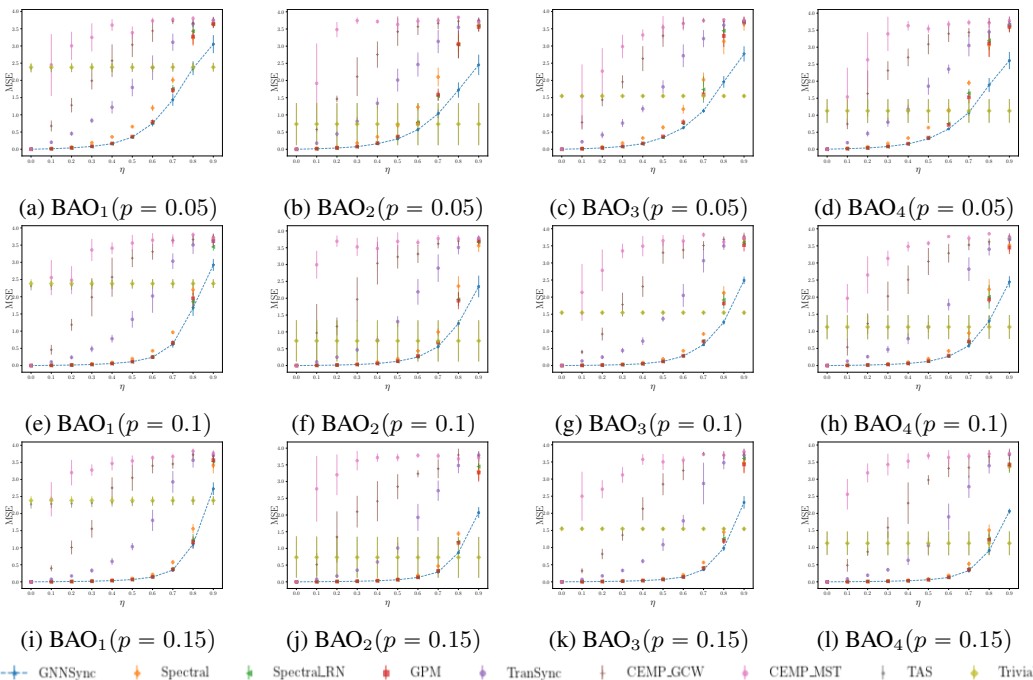

Figure 7: MSE performance comparison on GNNSync against baselines on angular synchronization ($k = 1$) for BAO models. $p$ is the network density and $\eta$ is the noise level. Error bars indicate one standard deviation.

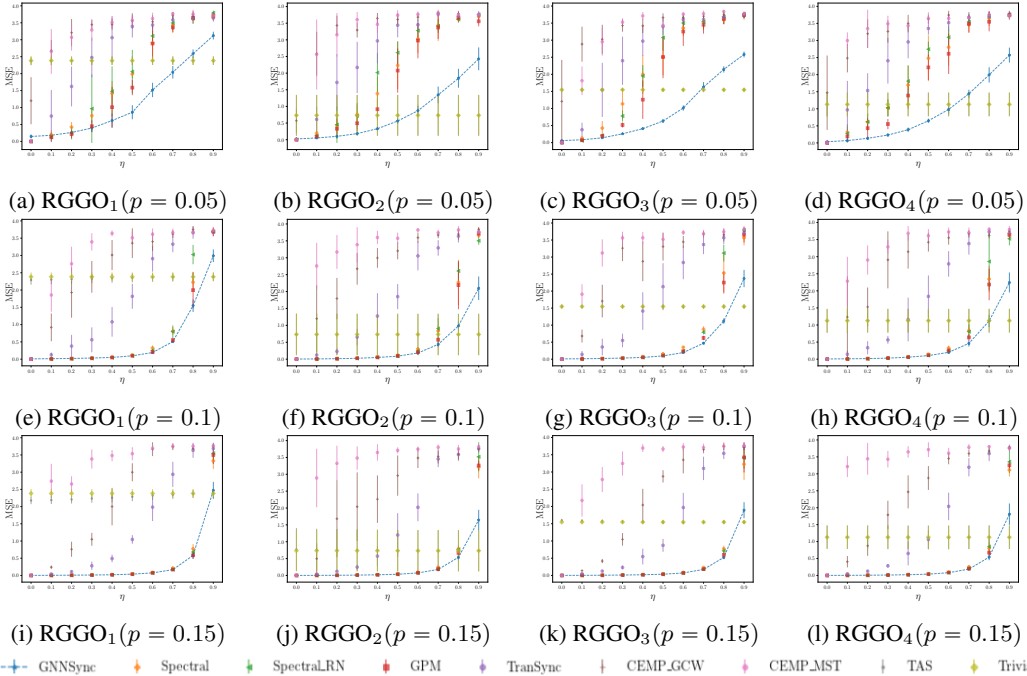

Figure 8: MSE performance comparison on GNNSync against baselines on angular synchronization ($k = 1$) for RGGO models. $p$ is the network density and $\eta$ is the noise level. Error bars indicate one standard deviation.

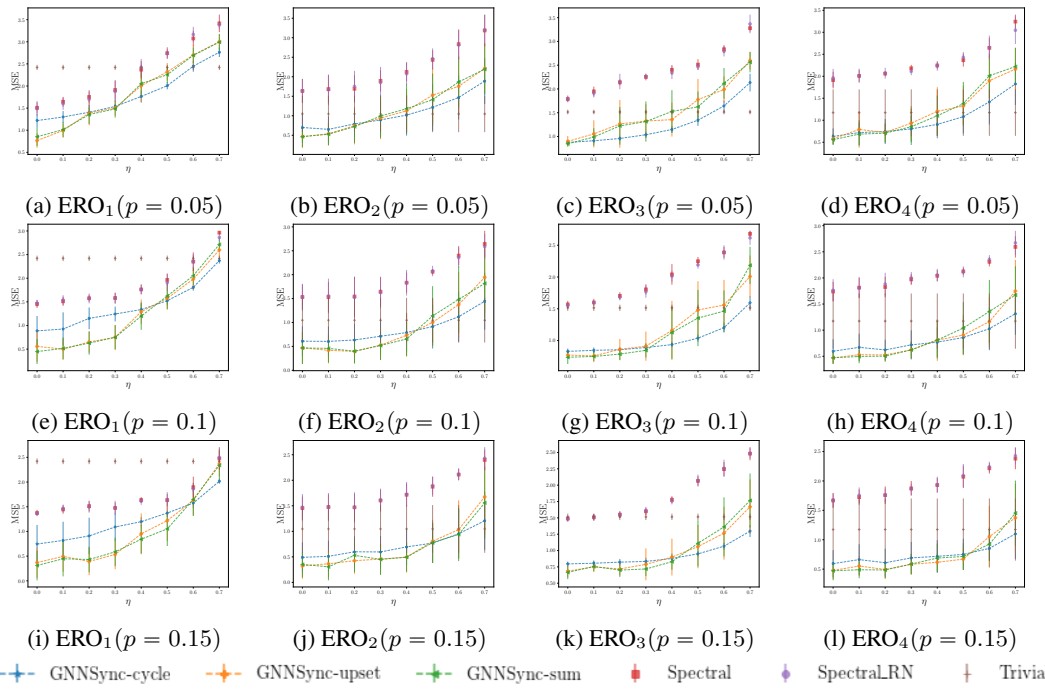

Figure 9: MSE performance comparison on GNNSync against baselines on $k-$synchronization with $k = 2$ for ERO models. $p$ is the network density and $\eta$ is the noise level. Error bars indicate one standard deviation.

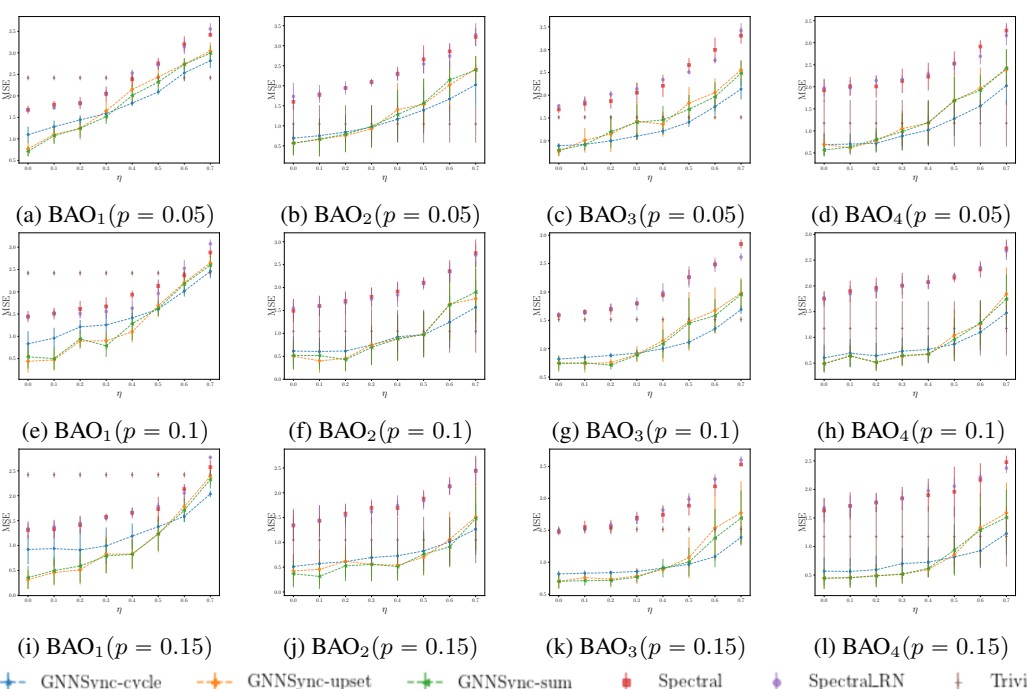

Figure 10: MSE performance comparison on GNNSync against baselines on $k-$synchronization with $k = 2$ for BAO models. $p$ is the network density and $\eta$ is the noise level. Error bars indicate one standard deviation.

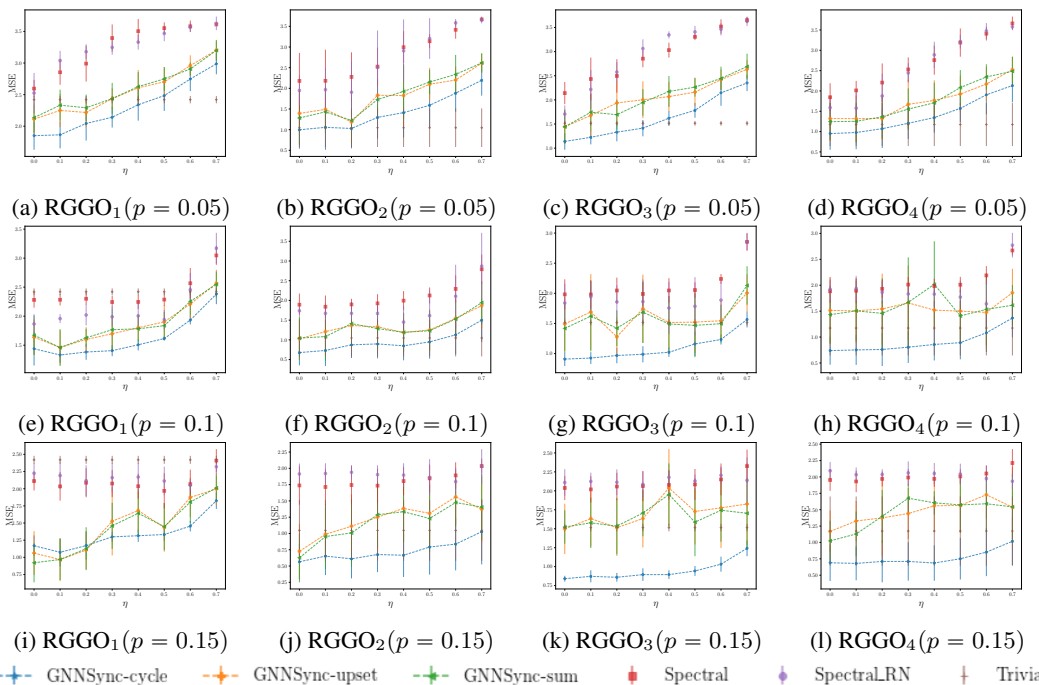

Figure 11: MSE performance comparison on GNNSync against baselines on $k-$synchronization with $k = 2$ for RGGO models. $p$ is the network density and $\eta$ is the noise level. Error bars indicate one standard deviation.

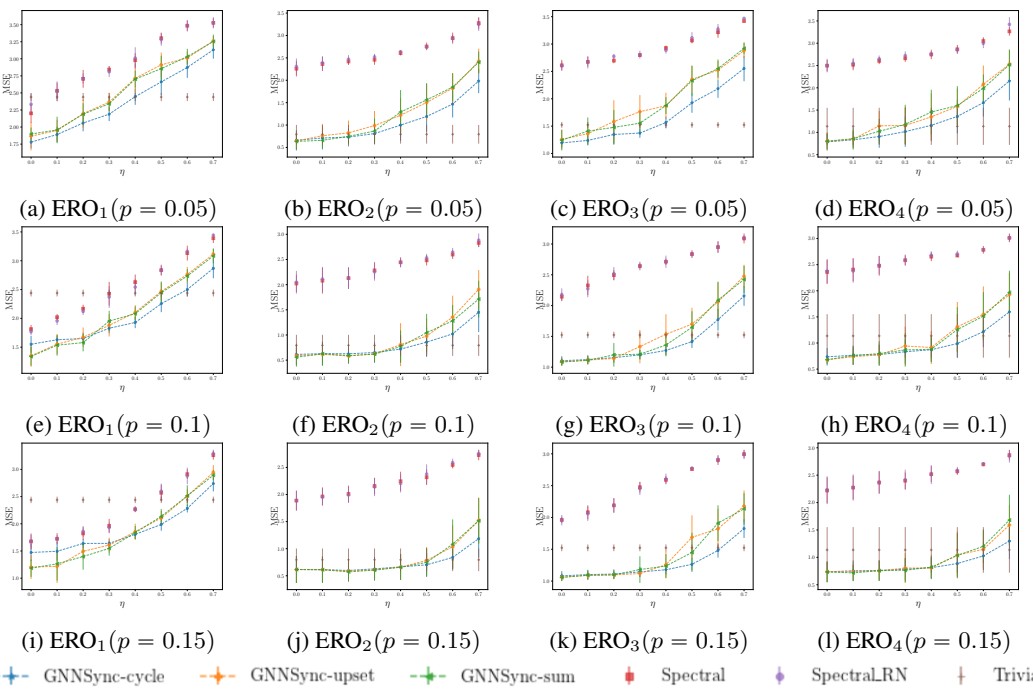

Figure 12: MSE performance comparison on GNNSync against baselines on $k-$synchronization with $k = 3$ for ERO models. $p$ is the network density and $\eta$ is the noise level. $p$ is the network density and $\eta$ is the noise level. Error bars indicate one standard deviation.

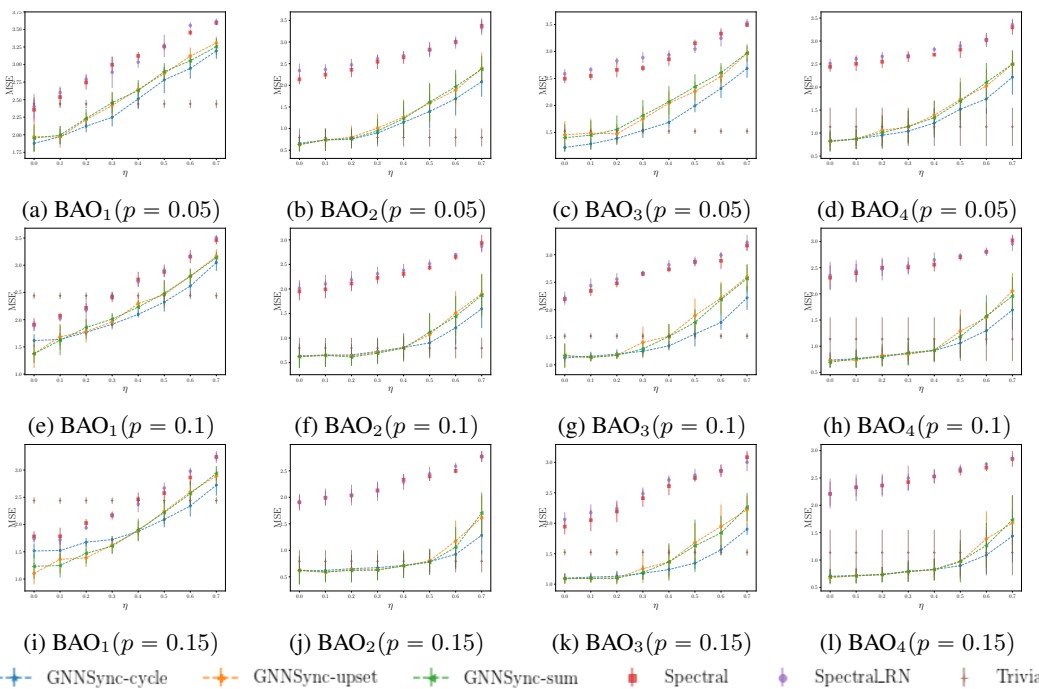

Figure 13: MSE performance comparison on GNNSync against baselines on $k-$synchronization with $k = 3$ for BAO models. $p$ is the network density and $\eta$ is the noise level. Error bars indicate one standard deviation.

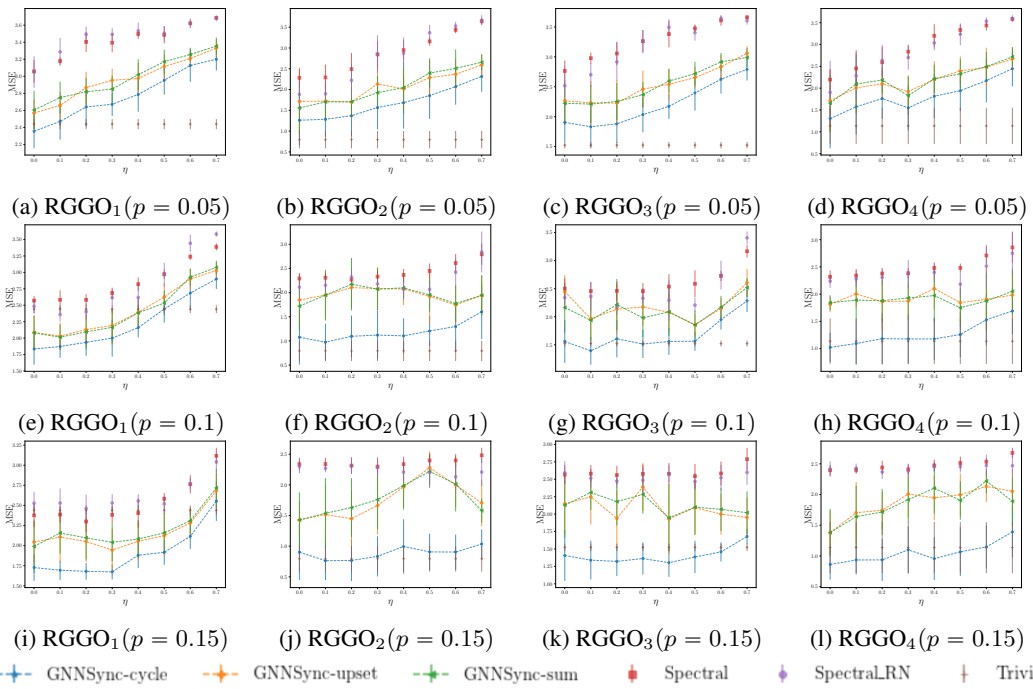

Figure 14: MSE performance comparison on GNNSync against baselines on $k-$synchronization with $k = 3$ for RGGO models. $p$ is the network density and $\eta$ is the noise level. Error bars indicate one standard deviation.

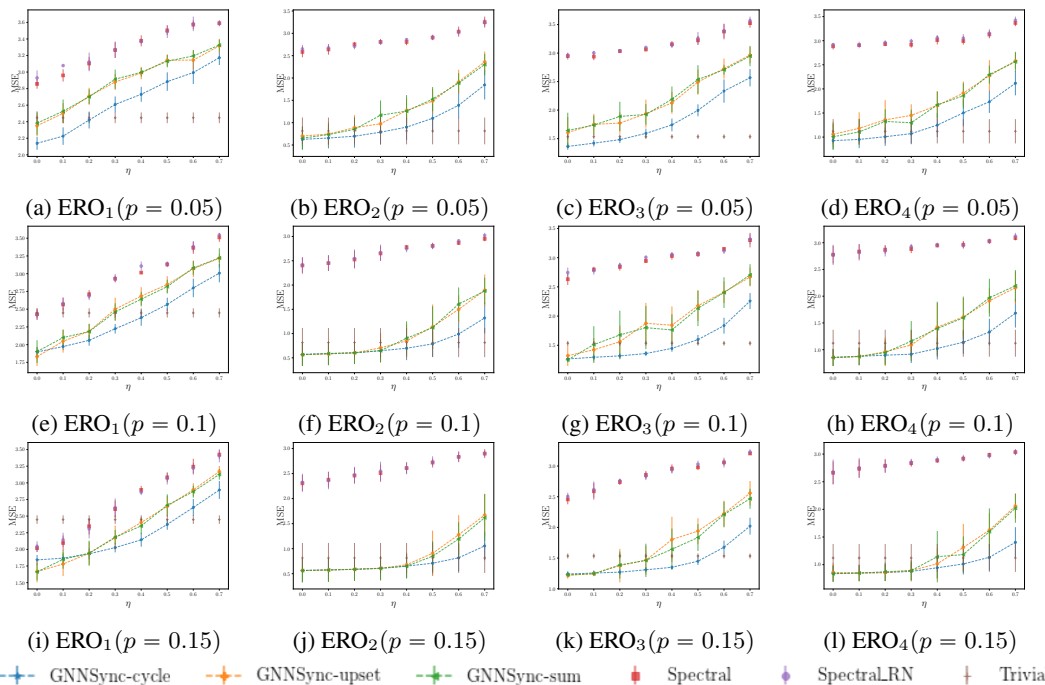

Figure 15: MSE performance comparison on GNNSync against baselines on $k-$synchronization with $k = 4$ for ERO models. $p$ is the network density and $\eta$ is the noise level. Error bars indicate one standard deviation.

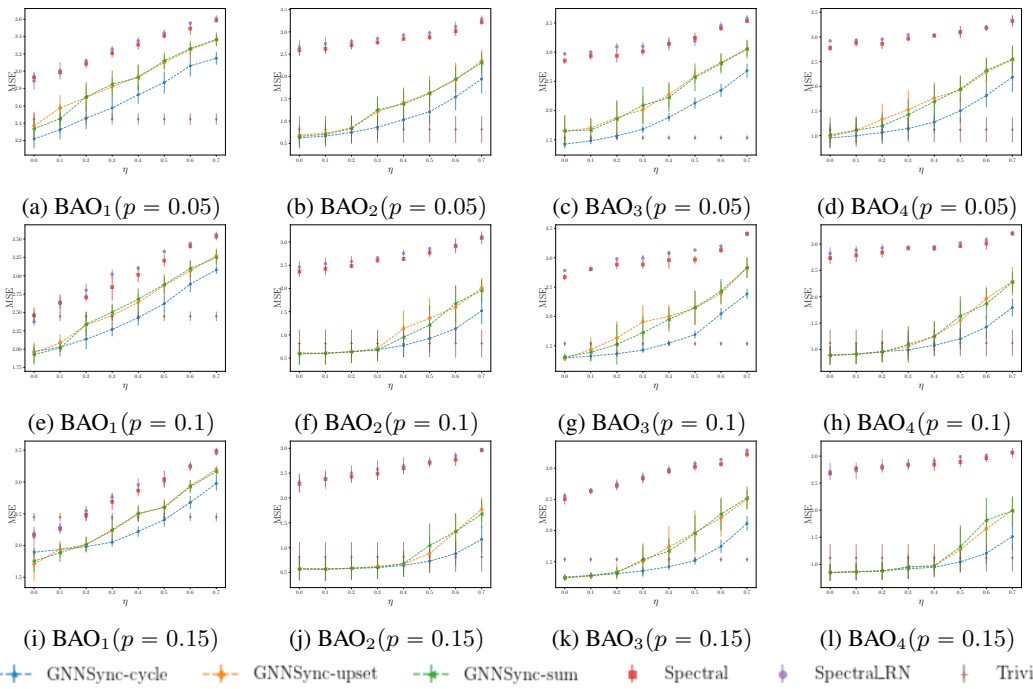

Figure 16: MSE performance comparison on GNNSync against baselines on $k-$synchronization with $k = 4$ for BAO models. $p$ is the network density and $\eta$ is the noise level. Error bars indicate one standard deviation.

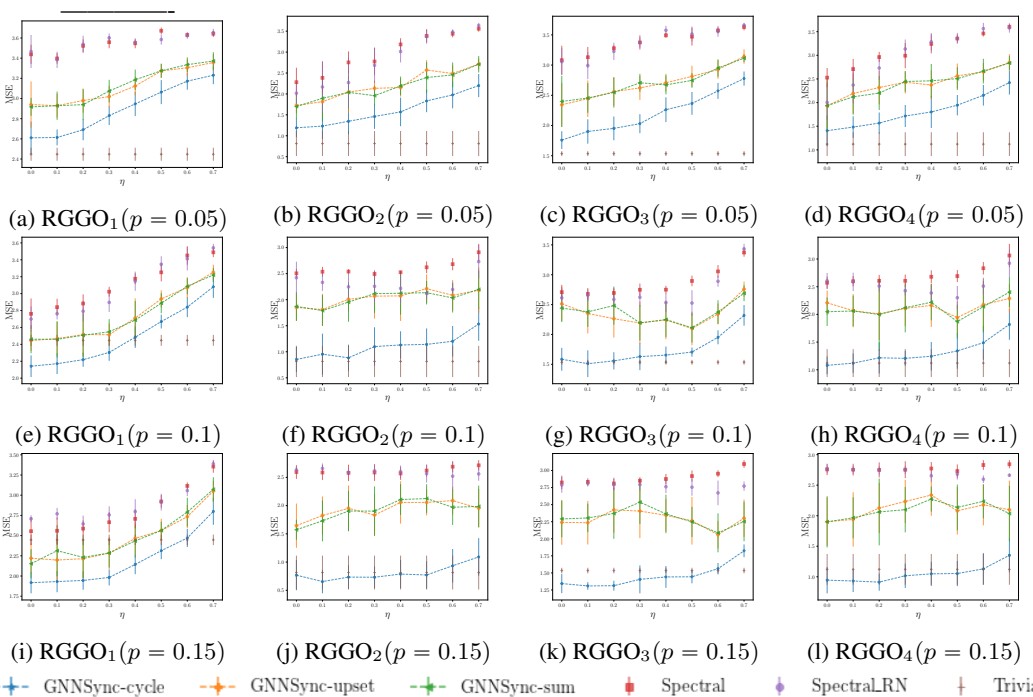

Figure 17: MSE performance comparison on GNNSync against baselines on $k-$synchronization with $k = 4$ for RGGO models. $p$ is the network density and $\eta$ is the noise level. Error bars indicate one standard deviation.

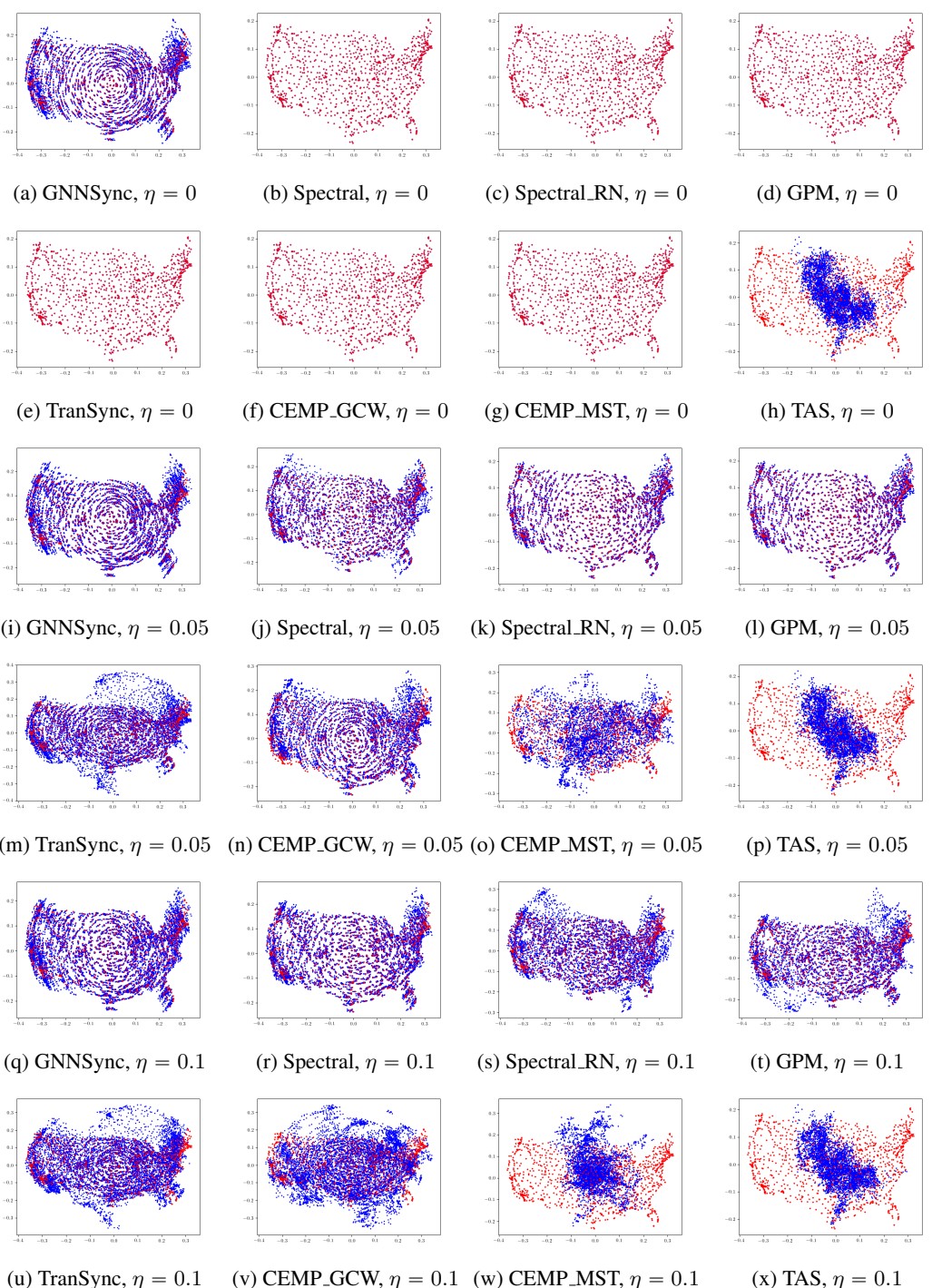

Figure 18: Result visualization for the Sensor Network Localization task on the U.S. map using option "1" as ground-truth angles for low-noise input data. Red dots indicate ground-truth locations and blue dots are estimated city locations.

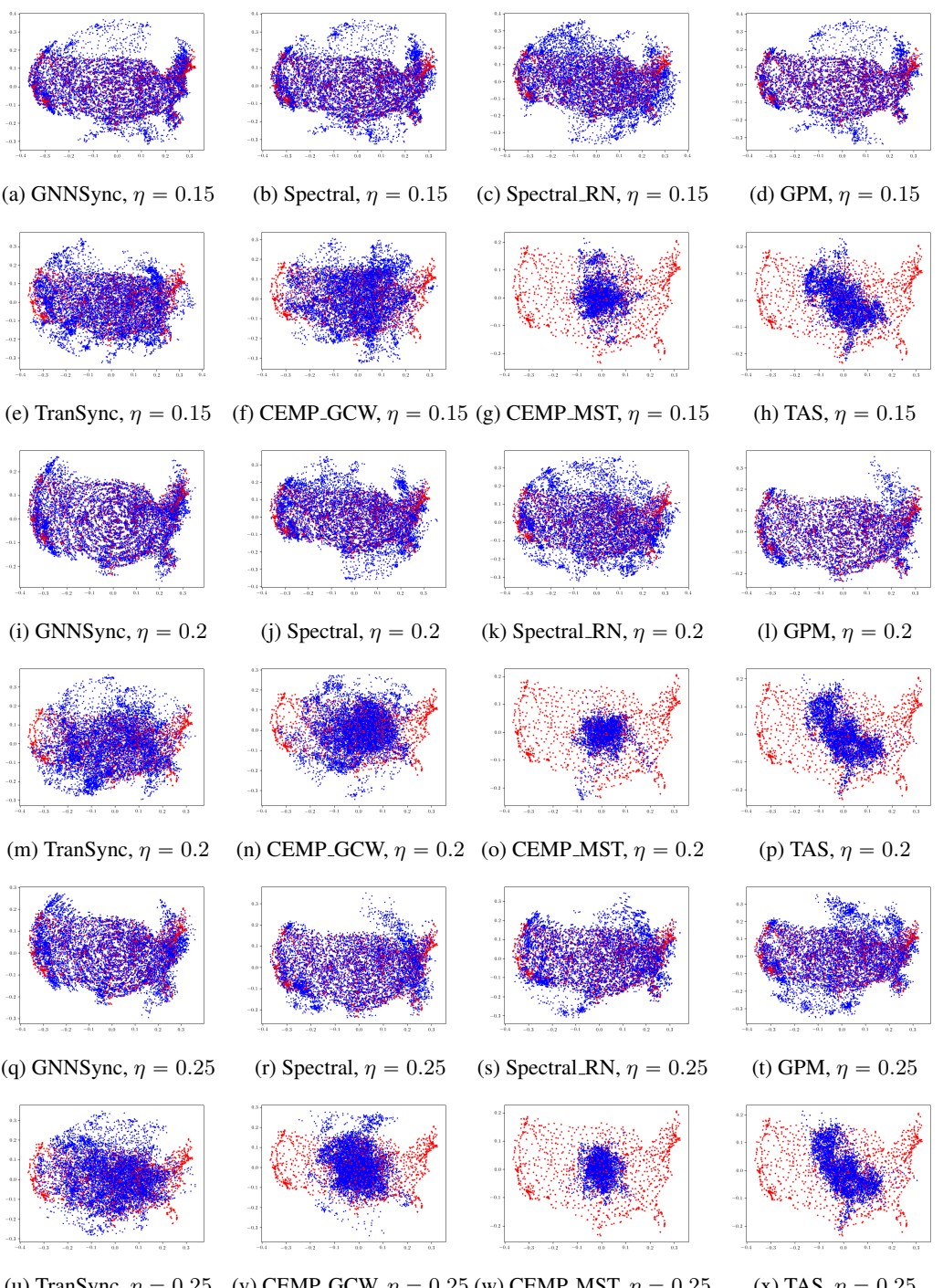

Figure 19: Result visualization for the Sensor Network Localization task on the U.S. map using option "1" as ground-truth angles for high-noise input data. Red dots indicate ground-truth locations and blue dots are estimated city locations.

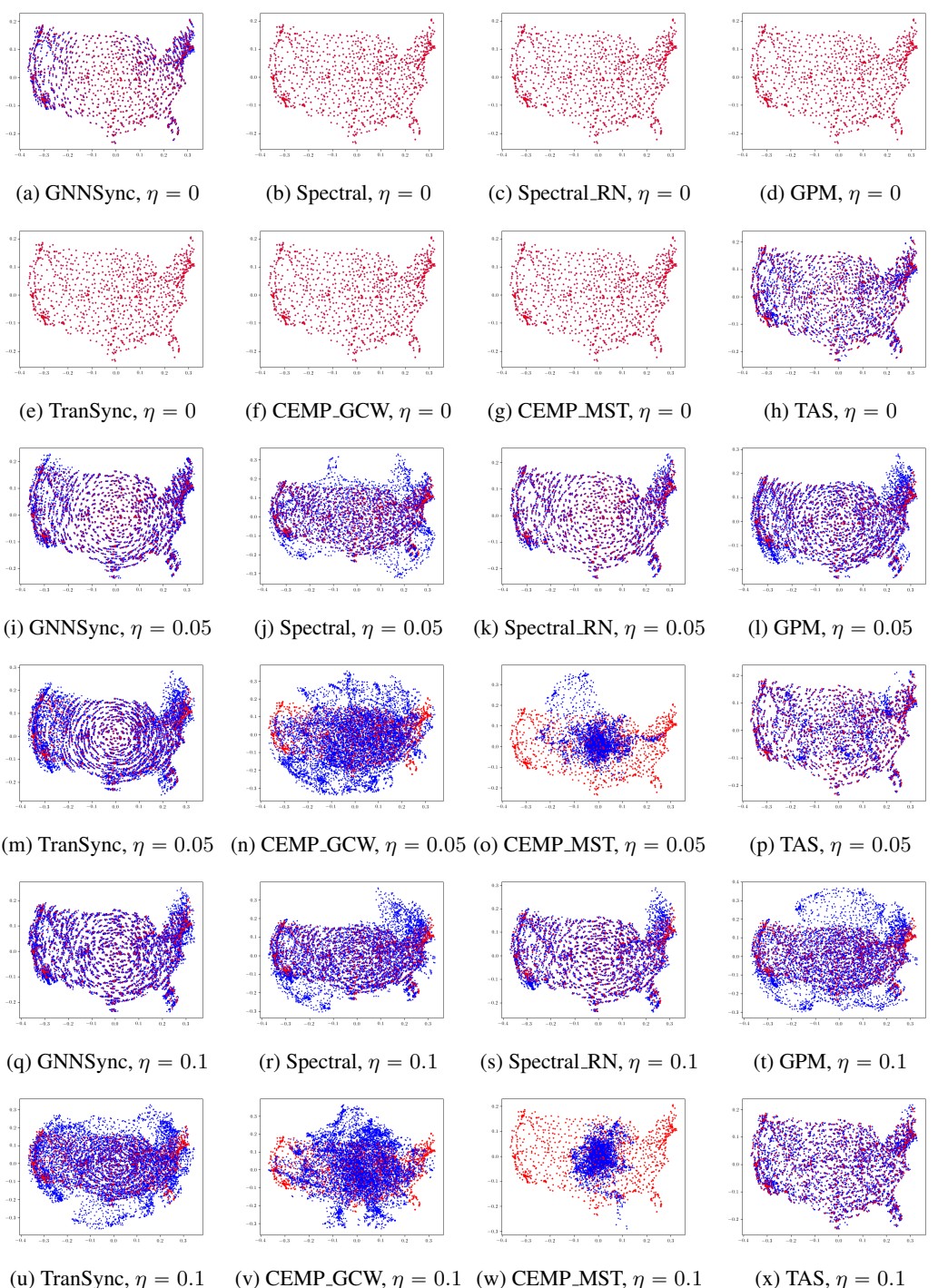

Figure 20: Result visualization for the Sensor Network Localization task on the U.S. map using option "2" as ground-truth angles for low-noise input data. Red dots indicate ground-truth locations and blue dots are estimated city locations.

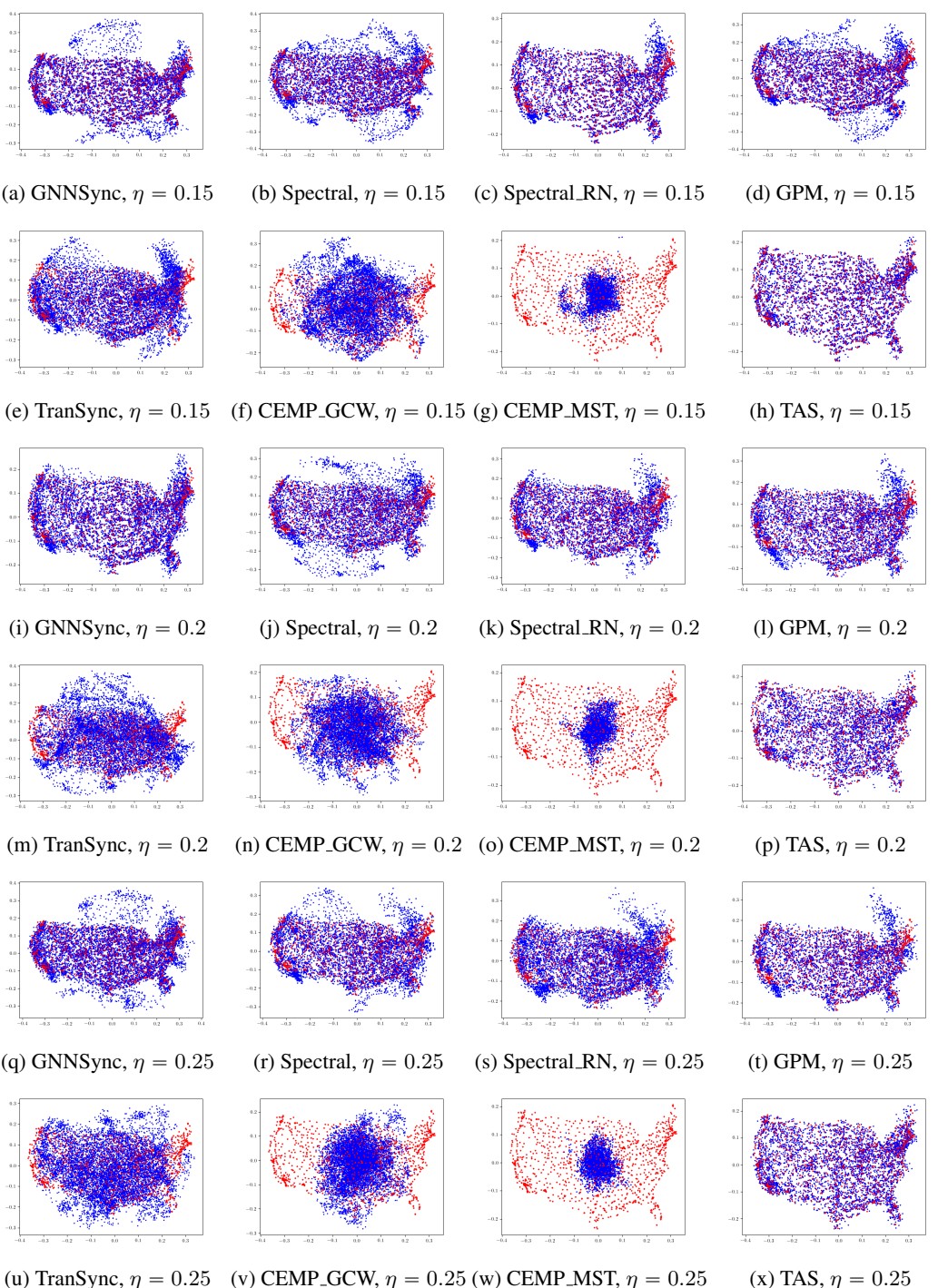

(a) GNNSync, $\eta = 0.15$    (b) Spectral, $\eta = 0.15$    (c) Spectral_RN, $\eta = 0.15$    (d) GPM, $\eta = 0.15$

(e) TranSync, $\eta = 0.15$    (f) CEMP_GCW, $\eta = 0.15$   (g) CEMP_MST, $\eta = 0.15$    (h) TAS, $\eta = 0.15$

(i) GNNSync, $\eta = 0.2$    (j) Spectral, $\eta = 0.2$    (k) Spectral_RN, $\eta = 0.2$    (l) GPM, $\eta = 0.2$

(m) TranSync, $\eta = 0.2$    (n) CEMP_GCW, $\eta = 0.2$   (o) CEMP_MST, $\eta = 0.2$    (p) TAS, $\eta = 0.2$

(q) GNNSync, $\eta = 0.25$    (r) Spectral, $\eta = 0.25$    (s) Spectral_RN, $\eta = 0.25$    (t) GPM, $\eta = 0.25$

(u) TranSync, $\eta = 0.25$   (v) CEMP_GCW, $\eta = 0.25$   (w) CEMP_MST, $\eta = 0.25$    (x) TAS, $\eta = 0.25$

Figure 21: Result visualization for the Sensor Network Localization task on the U.S. map using option "2" as ground-truth angles for high-noise input data. Red dots indicate ground-truth locations and blue dots are estimated city locations.

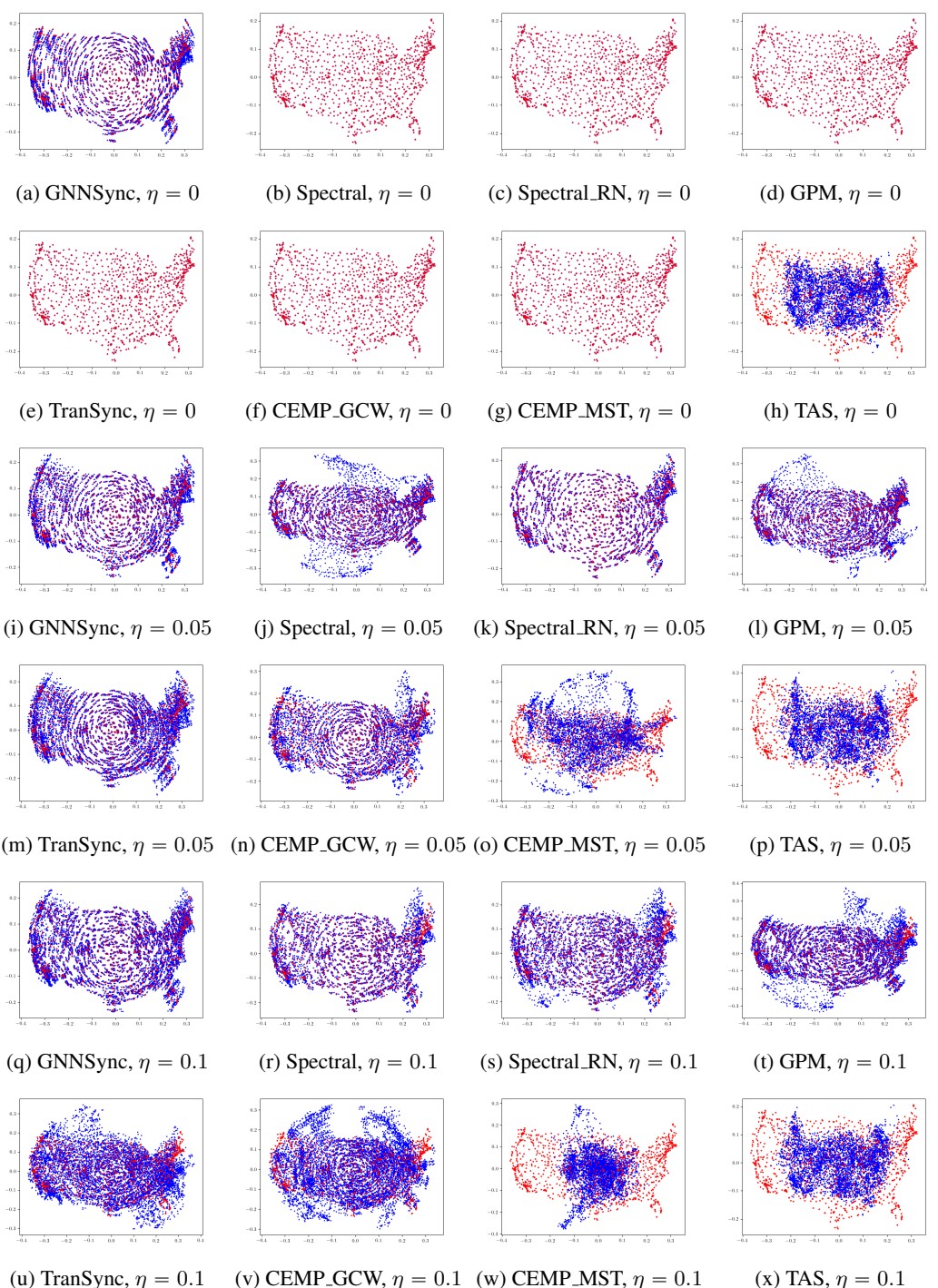

Figure 22: Result visualization for the Sensor Network Localization task on the U.S. map using option "3" as ground-truth angles for low-noise input data. Red dots indicate ground-truth locations and blue dots are estimated city locations.

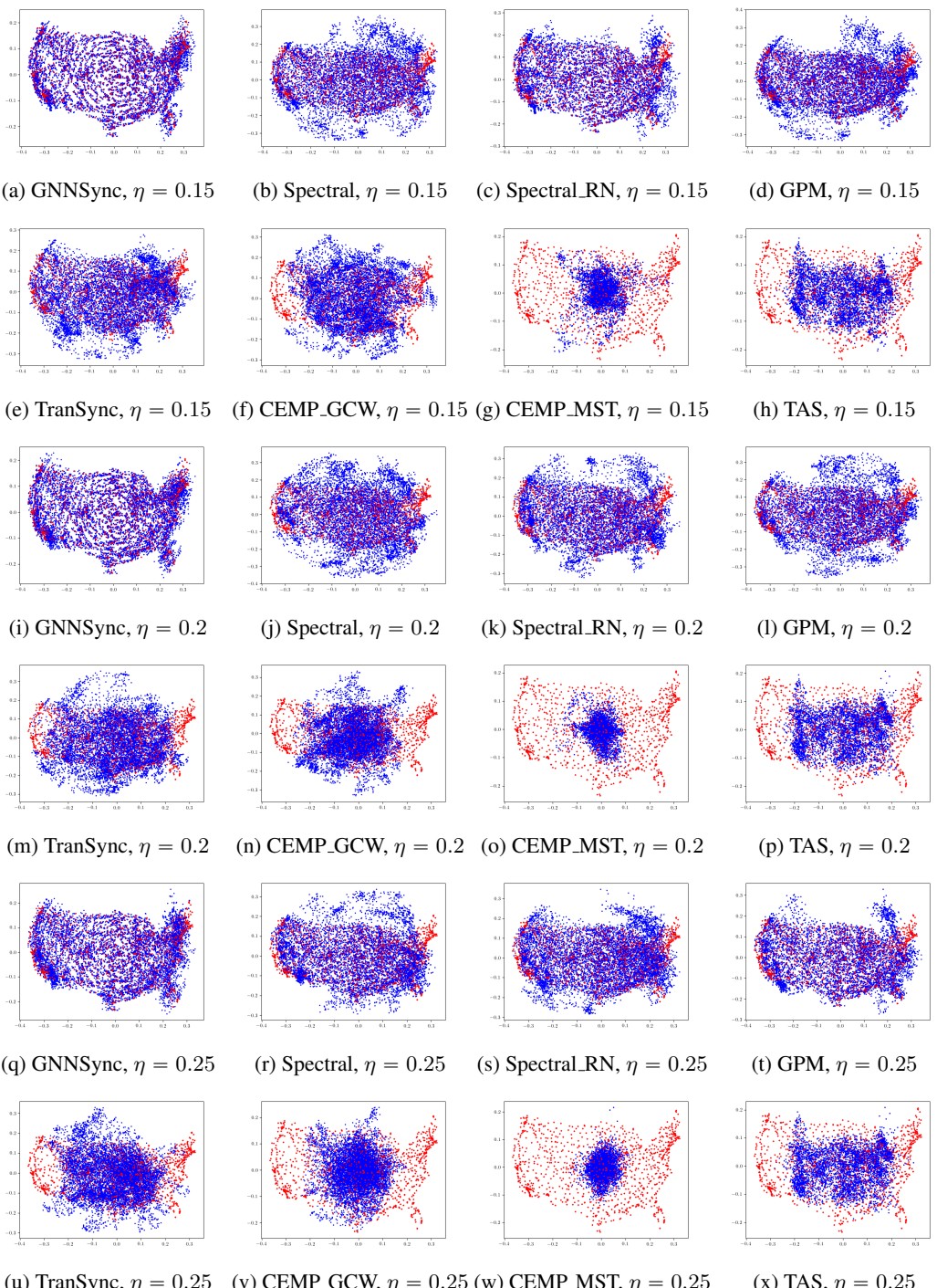

Figure 23: Result visualization for the Sensor Network Localization task on the U.S. map using option "3" as ground-truth angles for high-noise input data. Red dots indicate ground-truth locations and blue dots are estimated city locations.

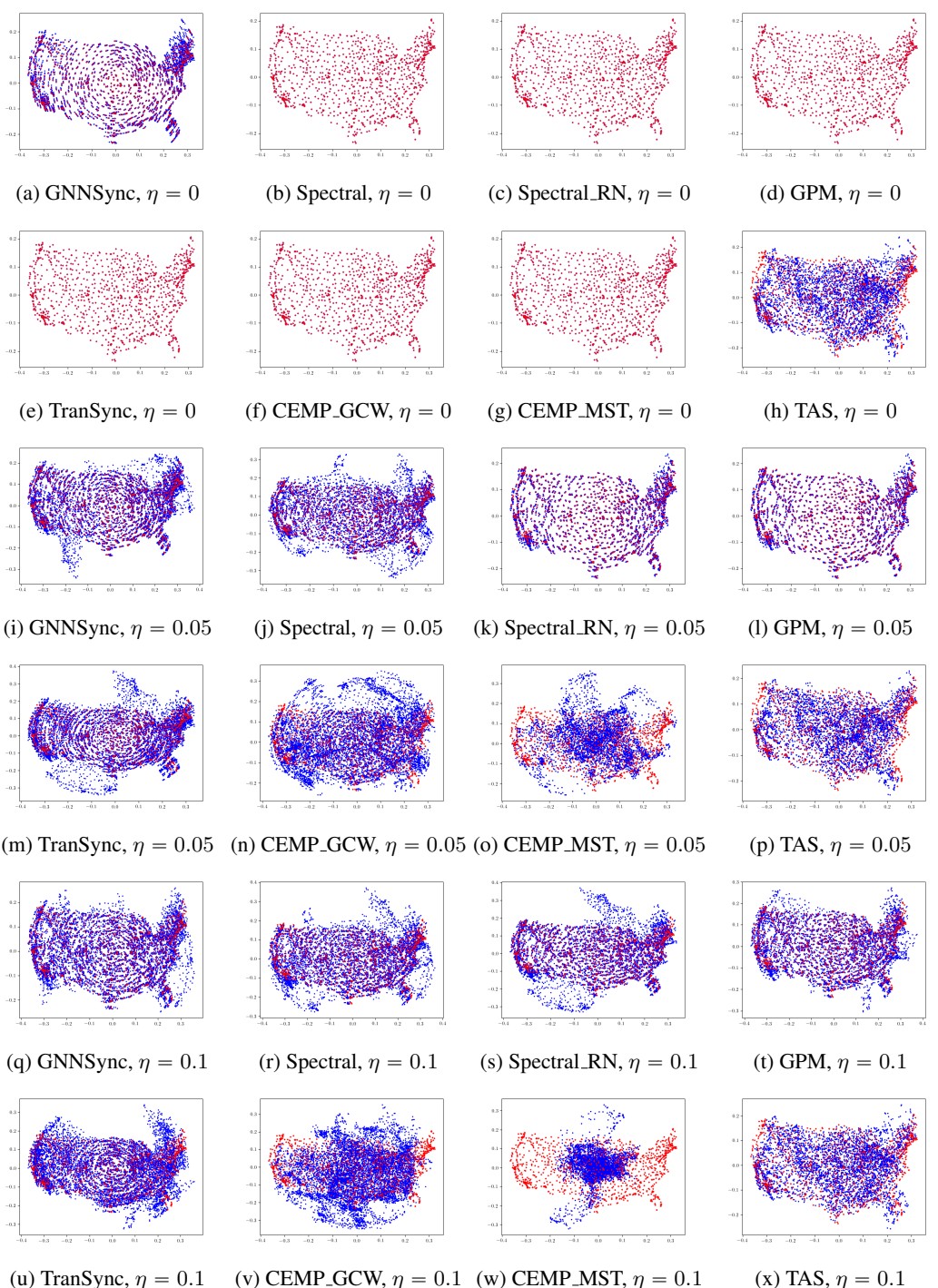

Figure 24: Result visualization for the Sensor Network Localization task on the U.S. map using option "4" as ground-truth angles for low-noise input data. Red dots indicate ground-truth locations and blue dots are estimated city locations.

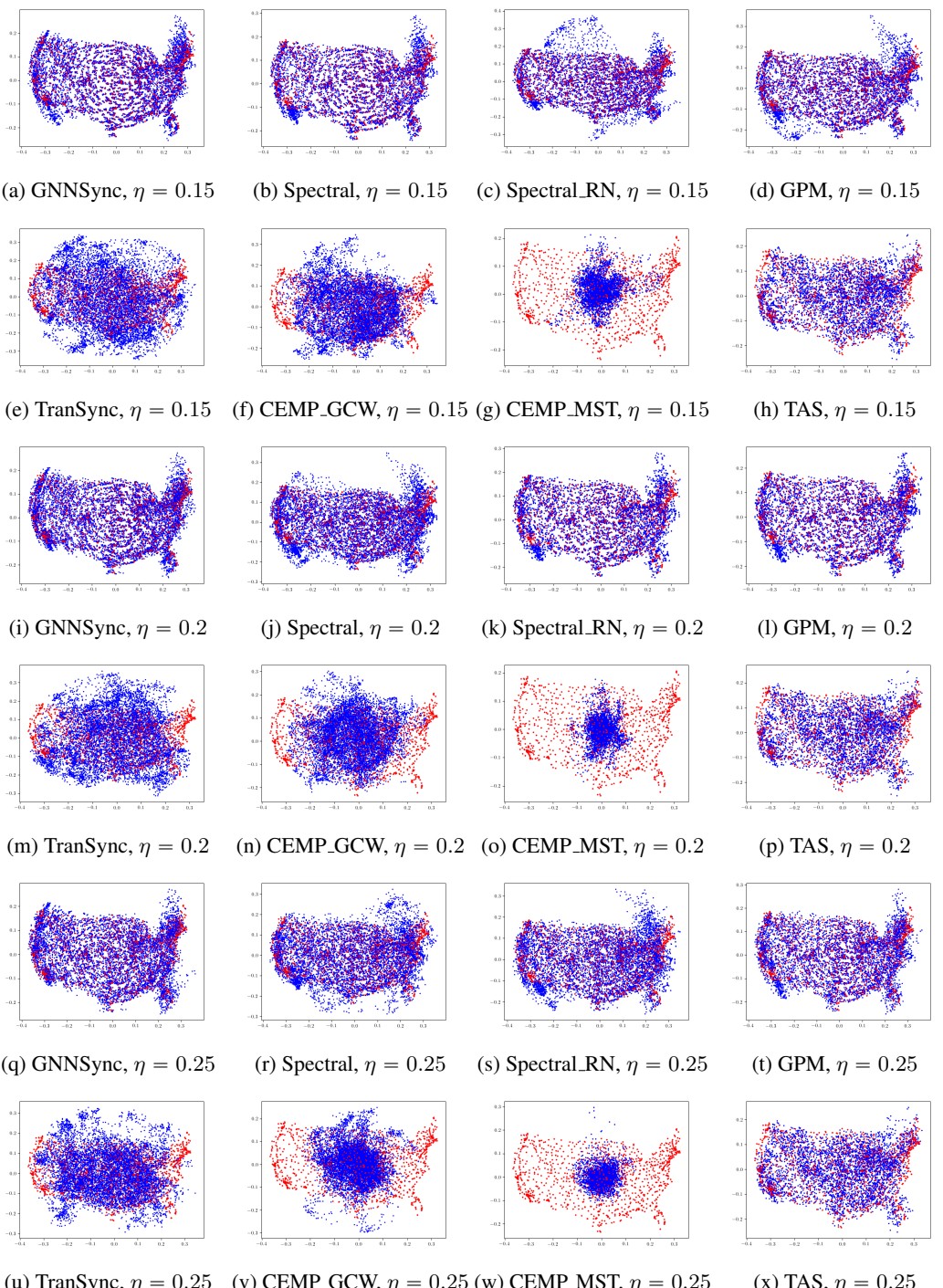

(a) GNNSync, $\eta = 0.15$    (b) Spectral, $\eta = 0.15$    (c) Spectral_RN, $\eta = 0.15$    (d) GPM, $\eta = 0.15$

(e) TranSync, $\eta = 0.15$    (f) CEMP_GCW, $\eta = 0.15$    (g) CEMP_MST, $\eta = 0.15$    (h) TAS, $\eta = 0.15$

(i) GNNSync, $\eta = 0.2$    (j) Spectral, $\eta = 0.2$    (k) Spectral_RN, $\eta = 0.2$    (l) GPM, $\eta = 0.2$

(m) TranSync, $\eta = 0.2$    (n) CEMP_GCW, $\eta = 0.2$    (o) CEMP_MST, $\eta = 0.2$    (p) TAS, $\eta = 0.2$

(q) GNNSync, $\eta = 0.25$    (r) Spectral, $\eta = 0.25$    (s) Spectral_RN, $\eta = 0.25$    (t) GPM, $\eta = 0.25$

(u) TranSync, $\eta = 0.25$    (v) CEMP_GCW, $\eta = 0.25$    (w) CEMP_MST, $\eta = 0.25$    (x) TAS, $\eta = 0.25$

Figure 25: Result visualization for the Sensor Network Localization task on the U.S. map using option "4" as ground-truth angles for high-noise input data. Red dots indicate ground-truth locations and blue dots are estimated city locations.

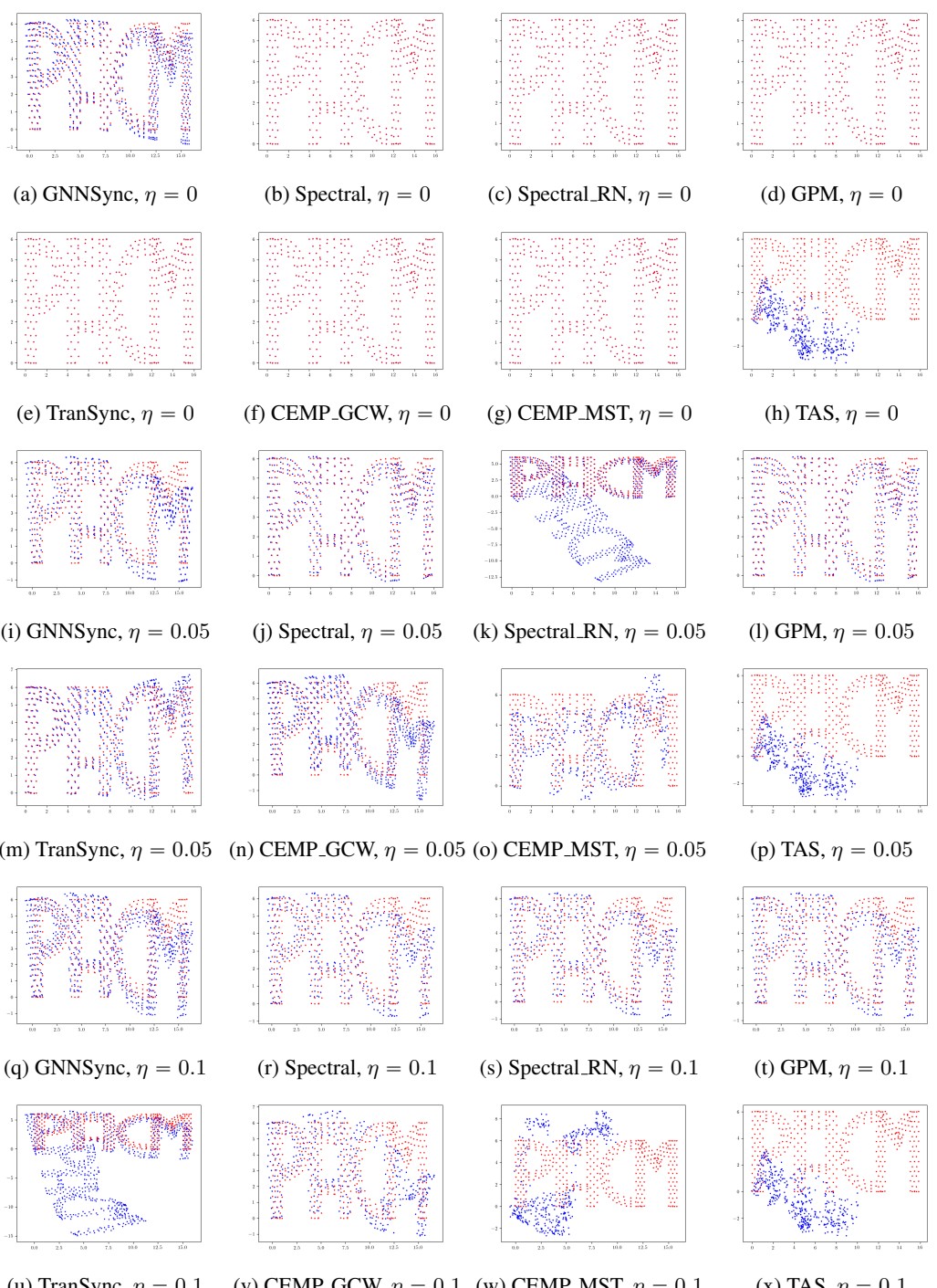

Figure 26: Result visualization for the Sensor Network Localization task on the PACM point cloud using option "1" as ground-truth angles for low-noise input data. Red dots indicate ground-truth locations and blue dots are estimated city locations.

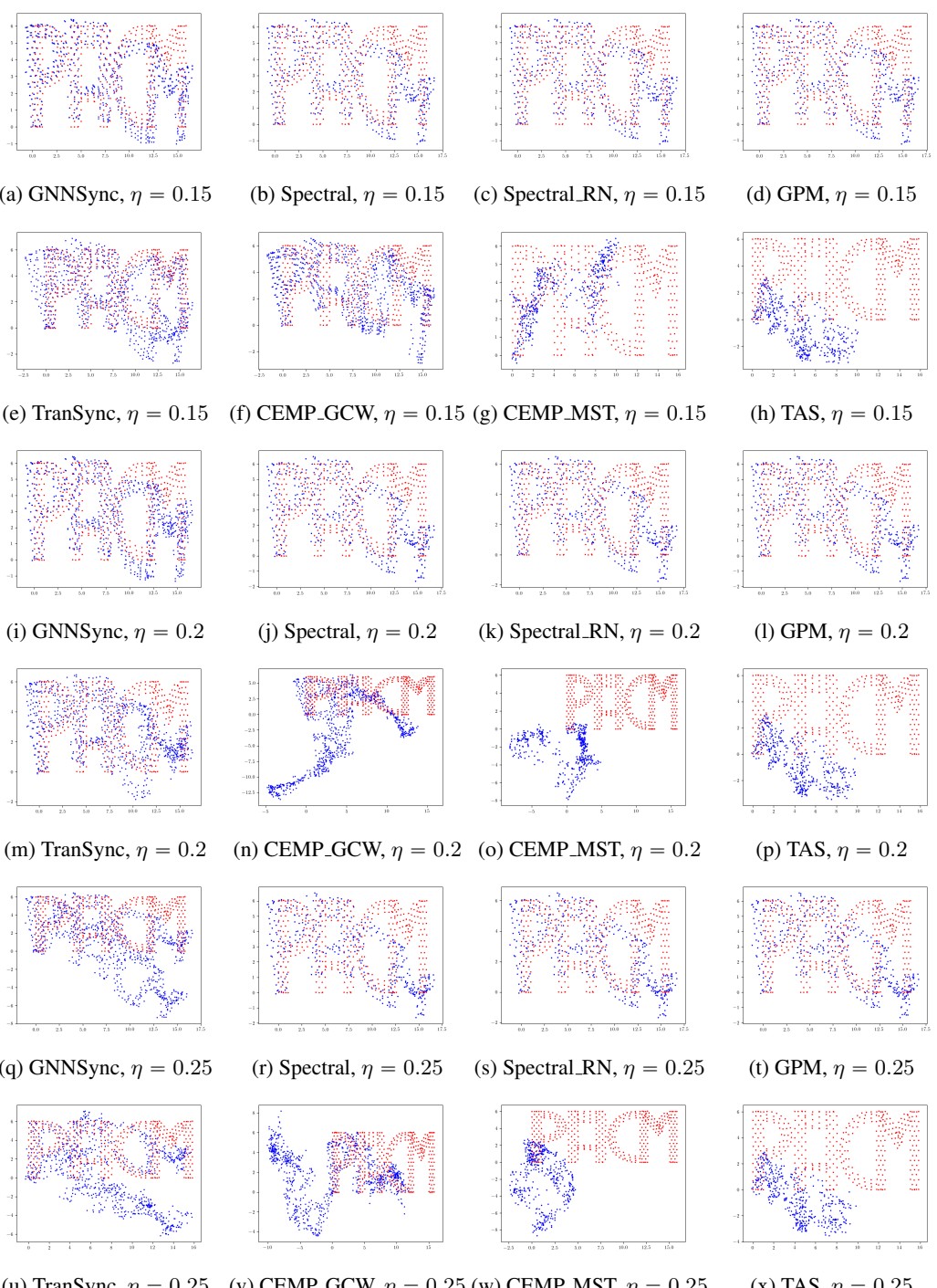

Figure 27: Result visualization for the Sensor Network Localization task on the PACM point cloud using option "1" as ground-truth angles for high-noise input data. Red dots indicate ground-truth locations and blue dots are estimated city locations.

### D.2 EXTENDED ABLATION STUDY RESULTS

Ablation study results are reported in Fig. 28 and 29, while the rest are omitted but could lead to the same conclusion. Note that for $k > 1$, we ablation study results are based on using $\mathcal{L}_{\text{cycle}}$ as the training loss function.

Improvements over all possible baselines when taking their output as input features for $k = 1$ are reported in Fig. 30, 31 and 32, where we omit results for $\eta = 0.9$ as in general all methods fall behind the trivial solution at $\eta = 0.9$. We find that in most cases GNNSync could improve over baselines, and could do worse often only when all methods fall behind the trivial baseline.

To show the effect of a linear combination of $L_{\text{cycle}}$ and $L_{\text{upset}}$, we empirically test $L_{\text{cycle}} + \tau L_{\text{upset}}$, with $\tau$ varying from 0 to 0.9; see Fig. 33 (the others are omitted but could lead to the same conclusion) for details. The performance for different choices of $\tau$ do not vary significantly, providing further evidence that it suffices to simply pay attention to either of the two loss functions instead of their linear combination. The experiments also show that as the problem becomes harder (e.g. as the noise level increases and the network becomes sparser), a smaller coefficient of $L_{\text{upset}}$ (even zero) is preferred, which indicates that $L_{\text{cycle}}$ plays a more essential role in the more challenging scenarios.

To assess the effect of fine-tuning (via projected gradient steps) over the baselines, we apply the same number of projected gradient descent steps as GNNSync to the comparative baselines and report the performance in Figures 34 and 35. We observe that even when applying these fine-tuning steps, the baselines are usually beaten by our end-to-end trainable GNNSync pipeline.

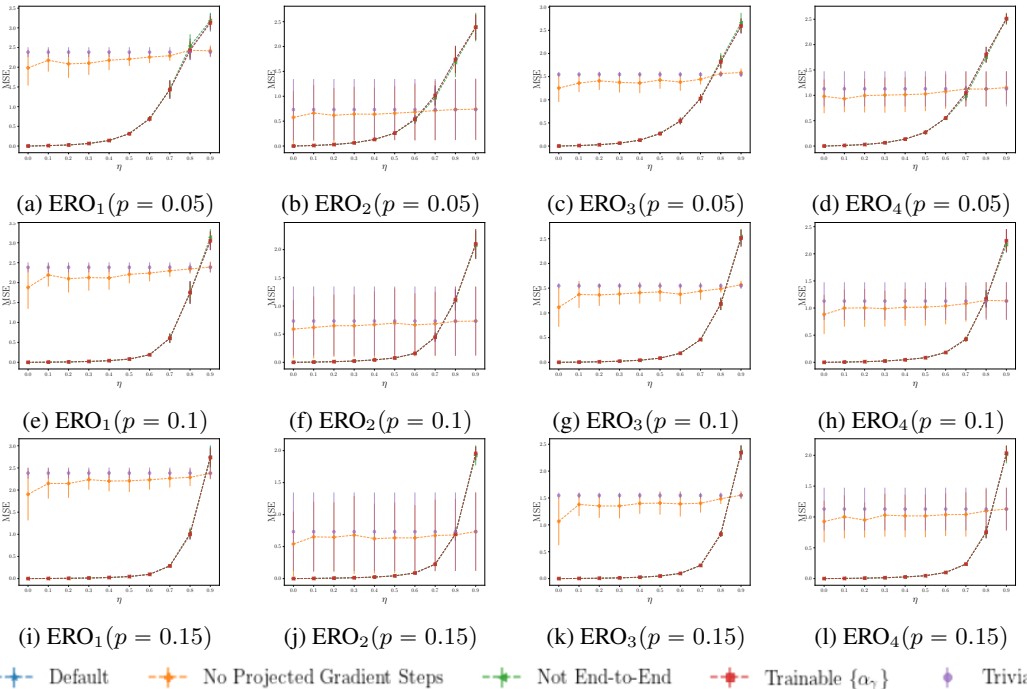

Figure 28: MSE performance comparison on GNNSync variants on angular synchronization ($k = 1$) for ERO models. $p$ is the network density and $\eta$ is the noise level. Error bars indicate one standard deviation.

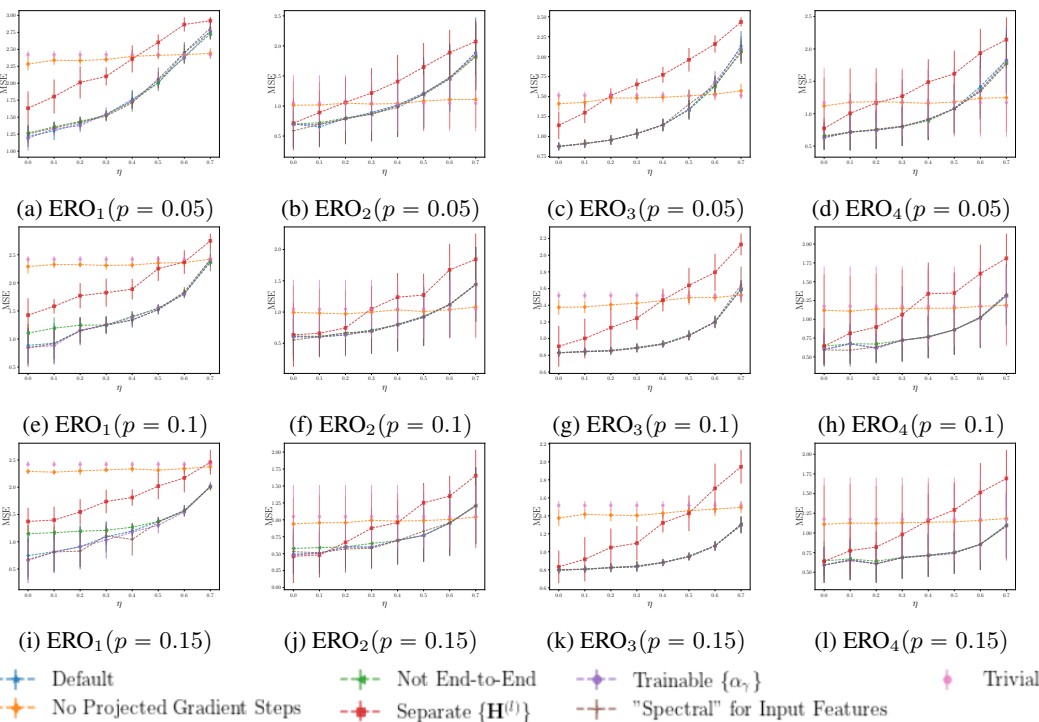

Figure 29: MSE performance comparison on GNNSync variants on $k-$synchronization with $k = 2$ for ERO models. $p$ is the network density and $\eta$ is the noise level. Error bars indicate one standard deviation.

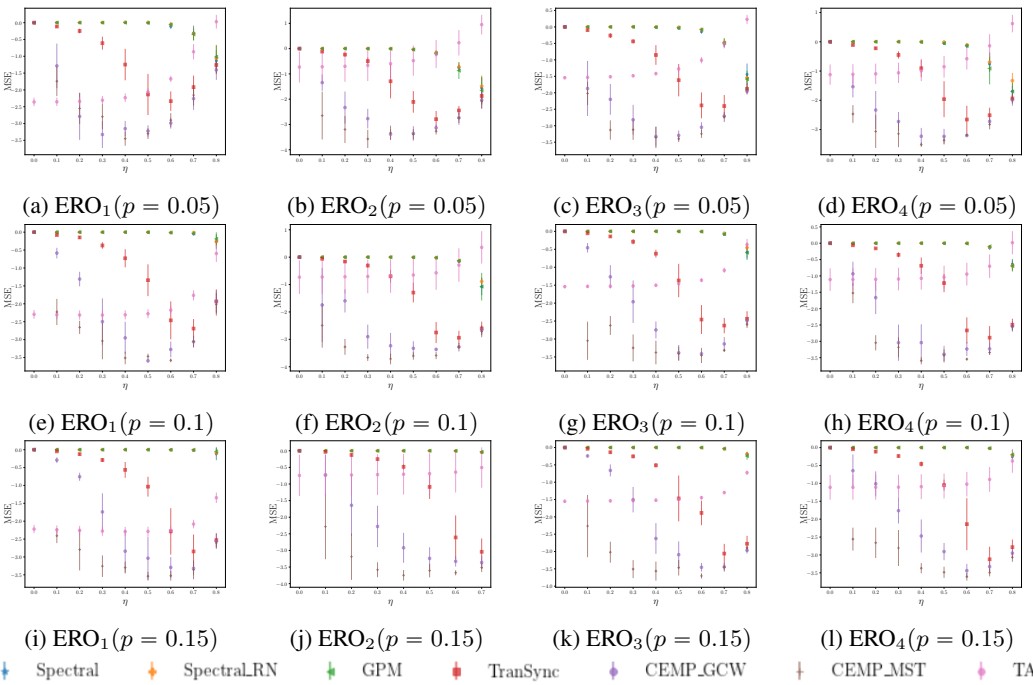

Figure 30: MSE performance improvement on GNNSync over variants on angular synchronization ($k = 1$) for ERO models. $p$ is the network density and $\eta$ is the noise level. Error bars indicate one standard deviation.

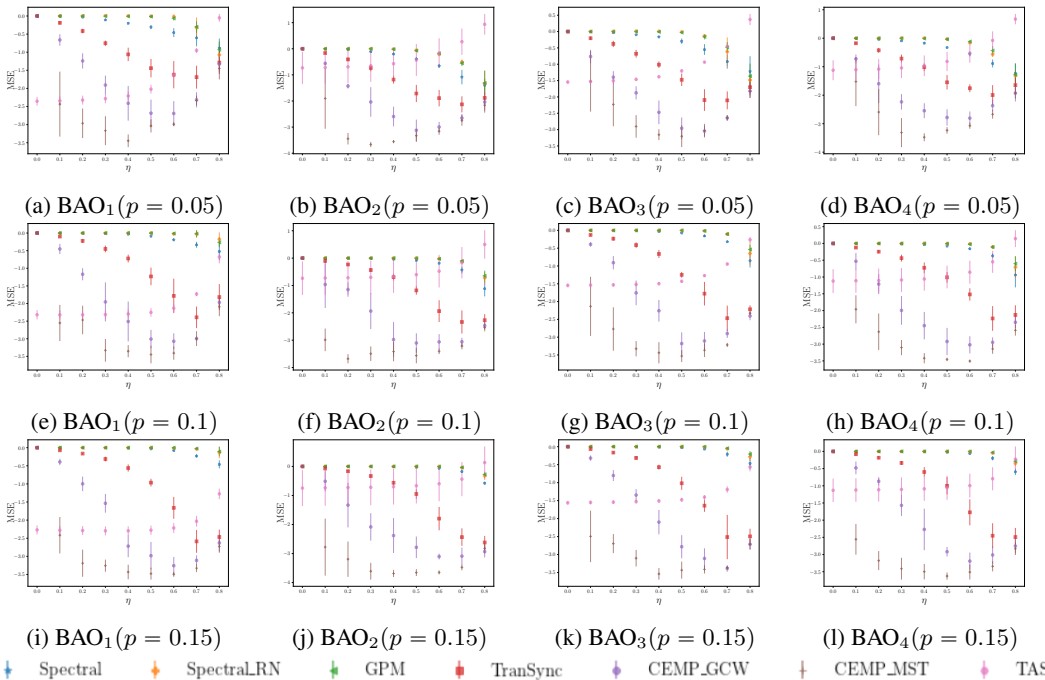

Figure 31: MSE performance improvement on GNNSync over variants on angular synchronization ($k = 1$) for BAO models. $p$ is the network density and $\eta$ is the noise level. Error bars indicate one standard deviation.

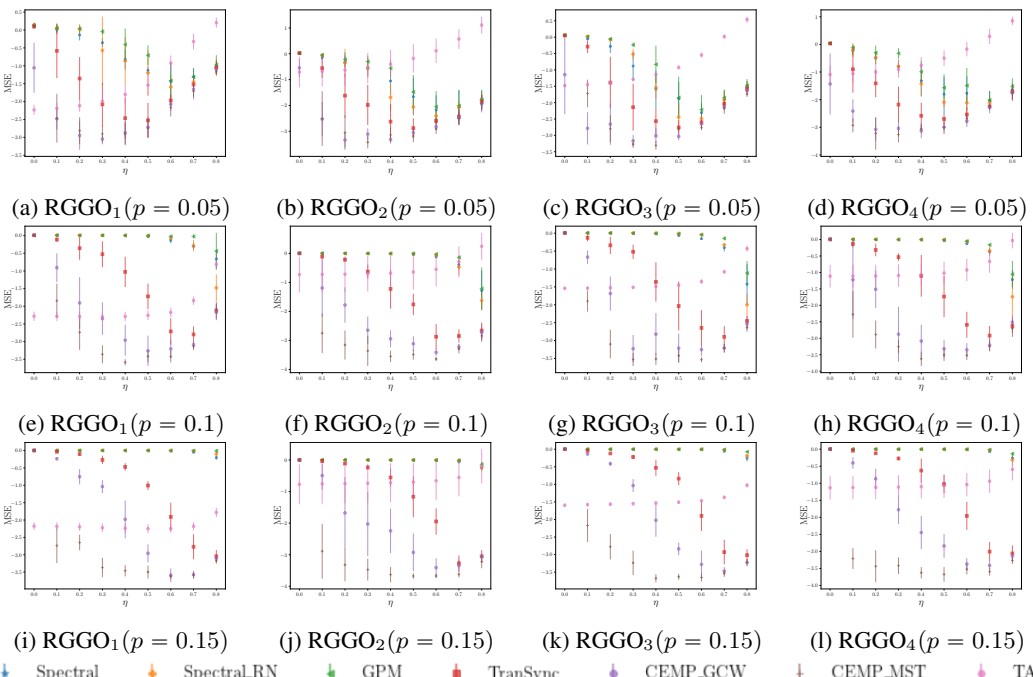

Figure 32: MSE performance improvement on GNNSync over variants on angular synchronization ($k = 1$) for RGGO models. $p$ is the network density and $\eta$ is the noise level. Error bars indicate one standard deviation.

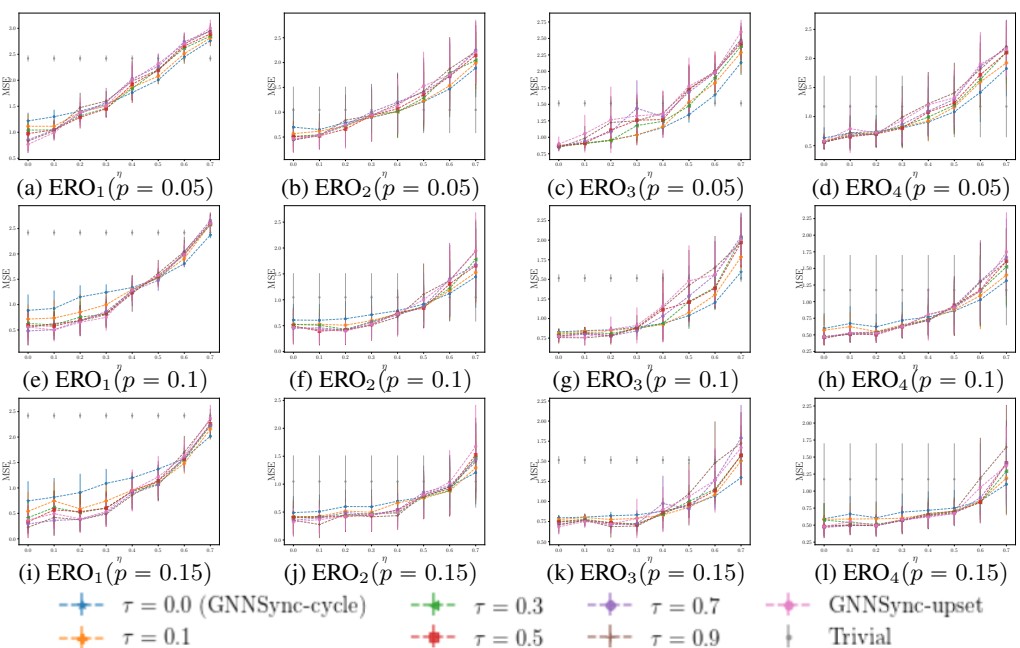

Figure 33: MSE comparison on GNNSync variants using as loss $\tau\mathcal{L}_{\text{upset}} + \mathcal{L}_{\text{cycle}}$ with different coefficients $\tau$, on $k-$synchronization with $k = 2$ for ERO models. $p$ is the network density and $\eta$ is the noise level. Error bars indicate one standard deviation.

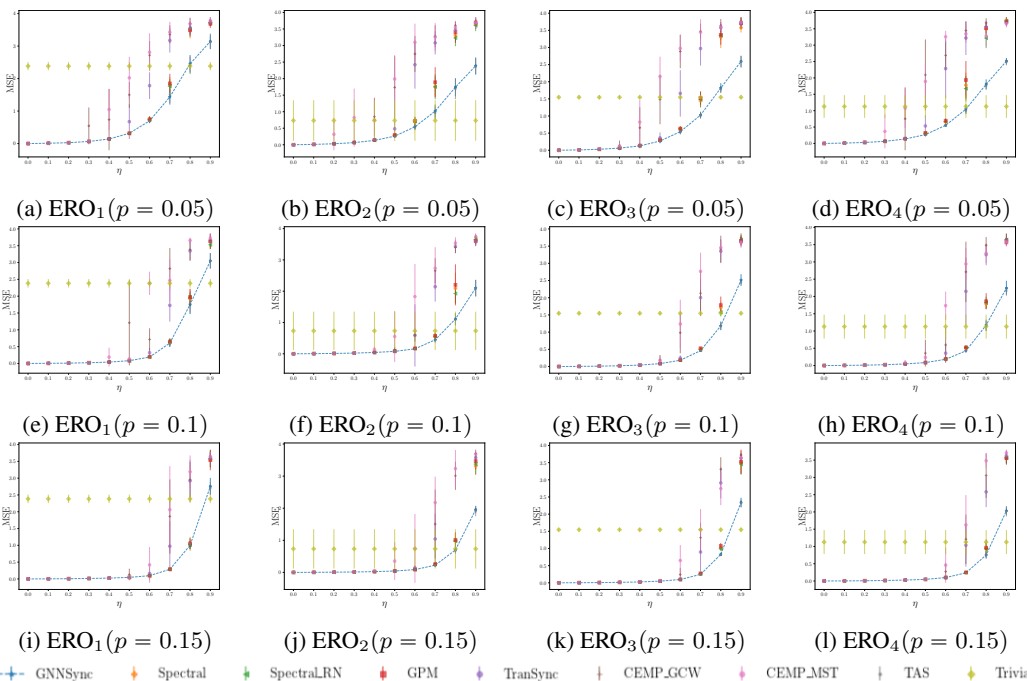

Figure 34: MSE performance comparison on GNNSync against fine-tuned baselines on angular synchronization ($k = 1$) for ERO models. $p$ is the network density and $\eta$ is the noise level. Error bars indicate one standard deviation.

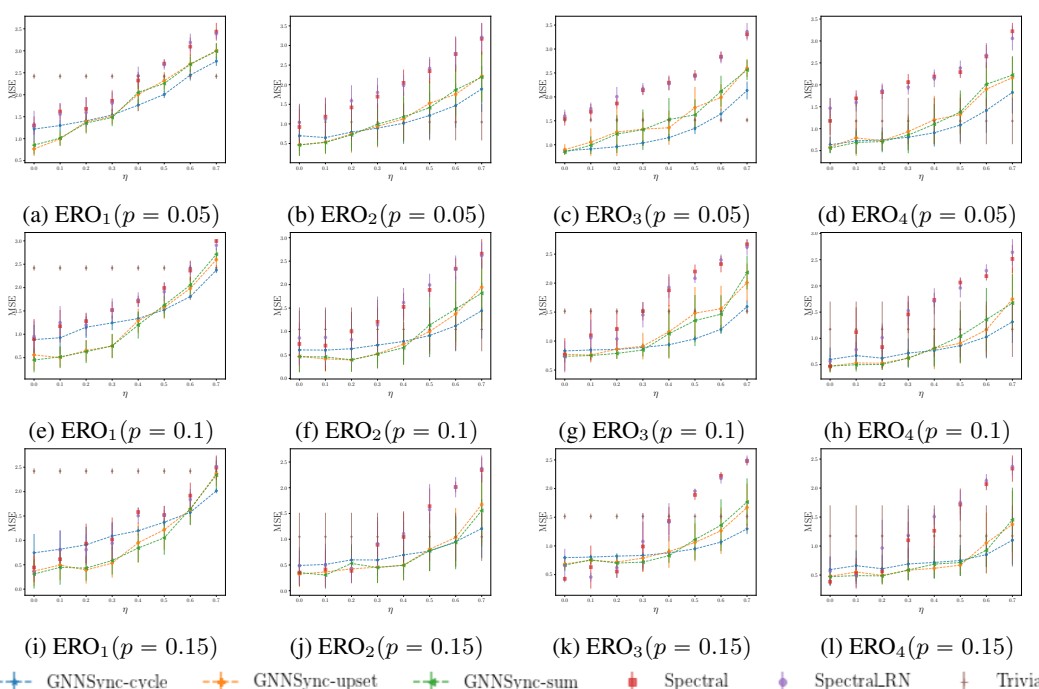

(a) ERO$_1(p = 0.05)$    (b) ERO$_2(p = 0.05)$    (c) ERO$_3(p = 0.05)$    (d) ERO$_4(p = 0.05)$

(e) ERO$_1(p = 0.1)$    (f) ERO$_2(p = 0.1)$    (g) ERO$_3(p = 0.1)$    (h) ERO$_4(p = 0.1)$

(i) ERO$_1(p = 0.15)$    (j) ERO$_2(p = 0.15)$    (k) ERO$_3(p = 0.15)$    (l) ERO$_4(p = 0.15)$

Figure 35: MSE performance comparison on GNNSync against fine-tuned baselines on $k-$synchronization with $k = 2$ for ERO models. $p$ is the network density and $\eta$ is the noise level. Error bars indicate one standard deviation.

