# OpenReview forum: "Robust Angular Synchronization via Directed Graph Neural Networks"
_ICLR.cc/2024/Conference — ICLR 2024 poster_

### Official Review · Reviewer_tjS3 · 2023-10-27

**Soundness:** 3 good
**Presentation:** 3 good
**Contribution:** 3 good
**Rating:** 8
**Confidence:** 2

**Summary:**

The paper addresses the problem of angular synchronization, which involves estimating a set of unknown angles from noisy measurements of their pairwise offsets. This problem appears in various applications, such as sensor network localization, phase retrieval, and distributed clock synchronization. The paper also extends the problem to a heterogeneous setting, named $k$-synchronization, where the goal is to estimate multiple groups of angles simultaneously from noisy observations.

To overcome the poor performance of existing methods in high-noise regimes, the authors propose GNNSync, a novel framework based on directed GNNs. The framework is end-to-end trainable and incorporates theoretically-grounded techniques to improve robustness to noise.

**Strengths:**

- The authors introduced GNNSync, a GNN-based method designed specifically for the angular synchronization problem and its heterogeneous extension, $k$-synchronization.
- I believe the main contribution of this paper resides in the proposal of new loss functions that encode synchronization objectives, with a particular focus on a cycle loss function that downweights noisy observations and enforces cycle consistency.
- There is extensive experimental validation for the proposed method, demonstrating that GNNSync outperforms existing state-of-the-art algorithms, especially in high noise scenarios, across various synthetic and real-world datasets.

**Weaknesses:**

I do not see significant weakness in this paper. In some cases regarding the performance of the proposed method, particularly in the case of BAO, where its performance does not significantly outperform the baseline methods. Would the authors be able to elaborate a bit on that?

**Questions:**

The authors mentioned that extending the current GNN-based framework from SO(2) to more general groups may introduce several challenges and complexities. I would like the authors to elaborate a bit on the potential difficulties in generalizing the current method.

---

> ### Author Response · Authors · 2023-11-16
>
> - "The authors mentioned that extending the current GNN-based framework from SO(2) to more general groups may introduce several challenges and complexities. I would like the authors to elaborate a bit on the potential difficulties in generalizing the current method."
>
> Response: Challenges that might arise in this direction concern group synchronization over non-compact groups, which has applications to sensor network localization (2D) and NMR spectroscopy (3D). For example, for the Euclidean group
> $Euc(2) = \mathbb{Z}_2 \times \mbox{SO}(2) \times \mathbb{R}^2$,
> existing approaches sequentially synchronize over $\mathbb{Z}_2,   \mbox{SO}(2)$ (or directly over $\mbox{O}(2)$), and finally over  $\mathbb{R}^2$. However, such a sequential approach is not optimal, as information on the pairwise  alignments over $\mathbb{R}^2$ can help improve synchronize over $\mathbb{Z}_2$ and $\mbox{SO}(2)$. To this end, a method that simultaneously synchronizes over the Euc(d) group directly could lead to an increase in the noise level tolerance. Another extension would be to the group $SO(d)$ for $d \geq 3$, where the difficulty stems from the fact that the pairwise measurement matrix would actually be a matrix whose entries are themselves $d \times d$ matrices, which calls for further methods for approximating the group elements from the estimated solution, and performing projection steps.

---

### Official Review · Reviewer_LD2r · 2023-11-08

**Soundness:** 3 good
**Presentation:** 3 good
**Contribution:** 2 fair
**Rating:** 6
**Confidence:** 2

**Summary:**

This paper studies the angular synchronization problem. It proposes GNNSync, which is a trainable framework using directed graph neural networks, to solve the problem. It also comes up with a new loss function to encode synchronization objectives. Numerical experiments are conducted to validate the superiority and robustness of GNNSync.

**Strengths:**

1. It incorporates the inductive biases of classical estimators within the design of GNNSync and casts the angular synchronization problem as a theoretically-grounded directed graph learning task.

2. It proposes a novel training loss that exploits cycle consistency to help disambiguate unknown angles.

**Weaknesses:**

It is unclear how to train GNNSync since the uncommon loss function in (5) is not smooth.

**Questions:**

1. The loss function (5) incorporates mod and min operation inside. Could the authors please clarify how to calculate the gradient of this loss?

2. Please clarify how to conduct projection in line 7 of Algorithm 1. Does the projection have a closed-form? How expensive is this projection step?

3. Why do you need to conduct several steps of projection per round?

---

> ### Author Response · Authors · 2023-11-16
>
> - "It is unclear how to train GNNSync since the uncommon loss function in (5) is not smooth."
>
> Response:
> Indeed this is not obvious. The key observation is that although the Frobenius norm, the min function, and modulo have non-differentiable points, these points have measure zero.  Discussions over the loss functions are provided in Appendix A.1, with some brief mentions in Prop. 1 in the main text. Moreover, as we use PyTorch autograd for gradient calculation, even in the presence of non-differentiable points, backpropagation can be carried out whenever an approximate gradient can be constructed. More details are given in the response for the next item. Finally, it may be reassuring that in our experiments, we do not observe any issue regarding convergence.
>
> - "The loss function (5) incorporates mod and min operation inside. Could the authors please clarify how to calculate the gradient of this loss?"
>
> Response: As stated in Appendix A.1, we use PyTorch autograd for gradient calculation. As the absolute value function is convex, autograd will apply the sub-gradient of the minimum norm.
> There also exist differentiable approximations for the modulo, and hence backpropagation can still be executed.
>
> - "Please clarify how to conduct projection in line 7 of Algorithm 1. Does the projection have a closed-form? How expensive is this projection step?"
>
> Response: The projection is conducted via torch.angle, which has a closed-form involving the arctan function. Indeed, the angle of a complex number $c = a + b\iota$ is the complex argument $arg(z)$ which for $a\neq 0$ is just arctan$(b/a)$; here $\iota$ is the imaginary unit. The computation is elementwise and is very efficient.
>
> - "Why do you need to conduct several steps of projection per round?"
>
> Response: As in the GPM method, the iterative steps are useful for the method to converge. In principle, we could use another number of steps other than 5 or set the number of steps as a hyperparameter, but we settle on a fixed value of 5 for simplicity.

---

### Official Review · Reviewer_UofK · 2023-11-09

**Soundness:** 3 good
**Presentation:** 2 fair
**Contribution:** 2 fair
**Rating:** 6
**Confidence:** 3

**Summary:**

This paper proposes a training framework (GNNSYNC) that incorporates directed graph neural networks to address the classical angular synchronization problem and its extended k-synchronization variant. Specifically, a directed GNN is first applied to learn
node embeddings which are used to generate an initial guess.  Then, projected gradient steps are applied to refine the solution.
Finally, the additional hyper parameters such as the feature matrix are learned by minimizing loss functions based on estimation error and cycle consistency. The proposed methods are evaluated by numerical experiments on synthetic and real datasets.

**Strengths:**

1. The paper claims that the proposed method is robust to the high noise level.
2. The paper devises new loss functions (upset/cycle) that allow to apply GNN techniques.
3. The paper extends the method to a more challenging heterogeneous setting.

**Weaknesses:**

1. While the paper claims that GNNSYNC is a theoretically grounded trainable framework. No theory is provided under any noise assumptions regarding 1) Can this method converge? 2) What kind of guarantee do we have (e.g. How close the solution provided by the algorithm is to the ground truth). While for standard algorithms such as GPM, we have well-established theoretical guarantees for certain types of noise.

2. While the paper performs extensive experiments, the comparison with the baseline under high noise level seems not convincing. When we have a certain noise level, the standard way is to first provide an initial guess by solving a generalized eigenvalue problem, then projecting to the SO(2) space and aligning with anchors (if they exist). Finally, Riemannian/projected gradient descent is used to finetune the solution by minimizing a loss function (which can be adjusted based on the noise level). It seems GNNSYNC consists of multiple stages while there is no further fine-tuning steps for the baseline methods.

Minors
* The paper mentions that the motivation for k-synchronization is practically interesting because of some applications in structural biology, but it seems no experiments are conducted on biological applications. Also, even the 'real-world' dataset is perturbed artificially, which makes it not easy to see if the method can generalize well.
* As mentioned by the authors, GNNSYNC shares many similarities with the GNNRank framework.
* Currently, GNNSYNC is limited to SO(2) group.

**Questions:**

1. It seems GNNSYNC itself is a complicated framework with many components. I am wondering if every component is necessary for it to solve the concise angular synchronization problem. Could you please also illustrate why the proposed method works well in high noise regimes while the other methods do not?

2. I would appreciate it if the authors could justify that the proposed method is exactly better than the standard method with fine-tuning (by using either the standard log-likelihood function or the newly designed loss functions.)

---

> ### Author Response · Authors · 2023-11-16
> **The first part of the initial response**
>
> - "While the paper claims that GNNSYNC is a theoretically grounded trainable framework. No theory is provided under any noise assumptions regarding 1) Can this method converge? 2) What kind of guarantee do we have (e.g. How close the solution provided by the algorithm is to the ground truth). While for standard algorithms such as GPM, we have well-established theoretical guarantees for certain types of noise."
>
> Response: While theoretical guarantees for GNNSync would be desirable, formal convergence results are typically not feasible for black-box deep GNN models. There are insightful theoretical results available for models with specific assumptions on the noise. For example, the GPM paper (reference [1] below) provides theoretical guarantees for i.i.d. noise such as white Gaussian noise when the noise level is not too high. In contrast, our paper does not make any assumptions on the noise distribution; Sec. 6.1 even includes examples with correlated noise, with the vector of ground truth angles generated by a multivariate normal of mean vector $(\pi, \ldots, \pi)$ and randomly generated covariance matrix ${\bf{w}} {\bf{w}}^T$. Here ${\bf{w}}$ is an independently generated standard normal vector; taking the product ${\bf{w}} {\bf{w}}^T$ gives a random matrix which is typically not diagonal. The synthetic data set also include non-Gaussian noise.
>
> From a practical viewpoint, theoretical guarantees of course do not guarantee the best performance. At high noise levels, GNNSync outperforms methods like GPM which come with theoretical guarantees. Moreover, we argue that theoretical grounding can take many different forms. Our paper establishes useful theoretical properties of GNNSync that directly support its application to the angular synchronization problem. Our theoretical results, Propositions 1 and 2, provide theoretical guarantees on some properties of the loss functions and on the robustness of GNNSync. More discussions of the loss function as well as robustness are provided in Appendix A.
>
> - "While the paper performs extensive experiments, the comparison with the baseline under high noise level seems not convincing. When we have a certain noise level, the standard way is to first provide an initial guess by solving a generalized eigenvalue problem, then projecting to the SO(2) space and aligning with anchors (if they exist). Finally, Riemannian/projected gradient descent is used to finetune the solution by minimizing a loss function (which can be adjusted based on the noise level). It seems GNNSYNC consists of multiple stages while there is no further fine-tuning steps for the baseline methods."
>
> Response: The papers of the baseline methods do not advise how to fine-tune their methods in the presence of high noise levels. GNNSync, on the other hand, includes project gradient descent steps; we note that the whole framework of GNNSync is integrated end-to-end instead of consisting of just a collection of multiple subsequent steps. Prompted by your concern, we have now added experiments to compare GNNSync with fine-tuned baselines whose outputs are given by adding the same number of project gradient steps as GNNSync to their initial outputs as a post-processing step. We observe in our revised paper, in particular in the new Figures 34 and 35, that GNNSync is indeed outperforming other methods, on average, even when fine-tuning is added to the baselines, probably due to the end-to-end trainable pipeline with our novel loss functions.
>
> - "It seems GNNSYNC itself is a complicated framework with many components. I am wondering if every component is necessary for it to solve the concise angular synchronization problem. Could you please also illustrate why the proposed method works well in high noise regimes while the other methods do not?"
>
> Response: Perhaps unfortunately, every component is indeed necessary, as illustrated in our ablation studies in Sec. 6.4. Intuitively, the proposed method works well even in high noise regimes mainly for two reasons.
> (1) The graph assignment of an edge in eq. (4) of the paper is estimated using only the smallest entries in the residual matrices. As long as not all residuals are corrupted, some signal will still be detected.
> (2) The confidence matrix $C$ which is used in the cycle loss in eq. (5)  of the paper downweighs edges with high residuals, thus minimizing the impact of noise.

---

> ### Author Response · Authors · 2023-11-16
> **The second part of the initial response**
>
> - "I would appreciate it if the authors could justify that the proposed method is exactly better than the standard method with fine-tuning (by using either the standard log-likelihood function or the newly designed loss functions.)"
>
> Response: There is no log-likelihood function available for GNNSync as the noise is not assumed to be of any parametric form.
> Thus, GNNSync is `better' than the standard method in the sense that it is applicable in a much wider setting.
> Finally, not only is GNNSync more widely applicable, but in our experiments, it also performs better than the baselines in many settings, see for example the new Figures 34 and 35 in the updated paper which include the added experiments that could be viewed as fine-tuning.
>
> - "The paper mentions that the motivation for k-synchronization is practically interesting because of some applications in structural biology, but it seems no experiments are conducted on biological applications. Also, even the 'real-world' dataset is perturbed artificially, which makes it not easy to see if the method can generalize well."
>
> Response: The phrasing in the paper was perhaps not completely clear; we have re-arranged it to emphasize the dependence on the different groups. Currently, the publicly available biological data sets are in SO(3), and hence we have not applied our SO(2) method to them.
> In terms of our SNL experiments, unfortunately, there are no public benchmarks available with ground truths, and competitive papers normally only conduct experiments on synthetic data sets; see for example references [1] and [2] below. The only 'real-world' data set we have found in [3] is the U.S. map data with added artificial noise, which is also 'semi-real' like in our case; and we indeed use this U.S. map data set, with further noise models. As also stated in Appendix B.2, papers on applications using angular synchronization, such as tracking the trajectory of a moving object
> using directional sensors and habitat monitoring in an infrastructureless environment, do not publish their data sets. Instead, we resort to creating a collection of synthetic data sets. Our synthetic data sets are constructed with both correlated and uncorrelated ground-truth angles, with various measurement graphs, to better mimic real-world scenarios, as discussed in Appendix B.1. We hope that making these data sets available can help fill the data paucity issue and that they will be used as benchmark data sets by future studies.
>
>
> - "As mentioned by the authors, GNNSYNC shares many similarities with the GNNRank framework."
>
> Response: They are similar in the sense that they both utilize the underlying directed graph structures. However, as detailed in the first paragraph of Sec. 2.3, they are very different in many fundamental aspects: problem definition (as well as extension to k-synchronization), loss functions, and the architecture after the directed graph neural network part.
>
> - "Currently, GNNSYNC is limited to SO(2) group."
>
> Response: This paper concentrates on SO(2) as this group is already nontrivial to deal with, and is a natural starting point for the group synchronization problems (see reference [2] below). However, the current framework could be extended to more complex structures, such as synchronization in Euc(2), or in more general SO(d) problems, which would involve more intricate edge information.  For the group $SO(d)$ for $d \geq 3$, the difficulty stems from the fact that the pairwise measurement matrix would actually be a matrix whose entries are themselves $d \times d$ matrices, which calls for further methods for approximating the group elements from the estimated solution, and performing projection steps. However, the pipeline is, in principle, extendable to this situation.
>
> References:
>
> [1] Boumal, N. (2016). Nonconvex phase synchronization. SIAM Journal on Optimization, 26(4), 2355-2377.
>
> [2] Singer, A. (2011). Angular synchronization by eigenvectors and semidefinite programming. Applied and computational harmonic analysis, 30(1), 20-36.
>
> [3] Cucuringu, M., Lipman, Y., Singer, A. (2012). Sensor network localization by eigenvector synchronization over the Euclidean group. ACM Transactions on Sensor Networks (TOSN), 8(3), 1-42.

---

> > ### Comment · Reviewer_UofK · 2023-11-20
> >
> > I thank the authors for providing detailed explanations. I have increased the score to 6.

---

> > > ### Author Response · Authors · 2023-11-20
> > >
> > > Thank you very much for your support and feedback!

---

### Official Review · Reviewer_BzAL · 2023-11-10

**Soundness:** 3 good
**Presentation:** 2 fair
**Contribution:** 2 fair
**Rating:** 5
**Confidence:** 2

**Summary:**

This paper considers the problem of angular synchronization, in which the goal is to compute one or multiple sets of angles given noisy measurements of their differences , which are given as a weighted directed graph.  The output is given mod $2\pi$ and up to an additive constant angle. The main contribution is in the case that the measurements are noisy. These errors lead to inconsistencies in the sum of angles that belong to directed cycles of the graph, which can have non-zero due to the noise. The loos function that is used to apply a projected gradient descent algorithm encodes these inconsistencies and thus the gradient descent algorithm is trying to minimize them. Furthermore, the authors have implemented the algorithm and run experiments on synthetic data and under various noise models to demonstrate the algorithm’s accuracy and robustness to noise in comparison with prior work.

**Strengths:**

The proposed algorithms outperform existing ones in the literature for high levels of noise.

**Weaknesses:**

The paper doesn’t seem to have a lot of technical novelty and depth.

**Questions:**

In the problem definition, the phrase “at most one of $A_{i,j}$ $A_{j,i}$ can be non-zero by construction” does not seem to be consistent with the definition above. Can you clarify?

---

> ### Author Response · Authors · 2023-11-16
>
> - "The paper doesn’t seem to have a lot of technical novelty and depth."
>
>
> Response: We apologize for perhaps not having demonstrated our contributions clearly enough via the bullet points at the end of Sec. 1; we will consolidate our contributions here to demonstrate our technical novelty and depth.
> 1) We demonstrate that the angular synchronization problem aligns well with the structure of a directed graph with pairwise angular offsets as the edges. This insight provides a framing which allows us to utilize directed graph neural networks to tackle the problem.
> 2) In addition, we add to our architecture inductive bias from previous work in angular synchronization by the projected gradient steps.
> 3) It does not suffice to formulate the problem and to create the architecture, but we also need suitable loss functions to train the models. The loss functions which we devise are novel and include geometric considerations; the cycle loss is based on the fact that the angles in a cycle sum to zero modulo $2 \pi$. Based on these loss functions, we can train the model in a completely unsupervised manner.
>
> - "In the problem definition, the phrase “at most one of $A_{i,j}, A_{j,i}$ can be non-zero by construction” does not seem to be consistent with the definition above. Can you clarify?"
>
> Response: Thank you for alerting us to this definition not being clearly formulated. Indeed we choose to keep one of $A_{i,j}$ and $A_{i,j}$ randomly, and set the other one to zero. We have revised our Sec. 3 accordingly. To clarify, the goal of this restriction is mainly to save computational complexity, as message passing and loss computation can be done on only half of the existing edges. By definition, as long as one of $A_{i,j}$ and $A_{j,i}$ is known, the other can be computed as their sum modulo $2\pi$ is zero ($A$ is skew-symmetric in the modulo $2\pi$ sense). This can be further explained via the construction procedure of our random graph outlier models detailed in Appendix B.1.
> 4) We also theoretically assess the properties of our novel loss function designs in  Prop. 1 and Appendix A.1, and show why GNN is a good choice by giving a theoretical foundation for its robustness in Prop. 2.

---

### Meta-Review · Area_Chair_QjZE · 2023-12-06

**Metareview:**

This paper considers the problem of angular synchronization, where the goal is to estimate angles from noisy measurements, a common problem in sensor localization and phase retrieval.
The authors introduce a novel solution, GNNSync, which uses directed graph neural networks to learn node embeddings that help produce an initial estimate. Then, using projected gradient steps, they refine this solution. Tests on synthetic and real data demonstrate the robustness and a better performance of GNNSync over other baseline methods in the high noise regime.

Although this paper does not derive theoretical guarantees for the proposed method, the reviewers found that proposing a loss function that allows to use GNNs to address this (theoretically challenging) problem from a practical perspective is a contribution that can be of interest to the ICLR audience.

The additional numerical comparisons against fine-tuned baselines indicate that the magnitude of improvement is somewhat less pronounced. However, the method still exhibits commendable performance. The paper could stimulate further scholarly exploration of the angular synchronization problem in high-noise situations. This could lead to theoretical advancements or the development of novel, application-oriented techniques.

**Justification For Why Not Higher Score:**

For a spotlight, I would have loved to see a bit broader numerical evaluations, or additional theoretical insights.

**Justification For Why Not Lower Score:**

Although there was no consensus among the reviewers, the ICLR community might be interested in the modelling approach and the numerical results.

---

### Decision · Program_Chairs · 2024-01-16

Accept (poster)